# PDMBench: A Standardized Platform for Predictive Maintenance Research

## Abstract

Predictive maintenance (PdM) is critical for industrial reliability and cost-efficiency, yet fragmented datasets, inconsistent evaluation protocols, and incompatible preprocessing pipelines hinder progress. We introduce **PDMBench**, a standardized and extensible platform for exploring and evaluating machine learning models on multimodal time-series data across diverse industrial predictive maintenance settings. PDMBench integrates 14 curated datasets spanning bearings, motors, gearboxes, and multi-component systems, capturing real-world complexities such as irregular sampling, heterogeneous sensor modalities, and varying fault modes. To enable fair and reproducible comparison, we design a unified and configurable preprocessing pipeline that normalizes signal quality, extracts consistent features, and standardizes input representations, bridging the gap between models requiring handcrafted features and those operating on raw sequences. The benchmark covers two core tasks, fault classification and remaining useful life prediction, and includes 22 models ranging from traditional classifiers to cutting-edge transformers. Models are evaluated across three dimensions: prediction, uncertainty, and efficiency. The **PDMBench** web interface supports interactive dataset exploration, model comparison, and diagnostic analysis. Experimental results reveal no universal best model, with performance varying by dataset, task, and component type, underscoring the importance of standardized benchmarking. **PDMBench** enables rigorous, scalable, and interpretable research for real-world predictive maintenance by aligning data, models, and metrics in a reproducible platform[1].

## 1 Introduction

Predictive Maintenance (PdM) has become essential for reducing downtime and enhancing safety across manufacturing, energy, healthcare, and transportation sectors Çınar et al. (2020); Zhu et al. (2019). By leveraging continuous sensor monitoring and data analytics, PdM enables early detection of equipment degradation, allowing operators to intervene before catastrophic failures occur Ran et al. (2019). The rise of the Industrial Internet of Things (IIoT) has made this vision practical: edge computing platforms now enable real-time collection of multimodal sensor data - vibration, temperature, current, acoustic signals - from industrial equipment at scale Kanaway & Sane (2017). This sensor-rich environment has driven the development of increasingly sophisticated machine learning approaches for fault prediction. Traditional methods rely on carefully engineered features extracted from time and frequency domains Wang et al. (2020); Kaparthi & Bumblauskas (2020), while modern deep learning architectures attempt to automatically discover fault signatures directly from raw sensor streams Li et al. (2024); Wu et al. (2023); Wang et al. (2024b). Despite these modeling advances, a fundamental challenge persists: researchers evaluate methods on different datasets using inconsistent preprocessing and incompatible metrics, making fair comparison nearly impossible and undermining confidence in how well approaches generalize to real-world deployment Zhu et al. (2019).

This fragmentation is particularly problematic because PdM operates in a uniquely complex problem space. As illustrated in Figure 1, real-world PdM systems must simultaneously address multiple

---

[1]The platform's codebase is publicly available on GitHub at `https://anonymous.4open.science/r/PDMBenchmark-C811/`, and the dataset can be accessed at `https://huggingface.co/submission096`.

interconnected challenges. On the data side, industrial sensors generate multimodal streams at vastly different temporal resolutions—vibration signals sampled at 50kHz, temperature readings at 1Hz, telemetry logs collected hourly—while equipment operates under diverse conditions that produce non-stationary signal distributions. On the task side, practitioners require both fault classification for diagnosis ("which component failed and how?") and remaining useful life (RUL) prediction for maintenance scheduling ("how many operating hours remain?"). On the evaluation side, deployment-ready systems need not only high prediction accuracy but also well-calibrated uncertainty estimates for risk assessment and computational efficiency for edge deployment. This multifaceted complexity makes standardized benchmarking both critical and challenging.

Addressing this gap requires tackling three interrelated challenges that current benchmarking practices fail to resolve systematically. **(C1) Data complexity:** PdM data is inherently multimodal, irregularly sampled, and highly sensitive to equipment type and load conditions Zhang et al. (2025); Jung et al. (2023), yet most studies evaluate models on single type of datasets under controlled conditions, leaving cross-domain robustness unvalidated. **(C2) Model compatibility:** State-of-the-art architectures must generalize across heterogeneous sensor configurations and fault types Hao et al. (2020); Ding et al. (2022), but inconsistent preprocessing such as raw signals versus handcrafted features, dataset-specific normalization prevents fair architectural comparison. **(C3) Evaluation fragmentation:** Domain experts require interpretable outputs, calibrated uncertainties, and edge-compatible efficiency Juodelyte et al. (2022), but existing benchmarks often report only prediction accuracy. This fragmentation has concrete consequences: fault classification studies on CWRU Smith & Randall (2015) using cross-validation cannot be compared with RUL prediction on FEMTO Nectoux et al. (2012a) using temporal splits or transfer learning on Paderborn Lessmeier et al. (2016), leaving fundamental questions unresolved—whether bearing-trained models generalize to motors, how preprocessing affects robustness, whether accurate models meet deployment constraints, and which architectures succeed across task types Zhu et al. (2019).

We introduce **PDMBench**, a standardized and extensible platform that directly addresses these challenges through a principled three-tier architecture (Figure 2). The **Data Level** unifies 14 publicly available datasets spanning bearings, motors, gearboxes, and multi-component systems which cover the sensor modalities, fault types, and operating conditions shown in Figure 1 through consistent preprocessing pipelines that preserve signal fidelity while enabling fair model comparison. The **ML Level** implements 22 diverse models ranging from classical baselines (SVM, XGBoost) to state-of-the-art transformers (TimesNet Wu et al. (2023), PatchTST Nie et al. (2023)), evaluating each across the three critical dimensions identified in Figure 1: prediction accuracy, uncertainty calibration, and computational efficiency. The **User Level** provides an interactive web interface for dataset exploration, model configuration, and interpretable diagnostics, bridging the gap between algorithmic research and practitioner needs.

Unlike some existing time-series benchmarks such as M4 De Santis et al. (2023) and UCR Baldán & García-Gil (2025), which focus on univariate forecasting or anomaly detection under clean, regularly sampled conditions, PDMBench embraces the messy reality of industrial PdM: irregular sampling rates, imbalanced fault distributions, and deployment constraints Jimenez et al. (2020). Our comprehensive evaluation reveals that no single model dominates across all settings—transformer-based architectures excel on structured bearing datasets but struggle with noisy motor signals, while lightweight models offer compelling accuracy-efficiency trade-offs for edge deployment. These findings underscore both the value of standardized benchmarking and the need for context-aware model selection in real-world PdM applications.

Our contributions can be summarized as follows:

- **A curated dataset suite** spanning 14 datasets across fault types, sensor modalities, and operational regimes.
- **A unified toolbox** for preprocessing, training, and evaluation across both handcrafted-feature and end-to-end models.
- **A comprehensive evaluation framework** encompassing accuracy, uncertainty (e.g., ECE, NLL, Brier Score), and efficiency (e.g., inference time, memory).
- **An interactive web interface** that supports explainability, model diagnosis, and practitioner involvement.

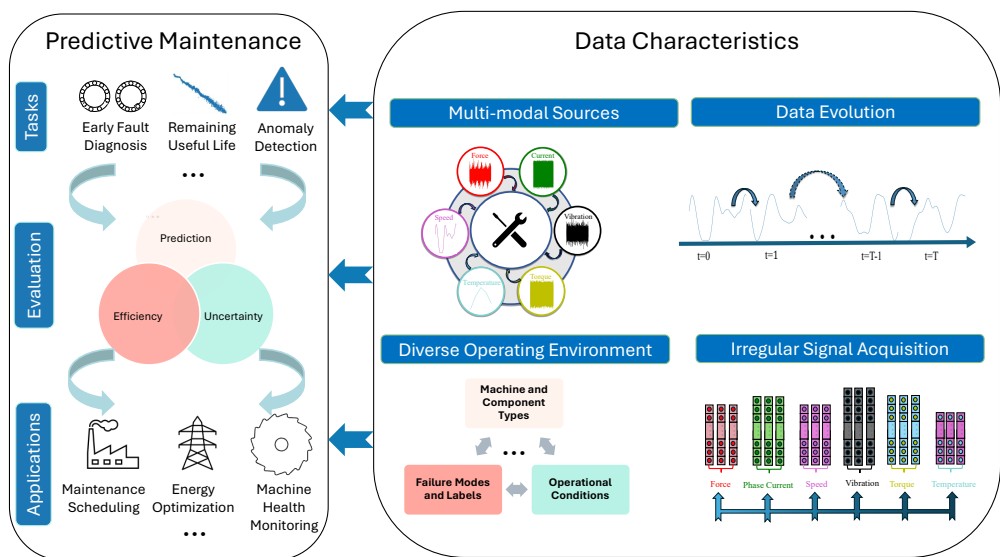

Figure 1: **Challenges and components in PdM research.** Multimodal, irregular sensor data collected under varying conditions feeds into core PdM tasks—fault diagnosis and RUL prediction, evaluated by accuracy, uncertainty, and efficiency, with direct implications for maintenance planning, energy use, and system health.

## 2 RELATED WORK

We organize related work across three main areas: predictive maintenance modeling, benchmark datasets, evaluation frameworks, and interactive tools for interpretable diagnostics.

**Predictive Maintenance and Modeling Paradigms.** PdM aims to anticipate equipment failures using sensor-derived signals, enabling proactive interventions in industrial settings Zhu et al. (2019); Ran et al. (2019). Traditional approaches rely on statistical techniques or decision trees Kaparthi & Bumblauskas (2020); Wang et al. (2020), but have limited scalability across heterogeneous machinery and conditions. The rise of IIoT has facilitated large-scale data collection Kanawaday & Sane (2017), allowing machine learning (ML) and deep learning (DL) models to flourish Efeoğlu & Tuna (2022); Rivas et al. (2020); Li et al. (2024). Recent DL advances include CNN-LSTM hybrids Hao et al. (2020), contrastive pretraining Kong et al. (2023), and transformer-based architectures such as TimeXer Wang et al. (2024b), TimeMixer Zhang et al. (2024b), TimesNet Wu et al. (2023), and FEDformer Zhou et al. (2022a). Despite promising accuracy, many of these models struggle with multimodal signals and domain shifts, necessitating benchmark-driven evaluation.

**Benchmark Datasets and Evaluation Frameworks.** PdM research heavily relies on a set of canonical datasets, particularly for bearing and motor fault diagnosis. These include CWRU Smith & Randall (2015), XJTU-SY Lei et al. (2019), FEMTO-PRONOSTIA Nectoux et al. (2012b), IMS Sacerdoti et al. (2023), HUST Thuan & Hong (2023), and MFPT for Machinery Failure Prevention Technology. Datasets for gearboxes (e.g., UoC Cao et al. (2018), WT Planetary Gearbox Liu et al. (2023a)) and multi-component systems (e.g., MAFAULDA Laboratory of Rotating Machinery Diagnostics, UFRJ (2000), Microsoft Azure Microsoft) have also emerged. However, most prior studies use these datasets in isolation with inconsistent preprocessing, segmentation, and evaluation metrics Jimenez et al. (2020); Kavasidis et al. (2023), undermining fair comparison and generalization. Benchmarks like M4 De Santis et al. (2023) and UCR Baldán & García-Gil (2025) address adjacent domains (e.g., forecasting, anomaly detection), but lack the modality diversity and diagnostic realism needed for PdM. Recent benchmarking efforts for time-series models (e.g., Autoformer Wu et al. (2021), Informer Zhou et al. (2021), Pyraformer Liu et al. (2021a), SCINet Liu et al. (2021b), MICN Wang et al. (2023)) have advanced model development, yet are not tailored for irregularly sampled, high-dimensional sensor signals common in industrial applications. Moreover, newer architectures like FiLM Zhou et al. (2022b), Crossformer Zhang et al. (2023), and iTransformer Liu et al. (2023b) em-

phasize long-range dependencies and frequency-aware modeling, but their deployment still requires manual tuning and format alignment, which PDMBench addresses via a standardized pipeline.

**AutoML and Benchmarking Platforms.** While general ML workflow tools like MLflow Chen et al. (2018), Scikit-learn Pedregosa et al. (2011), and time-series libraries such as SKtime Löning et al. (2019) provide algorithms and experiment tracking capabilities, they lack the domain-specific evaluation infrastructure that PdM research requires. AutoRUL Hoff et al. (2023) represents a significant advance by applying automated machine learning to RUL prediction, demonstrating that automated model selection can achieve competitive performance on individual datasets through ensemble methods and hyperparameter optimization. However, AutoRUL optimizes for per-dataset performance and does not address the evaluation fragmentation problem we document: as Huang et al. Huang et al. (2024) quantified, preprocessing choices alone create 26% RMSE differences on identical datasets, and the CRULE benchmarking framework Chen et al. (2024a) revealed that only one domain adaptation method showed statistically significant improvement under fair evaluation. Similarly, while MLEBench Chan et al. (2024) evaluates AI agents on diverse ML engineering tasks from Kaggle competitions, it focuses on agent capabilities rather than domain-specific method evaluation. PDMBench complements these efforts by providing what multiple surveys have identified as missing Ramasso & Saxena (2014); Wang et al. (2024a); Chen et al. (2024b): standardized cross-domain evaluation that reveals generalization patterns (e.g., our Figure 4 shows models achieving 98% F1 on bearings dropping to 67% on motors), curated fault-labeled datasets with documented preprocessing protocols, and calibration metrics essential for safety-critical maintenance decisions—capabilities that general AutoML tools and agent benchmarks do not provide.

**Interpretable Interfaces and Human-AI Collaboration.** While model performance remains a focus, real-world deployment of PdM systems also hinges on interpretability and usability for domain experts Manchadi et al. (2023). Few existing benchmarks offer tools to support human-in-the-loop decision-making, despite growing recognition of its value in high-stakes applications such as manufacturing and healthcare Çoban et al. (2018); Kavasidis et al. (2023). PDMBench fills this gap through an interactive visualization and diagnostic interface, enabling practitioners to explore raw signals, analyze model attention, and perform comparative evaluations, advancing the goal of trustworthy AI for maintenance.

In summary, while prior work has laid the foundation for PdM modeling and dataset collection, PDMBench provides the first unified platform that integrates diverse datasets, harmonized pipelines, model baselines, and interpretability tools in a single reproducible and extensible platform, paving the way for scalable and transparent PdM research.

## 3 PRELIMINARIES

We formalize the PdM problem setting by introducing key definitions central to **PDMBench**.

**Definition 1** (Sensor Modality Time Series). *Let $\mathbf{X}_m = [x_m^1, x_m^2, \ldots, x_m^T] \in \mathbb{R}^T$ denote the univariate time series collected from the $m$-th sensor modality, where each $x_m^t \in \mathbb{R}$ represents the recorded value at time step $t$.*

**Definition 2** (Segment). *A segment $\mathbf{x} \in \mathbb{R}^{M \times L}$ is a fixed-length window of $L$ consecutive time steps extracted from a multimodal time series comprising $M$ sensor modalities, where $L \leq T$.*

**Definition 3** (PdM Dataset). *A labeled dataset $\mathcal{D} = \{(\mathbf{x}_i, y_i)\}_{i=1}^N$ consists of $N$ multimodal segments $\mathbf{x}_i \in \mathbb{R}^{M \times L}$ and their corresponding task-specific labels $y_i \in \mathcal{Y}_{\mathcal{T}}$.*

**Definition 4** (PdM Tasks). *PDMBench supports two core predictive maintenance tasks:*

- *$\boldsymbol{\textit{Fault Classification (CLF)}}$: A classification task $\mathcal{T}_{clf}$ where each segment is mapped to a discrete fault category. The label space is defined as $\mathcal{Y}_{clf} = \{1, 2, \ldots, K\}$ for $K$ unique fault classes.*
- *$\boldsymbol{\textit{Remaining Useful Life Estimation (RUL)}}$: A regression task $\mathcal{T}_{rul}$ where each segment is mapped to a continuous value $y \in \mathbb{R}_+$ denoting the number of operational cycles remaining until failure.*

**Definition 5** (PdM Model). *A PdM model is a function $f_\theta : \mathbb{R}^{M \times L} \to \mathcal{Y}_{\mathcal{T}}$, parameterized by $\theta$, that maps an input segment to a prediction corresponding to task $\mathcal{T}$.*

**PDMBench** aims to evaluate learning systems under a standardized and reproducible setting for multimodal PdM data. Each benchmark instance is defined as a tuple $(\mathcal{D}, \mathcal{T}, \mathcal{E})$, where $\mathcal{D}$ represents

a preprocessed dataset containing standardized segments and task-specific labels, $\mathcal{T}$ denotes the downstream task to be performed, either fault classification (CLF) or remaining useful life estimation (RUL), and $\mathcal{E}$ specifies the evaluation protocol, including the metrics used to assess model performance. Models $f_\theta : \mathbf{x} \mapsto \hat{y}$ are trained to minimize task-specific loss functions $\mathcal{L}_\mathcal{T}(\hat{y}, y)$, such as cross-entropy for classification or mean squared error for RUL prediction. **PDMBench** further evaluates each model's robustness, calibration, and efficiency to facilitate fair and interpretable comparisons across diverse tasks and datasets.

**Problem 1.** *Multimodal Predictive Maintenance Benchmarking.*
*Given:* A diverse collection of multivariate time series datasets $\{\mathcal{D}_j\}_{j=1}^J$, where each $\mathcal{D}_j$ comprises multimodal signal segments $\mathbf{x} \in \mathbb{R}^{M \times L}$ and associated labels $y \in \mathcal{Y}_\mathcal{T}$ for task $\mathcal{T} \in \{CLF, RUL\}$.
*Find:* A unified evaluation suite for training and comparing models $f_\theta$ across all datasets and tasks, under consistent preprocessing pipelines, segmentation strategies, and evaluation metrics.

### 3.1 GAPS IN EXISTING BENCHMARKS AND THE NEED FOR **PDMBENCH**

**Fragmentation Across Datasets.** PdM research has traditionally relied on domain-specific datasets, particularly for rotating machinery components like bearings. Common benchmarks include CWRU Smith & Randall (2015), Paderborn Lessmeier et al. (2016), XJTU-SY Lei et al. (2019), FEMTO Nectoux et al. (2012b), and HUST Thuan & Hong (2023), with additional datasets from IMS (IMS), MFPT for Machinery Failure Prevention Technology, and Ottawa Sehri & Dumond (2023). Despite their utility, these datasets are used in isolation with inconsistent preprocessing, divergent feature extraction, and incompatible metrics, limiting fair comparisons and generalizability Zhu et al. (2019). They also differ in design: CWRU and Paderborn contain seeded faults under controlled conditions, while PRONOSTIA and XJTU-SY capture natural degradation in run-to-failure experiments. Beyond bearings, datasets focused on gearboxes Zhang et al. (2024a) and induction motors Treml et al. (2020) introduce further variation in failure types, sampling rates, and sensor modalities. This heterogeneity, though valuable, poses a major challenge for building scalable, unified PdM solutions.

**Barriers to Model Transferability.** The rise of deep learning has introduced powerful architectures for time series classification and forecasting, including CNNs, transformers, and hybrids. Models such as TimeXer Wang et al. (2024b), TimeMixer Zhang et al. (2024b), TimesNet Wu et al. (2023), FEDformer Zhou et al. (2022a), and others have achieved promising results in fault classification and RUL prediction. However, many of these models are optimized for clean, regularly sampled, single-modal data and do not naturally extend to the irregular, multimodal inputs common in industrial PdM settings. Their deployment often depends on dataset-specific preprocessing, such as input alignment or resampling, which compromises reproducibility. Furthermore, their high sensitivity to hyperparameters and dependence on clean input distributions make them difficult to generalize across datasets without extensive tuning. As a result, despite their architectural sophistication, the lack of a standardized preprocessing and evaluation platform continues to hinder reliable benchmarking and model deployment in PdM applications.

**From Problem Formulation to Platform Design.** Having formalized the PdM benchmarking problem, we now present **PDMBench**, a standardized and extensible benchmarking platform for predictive maintenance research. **PDMBench** is designed to systematically address the three core challenges identified in the Introduction: data complexity, model compatibility, and the need for effective human-AI collaboration. Formally, each benchmark instance is defined as PDM$(\mathcal{D}, \mathcal{P}, \mathcal{E})$, where $\mathcal{D}$ denotes the unified dataset collection, $\mathcal{P}$ the standardized preprocessing pipeline, and $\mathcal{E}$ the multi-dimensional evaluation suite encompassing accuracy, uncertainty, and efficiency metrics.

## 4 PDMBENCH DEVELOPMENT

To systematically address the three foundational challenges in PdM: *data complexity* (C1), *model compatibility* (C2), and *human-machine collaboration* (C3), we present **PDMBench**, a principled benchmarking platform built on three tightly integrated architectural layers. The **Data Level** harmonizes heterogeneous sensor streams through standardized preprocessing while preserving fault-relevant signal characteristics. The **ML Level** enables fair comparison across 22 diverse models from traditional classifiers to state-of-the-art transformers through unified training protocols and triadic evaluation (accuracy, calibration, efficiency). The **User Level** bridges research and practice

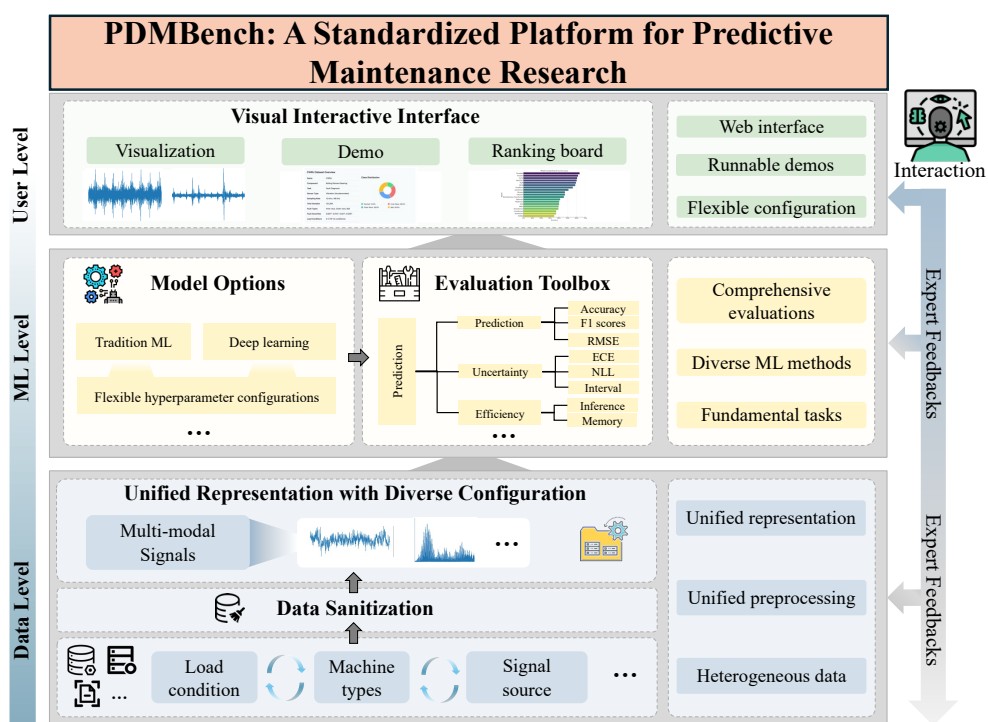

Figure 2: **PDMBench System Design.** The platform integrates heterogeneous sensor data at the *Data Level* through standardized preprocessing and representation. The *ML Level* enables fair model comparison through unified training and multi-perspective evaluation. The *User Level* provides a highly interactive interface for visualization, configuration, and interpretability, fostering human-AI synergy in PdM decision-making.

via interactive tools for dataset exploration, model diagnosis, and interpretable predictions. Together, these layers create a reproducible infrastructure that reflects real-world PdM complexities while supporting rigorous algorithmic comparison.

## 4.1 DATA LEVEL: HARMONIZING HETEROGENEOUS SENSOR STREAMS

The real-world landscape of PdM is characterized by extraordinary sensor heterogeneity. Sensor modalities vary significantly across applications, ranging from vibration and current to temperature, acoustic signals, and telemetry logs, and are recorded at sampling frequencies that span several orders of magnitude. Additionally, fault patterns differ by component type, and data collection conditions vary between controlled laboratory setups and real-world deployments. To manage this complexity, PDMBENCH integrates *14 publicly available datasets* categorized across four major mechanical subsystems: bearings, motors, gearboxes, and multi-component systems.

**Bearing fault datasets** include: CWRU Smith & Randall (2015), FEMTO Nectoux et al. (2012b), HUST Thuan & Hong (2023), IMS (IMS), MFPT for Machinery Failure Prevention Technology, Mendeley Bearing Kechik et al. (2020), Paderborn Lessmeier et al. (2016), and XJTU-SY Lei et al. (2019). These datasets span both fault classification and run-to-failure RUL prediction settings, covering vibration signals at sampling rates ranging from 9.6 kHz to 97.6 kHz.

**Motor fault datasets** include two sources: the Electric Motor dataset Sehri & Dumond (2023), which features multi-sensor recordings at 42 kHz, and the Rotor Broken Bar dataset Treml et al. (2020), focused on current signal faults in three-phase induction motors, sampled at 50 kHz. These datasets reflect the high-frequency and low signal-to-noise characteristics common to PdM diagnostics.

**Gearbox fault datasets** include the UConn Gearbox Cao et al. (2018), and WT Planetary Gearbox Liu et al. (2023a). Each captures degradation patterns under different load conditions and sensing setups, contributing critical diversity to the evaluation of mechanical fault generalization.

**Multi-component system datasets** include MAFAULDA Laboratory of Rotating Machinery Diagnostics, UFRJ (2000), a synthetic testbed simulating fault types across components using vibration signals, and the Microsoft Azure dataset Microsoft, which contains telemetry, sensor logs, and event-based RUL annotations sampled at hourly intervals. These datasets extend the benchmark to cover industrial-scale monitoring systems involving asynchronous modalities.

To ensure comparability across this diverse collection, PDMBENCH uses two types of preprocessing pipelines, a configurable preprocessing pipeline, and a unified preprocessing pipeline. The model-specific pipeline is proposed to preserve the unique processing of each method, and the unified pipeline is proposed to ensure fair comparison among all the methods. For the configurable preprocessing pipeline, we inherited the preprocessing methods from the original papers. While for the unified preprocessing pipeline, includes preprocessing steps such as normalization, segmentation. Particularly, all raw signals are normalized to zero mean and unit variance to mitigate differences arising from sensor calibration and mounting conditions. Signals are then segmented into fixed-length non-overlapping windows, enabling consistent input dimensions across models. However, the fixed-length non-overlapping window is determined based on the performance of each baseline on each benchmark.

As for features, we extract handcrafted features spanning the time, frequency, and envelope domains for traditional machine learning methods, while for deep learning models, we just follow the original papers and preserve the raw signals. All datasets are reformatted into structured triplets $(\mathbf{X}_i, y_i, \mathbf{m}_i)$, where $\mathbf{X}_i$ denotes the raw or feature-extracted segment, $y_i$ is the task-specific label, and $\mathbf{m}_i$ encodes contextual metadata. This abstraction ensures consistent interfacing with diverse model architectures while preserving diagnostic fidelity. We perform hyperparameter tuning using grid search, and we do not introduce any additional tuning strategies beyond this. However, whenever the original papers specify a particular procedure for hyperparameter selection, we adhere to those recommended settings in our experiments. Due to the high computation cost for all the baselines on all benchmarks, we report the results only on one fixed set of seeds. This ensures that all the baseline methods share the same training, validation, and test samples. And we chronologically split the whole dataset into training(60%), validation(10%), and test(30%) samples to ensure there is no data leakage. For the details about the preprocessing parameters, please refer to our GitHub.

## 4.2 ML Level: Benchmark Models and Standardized Evaluation

To facilitate comprehensive and reproducible evaluation, PDMBENCH includes 22 time-series models drawn from a broad spectrum of modeling paradigms. These range from traditional baselines to state-of-the-art transformer variants, each selected for its relevance to time-series classification and forecasting in industrial contexts.

Traditional baselines such as MLP Bengio et al. (2003), XGBoost Chen & Guestrin (2016), SVM Hearst et al. (1998), LSTM Graves & Graves (2012), and DLinear Zeng et al. (2023) offer global feedforward and decomposable linear processing capabilities, respectively. Transformer-based architectures include Autoformer Wu et al. (2021), which employs auto-correlation mechanisms for long-term forecasting, and Crossformer Zhang et al. (2023), designed to capture cross-dimensional dependencies. FEDformer Zhou et al. (2022a) and FiLM Zhou et al. (2022b) use frequency-domain representations, while Informer Zhou et al. (2021) introduces sparse attention for efficiency. Other models include the iTransformer Liu et al. (2023b), tailored for multivariate sequences via inverted attention, and the Reformer Kitaev et al. (2020), which enables efficient learning on extremely long sequences using locality-sensitive hashing. Hybrid and hierarchical models include TimesNet Wu et al. (2023), combining CNNs and transformers for 2D-periodicity, and SCINet Liu et al. (2021b), which applies sample interaction in a hierarchical setting. Recent architectures with strong multiscale properties include TimeXer Wang et al. (2024b), TimeMixer Zhang et al. (2024b), and MICN Wang et al. (2023), while lightweight yet expressive learners such as PatchTST Nie et al. (2023), PAttn Tan et al. (2024), and FreTS Yi et al. (2023) utilize patch-based and spectral-domain designs.

All models are trained under a standardized pipeline with consistent dataset splits, preprocessing, hyperparameter search, and early stopping strategies. To assess real-world applicability, models are evaluated along three axes: predictive performance (accuracy, F1, RMSE, MAE), uncertainty calibration (ECE, NLL, Brier Score), and computational efficiency (inference time, memory usage).

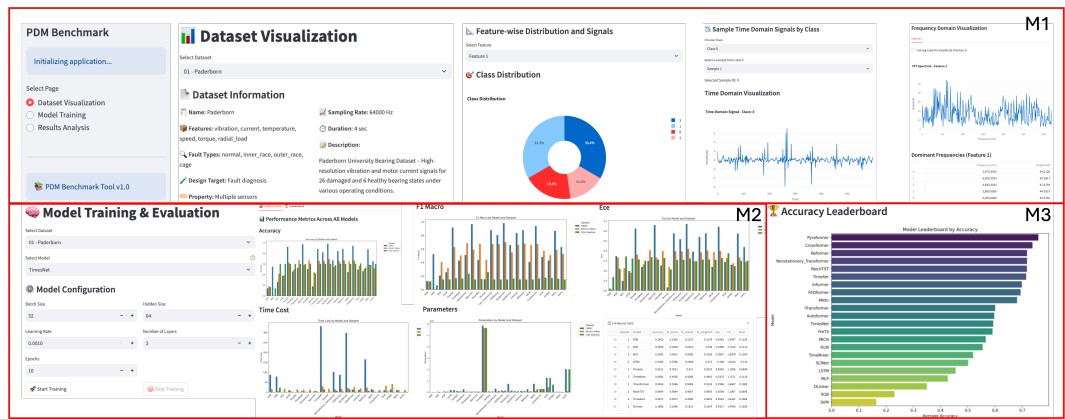

Figure 3: **PDMBench Interactive Interface .** The platform includes (M1) dataset visualization with time and frequency plots, (M2) model configuration and training diagnostics, and (M3) evaluation dashboards with sortable leaderboards and interpretability tools. Together, these modules support end-to-end experimentation and decision support.

This triadic evaluation protocol ensures a nuanced comparison that captures not only task-specific accuracy but also reliability and deployability in practical settings.

## 4.3 USER LEVEL: INTERACTIVE AND INTERPRETABLE AI INTERFACE

Most prior PdM benchmarks focus solely on model accuracy, offering little support for usability or interpretability, two critical factors in real-world decision-making environments. In contrast, PDMBENCH introduces a full-featured, web-based interface that transforms benchmarking into an accessible, explainable, and collaborative process. As shown in Figure 3, the platform is composed of three interactive modules designed to support exploration, configuration, and evaluation.

**The Exploration Module (M1)** enables users to visually examine datasets before modeling. It provides summaries of key dataset attributes, class distributions, and sensor properties, alongside time-domain and frequency-domain signal plots. Users can interactively filter by class, visualize dominant frequency components, and assess data quality through signal traces, facilitating intuition-building for domain experts.

**The Configuration Module (M2)** serves as an interface for model selection and training. It allows users to choose from a range of models, tune hyperparameters such as batch size and learning rate, and monitor real-time training diagnostics, including loss curves and attention maps. This modular design enables both novice and expert users to customize experiments while ensuring reproducibility.

**The Evaluation Module (M3)** functions as a comprehensive leaderboard and diagnostic hub. It presents a comparative view of models ranked by task-specific performance, calibration quality, and efficiency. Users can sort, filter, and analyze results across datasets, drill into model-specific metrics, and explore visual explanations such as attention heatmaps and feature importance distributions. This transparency builds trust and enables grounded interpretation in high-stakes settings.

Through its combination of standardized evaluation metrics, comprehensive analysis, and interactive visualization tools, PDMBENCH enables systematic model comparison and provides the transparency needed for confident deployment of ML models in industrial PdM applications.

## 5 RESULTS AND ANALYSIS

**Dataset Coverage and Variability.** Our evaluation spans **14 datasets** covering four major categories: *bearing systems* (Paderborn, HUST, CWRU, XJTU-SY, FEMTO, IMS, MFPT, Mendeley), *motor faults* (Electric Motor, Rotor Broken Bar), *gearbox assemblies* (UConn, WT Planetary), and *multi-component environments* (MAFAULDA, Microsoft Azure). This broad coverage surfaces critical variation in data fidelity, operating conditions, and signal complexity. For instance, models consistently achieve high F1 scores on bearing datasets, with HUST and FEMTO nearing perfect

classification, while performance on motor datasets remains far more inconsistent. Electric Motor, in particular, exposes weaknesses in many architectures, likely due to low signal-to-noise ratios and less structured failure patterns. These disparities underscore the need for fine-grained, domain-aware evaluation strategies. A model that excels in controlled bearing environments may not generalize to telemetry-rich, asynchronous systems like Azure. PDMBench surfaces these cross-domain generalization gaps and offers a foundation for future research in adaptive model transfer, fault-type conditioning, and sensor-specific tuning.

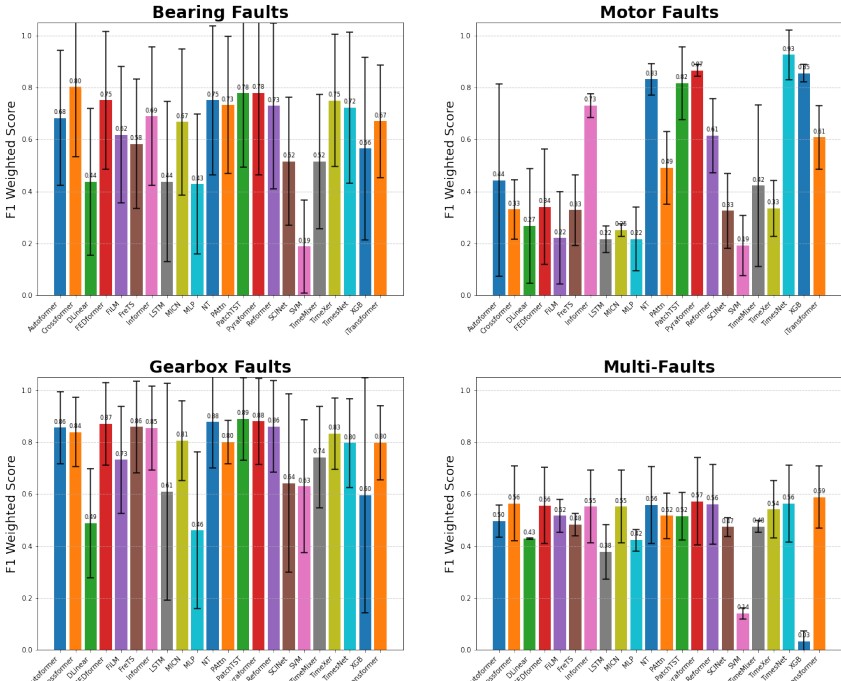

Figure 4: **Model Performance across Component Categories.** F1 scores for fault classification tasks are reported separately for bearing, motor, gearbox, and multi-component datasets. Models such as TimesNet and PatchTST demonstrate strong generalization in structured environments (e.g., bearings), while others, such as DLinear, show consistent robustness despite minimal complexity.

**Model-Level Insights: Accuracy is Not Uniform.** As shown in Figure 4, no single model dominates across all component types. *Transformer-based models* such as TimesNet and Crossformer lead on several bearing and gearbox datasets, showcasing their strength in long-range dependency modeling. Yet, simpler architectures like *DLinear* consistently maintain respectable F1 scores across diverse settings, despite having only a fraction of the parameters, highlighting the advantage of inductive bias in capturing core fault signals. Interestingly, *patch-based models* like PatchTST and PAttn exhibit strong performance under constrained or low-resource settings. Their structured input representations appear particularly effective in datasets with limited training instances, such as Rotor Broken Bar. This suggests that localized pattern aggregation may be more reliable than global attention in sparse environments, opening pathways for further exploration in hybrid localized-global transformer designs for industrial time series.

**Efficiency and Calibration: Tradeoffs and Deployment Readiness.** Figure 5 highlights two critical dimensions for deployment: uncertainty calibration and computational efficiency. Models vary dramatically in how well their predicted probabilities reflect actual confidence. For example, while Pyraformer and SCINet yield high classification accuracy on several datasets, they suffer from poor calibration (ECE values exceeding 0.4), making their output risky in safety-critical contexts. On the other hand, models like PAttn and FiLM achieve more stable confidence alignment, albeit with moderate predictive performance. Efficiency metrics show equally stark contrasts. TimesNet and Reformer incur large computational overheads, requiring over 200 seconds per epoch in some datasets, while DLinear completes inference in under 1 second. Importantly, these cost differences do not translate directly into accuracy gains. In fact, several models strike a favorable balance: *PatchTST and MICN* exhibit relatively low inference costs and strong calibration, suggesting their suitability

for resource-constrained real-time systems. These observations point to a need for *multi-objective optimization platforms* in PdM research, where predictive performance is balanced against reliability and deployability. Future work might focus on model pruning, calibration-aware training, and early-exit strategies tailored for maintenance environments with edge or cloud-based compute.

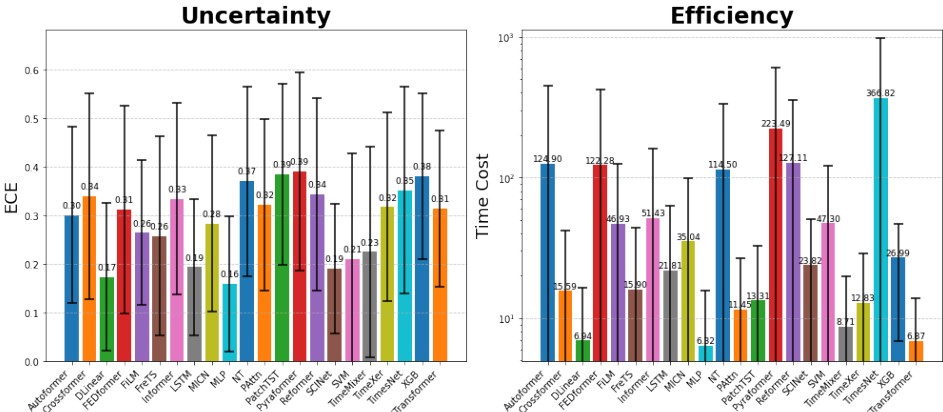

Figure 5: **Model Calibration and Efficiency.** Left: Expected Calibration Error (ECE) across models reveals large variability in predictive uncertainty alignment. Right: Inference time per epoch (log scale) illustrates computational cost disparities. Some models, like PatchTST and DLinear, strike a compelling balance between speed and reliability.

Taken together, our results highlight that **PdM model selection must be context-specific**. Models like TimesNet or Crossformer may be optimal in data-rich environments focused on bearings and gearboxes. Conversely, scenarios involving asynchronous, telemetry-driven monitoring, such as Azure or MAFAULDA, may benefit more from robust, interpretable, and efficient learners like DLinear or PatchTST. Most importantly, the wide performance variability across datasets, tasks, and metrics reinforces the value of PDMBench as a reproducible, extensible testbed. Aligning model performance with calibration, efficiency, and data regime enables practitioners to select models that meet not only accuracy goals but also operational constraints and risk tolerances. This benchmarking platform opens new directions in adaptive model deployment, curriculum learning across component types, and dynamic architecture selection in industrial AI.

## 6 CONCLUSION

We present **PDMBench**, a standardized platform that addresses key challenges in PdM research: fragmented datasets, inconsistent evaluation, and limited human-in-the-loop support. It unifies 14 real-world datasets, 22 time-series models, and a triadic evaluation framework covering accuracy, uncertainty, and efficiency. An interactive web interface further supports reproducible and transparent experimentation. Empirical results show that transformer-based models perform well in structured settings but often lack calibration and efficiency. Lightweight models offer reliable performance under noisy or low-resource conditions, while patch-based architectures provide a promising balance of robustness and deployability. No single model performs best across tasks or domains, underscoring the need for context-aware model selection, precisely the goal of PDMBench.

## REPRODUCIBILITY STATEMENT

We provide material to ensure that our work is fully reproducible. In particular, we provide detailed justification for our unified multi-modal input approach and evaluation framework in Appendix B; we include comprehensive descriptions of all 14 datasets and 22 baseline models in Appendix C, along with complete experimental results covering accuracy, calibration, and efficiency metrics in Appendix D. The preprocessing pipeline for converting heterogeneous sensor streams into stan-dardized representations is detailed in Section 4.1, with algorithmic specifications provided in the supplementary materials. User interface documentation and platform usage guidelines are outlined

in Appendix E. An anonymized version of the code used to reproduce our results can be found at `https://anonymous.4open.science/r/PDMBenchmark-C811/`. All datasets used in our experiments are publicly accessible at `https://huggingface.co/submission096`.

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

## A    APPENDIX CONTENT

- Appendix B: Justification and implications for unified multimodal input representation in PDM datasets.

- Appendix C: Detailed dataset descriptions (14 datasets across bearings, motors, gearboxes, and multi-component systems) and comprehensive baseline model specifications (22 time-series models ranging from traditional ML to state-of-the-art transformers).

- Appendix D: Complete experimental results tables showing performance of all baselines across all datasets, covering accuracy analysis (ACC, F1-macro, F1-weighted), calibration analysis (ECE, NLL, Brier), efficiency analysis (training time, parameter count), and explanation analysis (SHAP).

- Appendix E: User interface documentation and platform usage guidelines, including screenshots of the data visualization, model training, and result analysis components of the PDMBench platform.

## B    UNIFYING MULTI-MODAL INPUTS: JUSTIFICATION AND IMPLICATIONS

A common concern in multimodal predictive maintenance is the apparent mismatch between the diversity of sensor modalities in the raw data and the predominance of single-modal or unimodal model architectures in evaluation. PDMBENCH directly addresses this challenge through a principled unification strategy at the data preprocessing stage, converting heterogeneous sensor streams into a standardized segment representation, which enables consistent benchmarking across models, regardless of their input assumptions.

**Multi-Modal to Single Representation Transformation.**    Each raw dataset $D_j$ comprises time series from multiple sensor modalities $\{X_m\}_{m=1}^M$, where modalities can be vibration, current, temperature, speed, and torque signals, often sampled asynchronously and with varying fidelity. Through our harmonized preprocessing pipeline $P$, these disparate modalities are co-processed into aligned, fixed-length, normalized segments $x \in \mathbb{R}^{M \times L}$. This transformation ensures that all downstream models operate on a consistent segment format, regardless of whether they were originally designed for unimodal or multimodal inputs. This design choice does not discard modality-specific information but rather encapsulates it within a unified segment that preserves cross-modal correlations through temporal synchronization, statistical normalization, and metadata encoding. Critically, the standardized input representation supports models ranging from handcrafted-feature baselines (e.g., SVM, XGBoost) to complex sequence learners (e.g., TimesNet, Crossformer) without requiring architecture-level multimodal fusion layers, which are often brittle, opaque, and hard to calibrate across diverse datasets.

**Why Not Use Multi-Modal Architectures Directly?** PDMBENCH prioritizes reproducibility, generalizability, and fair comparison. Many multimodal deep learning models make strong assumptions about modality structure (e.g., aligned sampling, explicit modality priors), which do not hold across the 14 datasets we benchmark. Moreover, such models often require custom fusion schemes and per-dataset tuning, which undermines the goal of standardized evaluation. By converting all datasets into a shared representational form, we decouple benchmarking from modality-specific model engineering, allowing us to fairly assess model robustness across varying sensor configurations without conflating performance with fusion strategy.

**Why the Unified Representation Remains Modality-Agnostic.** The unified representation $x \in \mathbb{R}^{M \times L}$ treats all sensor streams as equal contributors to the input segment. Because the pipeline applies the same preprocessing logic (e.g., z-normalization, fixed-window segmentation) across all modalities, no sensor type is privileged in model input. Models trained on this representation learn to extract features based on the predictive value of each modality for the target task (CLF or RUL), not based on arbitrary encoding order or signal type. This approach enables flexible scalability: new modalities can be added to the representation by extending the modality dimension $M$, without changing the model or retraining fusion components. It also avoids hand-tuned feature hierarchies that might bias models toward modalities with high signal-to-noise ratios or clearer patterns (e.g., vibration over temperature). The modality-agnostic nature of the segment ensures that performance gains are attributed to true cross-modal learning and not to structural biases in the model or pipeline.

## C DATASETS AND BASELINES

### C.1 DATASETS

#### C.1.1 BEARINGS DATASETS

Bearings are among the most failure-prone components in rotating machinery, and thus have been the focus of numerous predictive maintenance studies.

**Paderborn** Lessmeier et al. (2016): This dataset features both artificially induced and naturally degraded bearing faults. It includes multimodal sensor signals such as vibration, motor current, speed, and torque, offering a rich testbed for sensor fusion and fault transferability analysis.

**HUST** Thuan & Hong (2023): Collected under varying speeds and loads, this dataset captures vibration signals from five bearing types and six fault categories, including inner/outer race and ball defects. It is ideal for evaluating generalization under diverse operating conditions.

**IMS**: Created by the Center for Intelligent Maintenance Systems, it documents full bearing life cycles under constant speed/load settings. The dataset supports RUL prediction and anomaly detection with high temporal resolution.

**CWRU** Smith & Randall (2015): A widely used standard, this dataset includes vibration signals from bearings with precisely machined faults of varying sizes on different components (ball, inner, outer race), collected at multiple loads and speeds.

**XJTU-SY** Lei et al. (2019): Capturing run-to-failure data under three load-speed settings, this dataset includes five fault types. The consistent degradation trajectories and large size make it suitable for both fault classification and RUL estimation.

**MFPT** for Machinery Failure Prevention Technology: Provided by the Society for Machinery Failure Prevention Technology, this dataset features vibration measurements of bearings with outer/inner race defects under varying load conditions and extremely high sampling rates (97.656 kHz), enabling fine-grained frequency-domain analysis.

**FEMTO** Nectoux et al. (2012b): Designed for prognostics, this dataset comprises horizontal and vertical vibration measurements of bearings run to failure under three operational scenarios, with partial trajectory truncation to simulate real-world prediction settings.

**Mendeley Bearing** Kechik et al. (2020): This dataset addresses the challenge of fault detection under non-stationary conditions, featuring vibration signals from bearings tested across a wide speed range (600–1800 RPM).

**Electric Motor Vibrations** Sehri & Dumond (2023): Includes both vibration and acoustic signals collected under constant and variable speeds. It enables multimodal analysis of motor anomalies, including eccentricity and imbalance.

**Rotor Broken Bar** Treml et al. (2020): Focuses on broken rotor bar faults in induction motors. It provides motor current data under varying fault severities (1–4 broken bars), facilitating research in motor current signature analysis (MCSA).

### C.1.2 GEARBOX FAULT DATASETS

Gearboxes involve complex kinematics and multiple interacting components, requiring specialized datasets.

**Planetary Gearbox** Liu et al. (2023a): Vibration data collected from a planetary gearbox with diverse fault types (e.g., sun gear crack, planet spalling). The non-trivial structure of planetary gear systems makes this dataset valuable for studying fault localization.

**Gearbox UoC** Cao et al. (2018): Developed at the University of Connecticut, this dataset includes vibration and temperature signals from gearboxes exhibiting various gear and bearing faults, processed with FFT to yield frequency-domain features.

### C.1.3 MULTI-COMPONENT SYSTEM DATASETS

These datasets integrate multiple fault types and sensor modalities, reflecting the systemic complexity of real industrial environments.

**MAFAULDA** Laboratory of Rotating Machinery Diagnostics, UFRJ (2000): A large-scale dataset from a machinery fault simulator featuring 13 fault conditions including imbalance, misalignment, and shaft cracks. It includes vibration, acoustic, and speed signals, enabling multi-sensor and multi-class analysis.

**Microsoft Azure** Microsoft: A synthetic yet industrial-grade dataset with telemetry data (e.g., voltage, pressure, vibration) and system logs, capturing component-level and system-level failures. It supports holistic end-to-end PdM model development.

Table 1: Comparison of predictive maintenance datasets. The datasets are categorized by components (bearings, motors, gearboxes, and multicomponent fault systems) and designed for two main tasks: fault diagnosis (identifying specific fault types), and RUL prediction (estimating time to failure).

| Dataset | Design Target | Property | Sampling Rate |
|---|---|---|---|
| CWRU | Fault diagnosis | Vibration | 12 kHz / 48 kHz |
| XJTU-SY | RUL prediction & Fault diagnosis | Vibration | 25.6 kHz |
| IMS | RUL prediction | Vibration | 20.48 kHz |
| Paderborn | Fault diagnosis | Multiple sensors | 8 kHz / 16 kHz |
| FEMTO | RUL prediction | Multiple sensors | 25.6 kHz |
| MFPT | Fault diagnosis | Vibration | 97.6 kHz |
| HUST | Fault diagnosis | Vibration | 51.2 kHz |
| Electric Motor | Fault diagnosis | Multiple sensors | 42 kHz |
| Rotor Broken Bar | Fault diagnosis | Current | 50 kHz |
| WT Planetary Gearbox | Fault diagnosis | Vibration | 48 kHz |
| CQU Gearbox | Fault diagnosis | Multiple sensors | 20 kHz |
| UConn Gearbox | Fault diagnosis | Vibration | 20 kHz |
| MAFAULDA | Fault diagnosis | Multiple sensors | 51.2 kHz |
| Mendeley Bearing | Fault diagnosis | Vibration | 9.6 kHz |
| Microsoft Azure | RUL prediction | Telemetry, Logs | Hourly |

### C.2 BASELINES

To facilitate comprehensive and reproducible evaluation, PDMBench includes 22 time-series models drawn from a broad spectrum of modeling paradigms. These range from traditional baselines to

state-of-the-art transformer variants, each selected for its relevance to time-series classification and forecasting in industrial contexts.

Traditional baselines such as MLP Bengio et al. (2003), XGBoost Chen & Guestrin (2016), SVM Hearst et al. (1998), LSTM Graves & Graves (2012), and DLinear Zeng et al. (2023) offer global feedforward and decomposable linear processing capabilities, respectively. Transformer-based architectures include Autoformer Wu et al. (2021), which employs auto-correlation mechanisms for long-term forecasting, and Crossformer Zhang et al. (2023), designed to capture cross-dimensional dependencies. FEDformer Zhou et al. (2022a) and FiLM Zhou et al. (2022b) use frequency-domain representations, while Informer Zhou et al. (2021) introduces sparse attention for efficiency. Other models include the iTransformer Liu et al. (2023b), tailored for multivariate sequences via inverted attention, and the Reformer Kitaev et al. (2020), which enables efficient learning on extremely long sequences using locality-sensitive hashing. Hybrid and hierarchical models include TimesNet Wu et al. (2023), combining CNNs and transformers for 2D-periodicity, and SCINet Liu et al. (2021b), which applies sample interaction in a hierarchical setting. Recent architectures with strong multiscale properties include TimeXer Wang et al. (2024b), TimeMixer Zhang et al. (2024b), and MICN Wang et al. (2023), while lightweight yet expressive learners such as PatchTST Nie et al. (2023), PAttn Tan et al. (2024), and FreTS Yi et al. (2023) utilize patch-based and spectral-domain designs.

Table 2: An overview of comparison methods for PdM applications. TS for time series.

| Model | Primary Target | Temporal Handling | Data Properties | Backbone |
|---|---|---|---|---|
| Autoformer | Long-term series | Auto-correlation | Decomposition | Transformer |
| Crossformer | Multivariate TS | Cross-dimension | Inter-variable | Transformer |
| DLinear | Decomposable TS | Separate processing | Linear | MLP |
| FEDformer | Long-term TS | Frequency enhanced | Decomposable | Transformer |
| FiLM | Long-term TS | Frequency improved | Memory model | Legendre |
| FreTS | Cyclic TS | Frequency-domain | Spectral | MLP |
| Informer | Long sequence | ProbSparse | Efficient attention | Transformer |
| iTransformer | Multivariate TS | Inverted attention | Variable-focused | Transformer |
| MICN | Long-term TS | Multi-scale | Local+global context | GNN |
| MLP | General | N/A | Global | Feedforward |
| N-Transformer | Non-stationary TS | Adaptive | Series decomposition | Transformer |
| PAttn | TS | Patch-based | Attention balance | Transformer |
| PatchTST | Long-term sequences | Patched | Channel independence | Transformer |
| Pyraformer | Long-range TS | Pyramidal | Multi-resolution | Attention |
| Reformer | Extremely long sequences | LSH attention | Memory-efficient | Transformer |
| SCINet | General TS | Hierarchical | Sample interaction | CNN |
| TimeMixer | Multivariate TS | Decomposable | Multiscale | MLP |
| TimesNet | General TS | 2D-variation | Multi-periodicity | CNN+Transformer |
| TimeXer | TS with exog. variables | Multi-resolution | Global attention | Transformer |

## D  ADDITIONAL EXPERIMENT RESULTS

### D.1  EXPERIMENTAL SETUP

To ensure a fair and consistent comparison across all models, we developed a unified evaluation framework encompassing standardized data preprocessing procedures and consistent evaluation metrics. All models are trained and evaluated on the same training(60%), validation(10%), and test datasets(30%). Each time series is segmented into smaller time windows to facilitate analysis and accommodate device memory constraints. In addition to performing a conventional random split, we first group time segments by bearing number and assign all segments from the same bearing to a single dataset (training, validation, or test). This strategy is crucial for preventing data leakage, as randomly splitting individual time segments could result in similar patterns, originating from the same bearing across different datasets.

### D.2  EVALUATION

In predictive maintenance (PdM), various tasks are commonly addressed, including classification (CLF) and remaining useful life estimation (RUL), the latter being a regression problem. For the

Table 3: The statistics of comparison methods for PdM applications. NT for Nonstationary Transformer. TS for time series. P for Prediction

| Model | Task | Primary Target | Temporal Handling | Data Properties | Backbone |
|---|---|---|---|---|---|
| MLP | P | General | N/A | Global | Feedforward |
| TimeXer | P | TS with exog. variables | Multi-resolution | Global attention | Transformer |
| TimeMixer | P | Multivariate TS | Decomposable | Multiscale | MLP |
| TimesNet | P | General TS | 2D-variation | Multi-periodicity | CNN+Transformer |
| DLinear | P | Decomposable TS | Separate processing | Linear | MLP |
| NT | P | Non-stationary TS | Adaptive | Series decomposition | Transformer |
| FEDformer | P | Long-term TS | Frequency enhanced | Decomposable | Transformer |
| Pyraformer | P | Long-range TS | Pyramidal | Multi-resolution | Attention |
| Autoformer | P | Long-term TS | Auto-correlation | Decomposition | Transformer |
| Informer | P | Long sequence | ProbSparse | Efficient attention | Transformer |
| Reformer | P | Extremely long sequences | LSH attention | Memory-efficient | Transformer |
| MICN | P | Long-term TS | Multi-scale | Local+global context | GNN |
| Crossformer | P | Multivariate TS | Cross-dimension | Inter-variable | Transformer |
| FiLM | P | Long-term TS | Frequency improved | Memory model | Legendre |
| SCINet | P | General TS | Hierarchical | Sample interaction | CNN |
| PatchTST | P | Long-term sequences | Patched | Channel independence | Transformer |
| iTransformer | P | Multivariate TS | Inverted attention | Variable-focused | Transformer |
| PAttn | P | TS | Patch-based | Attention balance | Transformer |
| FreTS | P | Cyclic TS | Frequency-domain | Spectral | MLP |

CLF task, we adopt widely used evaluation metrics such as accuracy and F1 score to assess model performance. In contrast, the RUL task requires running bearings to failure to observe their complete lifespans. These time spans must then be normalized into percentages, which often introduces significant noise and variability into the data.

To address this challenge and ensure consistency in our benchmarks, we reformulate the RUL task as a classification problem. Specifically, we divide the full RUL timeline into 10 discrete intervals and train models to predict the interval (i.e., stage of degradation) a bearing belongs to. This transformation enables the use of standard classification metrics—such as accuracy and F1 score—for evaluating RUL predictions.

In addition to these standard performance metrics, we also incorporate calibration metrics, including Expected Calibration Error (ECE), Negative Log-Likelihood (NLL), and Brier score. These metrics are critical for real-world applications, where end users require not only accurate predictions but also reliable estimates of the model's confidence.

### D.3 QUANTITATIVE ANALYSIS

In this section, we present the experimental results across all 14 datasets using 20 different models. The results are summarized in Table 4 through Table 17. Each table reports three categories of evaluation metrics to comprehensively assess model performance on each dataset: (1) Accuracy metrics – including Accuracy and F1 score; (2) Calibration metrics – including Expected Calibration Error (ECE), Negative Log-Likelihood (NLL), and Brier score; (3) Efficiency metrics – including training time and parameter size.

### D.3.1 ACCURACY ANALYSIS

In this section, we focus on the analysis of accuracy metrics. Our analysis is conducted from two perspectives: (1) Dataset-wise analysis, where we examine the performance trends across different datasets and discuss the specific challenges each dataset poses for predictive maintenance (PdM); (2) Model-wise analysis, where we evaluate the performance of each model across all datasets to identify overall strengths, weaknesses, and consistency.

**Analysis from Datasets Perspective.** From the experimental results, we observe that the difficulty level varies significantly across datasets. Approximately half of the datasets—such as Paderborn, FEMTO, Azure, Rotor Broken Motor, Planetary, and MAFAULDA—demonstrate relatively higher difficulty, as the best-performing models achieve accuracies below 0.80. This indicates considerable

room for improvement and highlights these datasets as valuable benchmarks for advancing model performance. In contrast, datasets like Mendeley, UoC, MFPT, Electric Motor, and IMS exhibit high classification performance, with accuracies exceeding 0.95. While these results reflect the maturity of model performance on these datasets, they also suggest diminishing returns for further research, as performance gains may be marginal. The remaining datasets—XJTU, CWRU, and HUST—achieve around 0.90 accuracy, indicating moderate difficulty and continued relevance for further exploration.

These observations lead to two key insights: (i) There is a strong need to develop and release more challenging and diverse PdM datasets to stimulate progress in the field. Harder datasets can better differentiate model capabilities and promote deeper understanding of PdM problems. (ii) Despite the time and effort required to develop new datasets, our findings show that at least 9 out of 14 existing datasets still present meaningful challenges, making them valuable resources for ongoing research and performance enhancement.

**Analysis from Models Perspective.** Our results show that, overall, transformer-based state-of-the-art time series models outperform traditional machine learning approaches such as SVM and XGBoost. Similarly, vanilla deep learning models—such as MLP and LSTM—tend to underperform compared to transformer-based architectures across most datasets.

In relatively easy datasets (e.g., Mendeley, UoC, MFPT, Electric Motor, and IMS), traditional machine learning models achieve accuracy levels comparable to those of transformer-based models. However, despite their smaller number of trainable parameters, these models generally require longer training times to reach convergence, resulting in comparable computational costs to some transformer-based models.

Importantly, we observe that a larger model size does not necessarily translate to better performance. For instance, on the Paderborn dataset (Table 4), TimesNet, with nearly 30 times more parameters than Crossformer, fails to achieve superior performance. This highlights the importance of model architecture and data suitability over mere parameter count.

Additionally, we also observe that different segmentations will end in various performances. Here, we provide a case study using the XJTU dataset as shown in Table 18. In general, we find that longer segmentation ensures better accuracy performance. This is reasonable as longer segmentation brings more information for the model to capture. However, longer segmentation ends with more computation cost, which encourages a balance between accuracy and computation cost.

### D.3.2 CALIBRATION ANALYSIS

In this section, we analyze the calibration performance of each model across the datasets. Our analysis is conducted from two complementary perspectives: (1) analysis through datasets, where we compare the calibration results for each dataset to understand how data characteristics influence model confidence. (2) analysis through models, where we assess the overall calibration quality of each model and examine how different datasets affect their calibration behavior.

**Analysis from Datasets Perspective.** While model calibration is typically analyzed from the perspective of model architectures, our study highlights the importance of evaluating calibration performance from the dataset perspective, particularly in the predictive maintenance (PdM) domain. For example, although models achieve high accuracy (often exceeding 0.95) on datasets such as IMS, they exhibit poor calibration, indicating a mismatch between predicted probabilities and actual outcomes.

This observation has significant implications for real-world deployment, where not only accuracy but also confidence in predictions is critical. Poor calibration can lead to overconfident or misleading predictions, which poses risks in high-stakes industrial applications. Our findings suggest that, even for datasets where accuracy appears saturated, improving model calibration remains a valuable and necessary direction for future research. This shift in focus can enhance the reliability and trustworthiness of PdM systems in practical scenarios.

**Analysis from Models Perspective.** Our results reveal that model performance on calibration metrics often differs from their performance on accuracy metrics. Some models that excel in accuracy underperform in calibration. For instance, on the UoC dataset, the TimeMixer model significantly outperforms iTransformer in terms of accuracy but yields worse results in terms of

Expected Calibration Error (ECE). This highlights that high accuracy does not necessarily imply well-calibrated predictions.

Additionally, we observe that traditional machine learning models, which generally lag behind in accuracy, also perform poorly on calibration metrics such as ECE, NLL, and Brier score. These consistent shortcomings reinforce the argument that future research efforts should prioritize transformer-based architectures, as they offer greater potential for both improved accuracy and better-calibrated predictions.

### D.3.3 EFFICIENCY ANALYSIS

In this section, we analyze the efficiency performance of each model across all datasets, focusing on both time cost and parameter size.

Dataset-wise analysis. Our experimental results show that the same model can exhibit different time costs depending on the dataset. Interestingly, even datasets that are relatively easy for the selected models in terms of accuracy may still require longer training times to reach convergence. This suggests that dataset complexity is not the sole factor influencing time efficiency—data structure and feature characteristics also play a role.

Model-wise analysis. State-of-the-art transformer-based models typically have more parameters than traditional machine learning models. However, in terms of training time, traditional models may sometimes take longer to converge. This is likely due to their limited capacity for capturing complex patterns, requiring more iterations to achieve optimal performance. Additionally, we observe that the parameter size for each model remains relatively consistent across different datasets. This is expected, as parameter size is primarily determined by the model architecture, with only minor variation due to dataset-specific input feature dimensions.

Table 4: Paderborn dataset. NT for Nonstationary_Transformer. F1-mac stands for F1-macro score, and F1-w stands for F1-weighted score. ECE stands for expected calibration error. NLL stands for negative log likelihood. Parameters(k) indicate the number of parameters in thousands. The red font represents the best, the green font represents the second best, and the blue font represents the third best.

| model | ACC | F1-mac | F1-w | ECE | NLL | Brier | time cost | parameters(k) |
|---|---|---|---|---|---|---|---|---|
| SVM | 0.1666 | 0.0991 | 0.0981 | 0.1310 | 1.4361 | 0.1937 | 34.3 | 1.0 |
| XGB | 0.2177 | 0.1904 | 0.2426 | 0.0774 | 1.4006 | 0.1894 | 40.0 | 236.6 |
| MLP | 0.4243 | 0.1748 | 0.2989 | 0.1566 | 1.3647 | 0.1848 | 0.6 | 477.1 |
| LSTM | 0.4289 | 0.2011 | 0.2585 | 0.1553 | 1.3233 | 0.1797 | 6.5 | 1924.3 |
| TimeXer | 0.3695 | 0.2639 | 0.3987 | 0.0438 | 1.3410 | 0.1817 | 4.0 | 695.2 |
| TimeMixer | 0.3710 | 0.2101 | 0.2844 | 0.0574 | 1.3437 | 0.1824 | 1.5 | 506.6 |
| iTransformer | 0.4155 | 0.2957 | 0.4329 | 0.0787 | 1.3058 | 0.1772 | 0.9 | 475.1 |
| PatchTST | 0.4304 | 0.2165 | 0.3268 | 0.1256 | 1.3202 | 0.1791 | 1.4 | 419.2 |
| TimesNet | 0.4162 | 0.2645 | 0.3850 | 0.0249 | 1.2924 | 0.1746 | 122.7 | 56545.2 |
| DLinear | 0.4115 | 0.2386 | 0.3916 | 0.1447 | 1.3584 | 0.1840 | 0.9 | 723.6 |
| NT | 0.3435 | 0.2242 | 0.3201 | 0.0535 | 1.3569 | 0.1844 | 37.1 | 820.1 |
| FEDformer | 0.4212 | 0.2950 | 0.4193 | 0.0171 | 1.2887 | 0.1747 | 18.8 | 1301.0 |
| Pyraformer | 0.3800 | 0.1938 | 0.2950 | 0.0651 | 1.3294 | 0.1808 | 53.7 | 1341.3 |
| Autoformer | 0.3860 | 0.2648 | 0.3850 | 0.0651 | 1.3268 | 0.1804 | 30.8 | 703.1 |
| Informer | 0.4023 | 0.3468 | 0.3643 | 0.0956 | 1.3058 | 0.1775 | 7.4 | 908.9 |
| Reformer | 0.3568 | 0.1884 | 0.2812 | 0.0736 | 1.3580 | 0.1843 | 39.2 | 655.5 |
| MICN | 0.3820 | 0.2198 | 0.3211 | 0.0631 | 1.3391 | 0.1816 | 4.3 | 2560.4 |
| Crossformer | 0.4251 | 0.2804 | 0.4243 | 0.0583 | 1.2862 | 0.1743 | 1.0 | 2814.5 |
| FiLM | 0.4484 | 0.2750 | 0.3536 | 0.1945 | 1.3793 | 0.1866 | 4.3 | 12585.3 |
| SCINet | 0.3689 | 0.2462 | 0.3809 | 0.0548 | 1.3375 | 0.1814 | 7.2 | 363.2 |
| PAttn | 0.4208 | 0.2325 | 0.3908 | 0.1159 | 1.3148 | 0.1785 | 3.5 | 165.4 |
| FreTS | 0.4203 | 0.2463 | 0.3910 | 0.1282 | 1.3229 | 0.1795 | 1.5 | 20034.4 |

Besides current baselines, we also add experimental results using MLflow in Paderborn, HUST, CRWU, and XJTU datasets to show how the machine learning framework performs on PdM benchmarks. The experimental results are shown in Table 19. We observe that MLflow can improve the machine learning models' performance on the Paderborn and XJTU datasets, while degrading the

Table 5: HUST dataset. NT for Nonstationary_Transformer. F1-mac stands for F1-macro score, and F1-w stands for F1-weighted score. ECE stands for expected calibration error. NLL stands for negative log likelihood. Parameters(k) indicate the number of parameters in thousands. The red font represents the best, the green font represents the second best, and the blue font represents the third best.

| model | ACC | F1-mac | F1-w | ECE | NLL | Brier | time cost | parameters(k) |
|---|---|---|---|---|---|---|---|---|
| SVM | 0.1402 | 0.1272 | 0.1278 | 0.0202 | 1.9437 | 0.1224 | 87.0 | 2.5 |
| XGB | 0.5469 | 0.5252 | 0.5390 | 0.3468 | 1.7222 | 0.1116 | 78.6 | 486.9 |
| MLP | 0.2632 | 0.2085 | 0.2205 | 0.0967 | 1.8978 | 0.1203 | 1.1 | 728.1 |
| LSTM | 0.3785 | 0.2558 | 0.2710 | 0.1980 | 1.8165 | 0.1160 | 10.7 | 1924.4 |
| TimeXer | 0.9231 | 0.9170 | 0.9223 | 0.6249 | 1.2566 | 0.0845 | 1.2 | 1070.9 |
| TimeMixer | 0.4585 | 0.4286 | 0.4441 | 0.2373 | 1.7271 | 0.1116 | 4.6 | 2493.2 |
| iTransformer | 0.5266 | 0.5084 | 0.5218 | 0.2996 | 1.6667 | 0.1082 | 1.4 | 475.5 |
| PatchTST | 0.9694 | 0.9667 | 0.9693 | 0.6598 | 1.1967 | 0.0808 | 4.5 | 543.1 |
| TimesNet | 0.5071 | 0.4383 | 0.4619 | 0.2524 | 1.6431 | 0.1064 | 331.9 | 58531.7 |
| DLinear | 0.1806 | 0.1515 | 0.1614 | 0.0317 | 1.9439 | 0.1224 | 0.8 | 725.4 |
| NT | 0.8844 | 0.8783 | 0.8833 | 0.5771 | 1.2799 | 0.0855 | 101.5 | 3069.3 |
| FEDformer | 0.8185 | 0.8058 | 0.8170 | 0.5180 | 1.3460 | 0.0894 | 87.3 | 1531.4 |
| Pyraformer | 0.9819 | 0.9807 | 0.9819 | 0.6712 | 1.1834 | 0.0800 | 297.7 | 7301.0 |
| Autoformer | 0.6671 | 0.6587 | 0.6715 | 0.3930 | 1.5121 | 0.0994 | 12.9 | 933.5 |
| Informer | 0.8407 | 0.8360 | 0.8401 | 0.5407 | 1.3267 | 0.0884 | 11.6 | 1139.3 |
| Reformer | 0.8828 | 0.8769 | 0.8817 | 0.5772 | 1.2818 | 0.0857 | 165.2 | 2642.1 |
| MICN | 0.4465 | 0.4081 | 0.4253 | 0.2395 | 1.7551 | 0.1133 | 14.2 | 7354.9 |
| Crossformer | 0.9422 | 0.9389 | 0.9422 | 0.6433 | 1.2409 | 0.0837 | 3.0 | 2819.5 |
| FiLM | 0.4929 | 0.4825 | 0.4931 | 0.2893 | 1.7184 | 0.1110 | 9.8 | 12587.1 |
| SCINet | 0.4643 | 0.4280 | 0.4461 | 0.2607 | 1.7358 | 0.1119 | 13.1 | 6572.2 |
| PAttn | 0.8766 | 0.8685 | 0.8768 | 0.5745 | 1.2903 | 0.0863 | 10.0 | 364.4 |
| FreTS | 0.6517 | 0.6280 | 0.6438 | 0.4388 | 1.6391 | 0.1072 | 1.4 | 20264.8 |

Table 6: CWRU dataset. NT for Nonstationary_Transformer. F1-mac stands for F1-macro score, and F1-w stands for F1-weighted score. ECE stands for expected calibration error. NLL stands for negative log likelihood. Parameters(k) indicate the number of parameters in thousands. The red font represents the best, the green font represents the second best, and the blue font represents the third best.

| model | ACC | F1-mac | F1-w | ECE | NLL | Brier | time cost | parameters(k) |
|---|---|---|---|---|---|---|---|---|
| SVM | 0.4690 | 0.4203 | 0.3790 | 0.2105 | 1.1724 | 0.1588 | 1.8 | 0.6 |
| XGB | 0.9832 | 0.9801 | 0.9832 | 0.5113 | 0.7623 | 0.0947 | 1.7 | 65.8 |
| MLP | 0.4961 | 0.4002 | 0.4481 | 0.0911 | 1.0289 | 0.2073 | 6.1 | 402.5 |
| LSTM | 0.4901 | 0.4021 | 0.4288 | 0.1986 | 1.2688 | 0.1728 | 1.4 | 403.8 |
| TimeXer | 0.9071 | 0.8957 | 0.9074 | 0.3489 | 0.6457 | 0.1133 | 15.5 | 602.1 |
| TimeMixer | 0.4937 | 0.4634 | 0.4908 | 0.0320 | 1.0081 | 0.2018 | 8.7 | 208.1 |
| iTransformer | 0.7418 | 0.7272 | 0.7421 | 0.1976 | 0.8047 | 0.1518 | 13.6 | 400.5 |
| PatchTST | 0.9514 | 0.9431 | 0.9516 | 0.3857 | 0.6028 | 0.1026 | 12.6 | 400.3 |
| TimesNet | 0.9835 | 0.9839 | 0.9835 | 0.4116 | 0.5697 | 0.0944 | 27.2 | 56246.7 |
| DLinear | 0.4236 | 0.3049 | 0.3846 | 0.0271 | 1.0705 | 0.2159 | 8.3 | 0.7 |
| NT | 0.9154 | 0.9255 | 0.9154 | 0.3652 | 0.6423 | 0.1126 | 6.9 | 443.5 |
| FEDformer | 0.9385 | 0.9410 | 0.9384 | 0.3773 | 0.6162 | 0.1060 | 6.3 | 674.8 |
| Pyraformer | 0.9781 | 0.9800 | 0.9781 | 0.4026 | 0.5737 | 0.0953 | 9.7 | 445.8 |
| Autoformer | 0.8656 | 0.8394 | 0.8609 | 0.3187 | 0.6881 | 0.1236 | 7.5 | 404.6 |
| Informer | 0.7480 | 0.7728 | 0.7468 | 0.2247 | 0.7908 | 0.1497 | 2.1 | 610.4 |
| Reformer | 0.9212 | 0.9231 | 0.9212 | 0.3613 | 0.6334 | 0.1102 | 24.8 | 357.0 |
| MICN | 0.8676 | 0.8787 | 0.8672 | 0.3501 | 0.7167 | 0.1314 | 4.3 | 1151.5 |
| Crossformer | 0.9863 | 0.9863 | 0.9863 | 0.4172 | 0.5723 | 0.0951 | 9.9 | 2791.5 |
| FiLM | 0.7629 | 0.7469 | 0.7615 | 0.2720 | 0.8233 | 0.1570 | 21.8 | 3145.8 |
| SCINet | 0.5404 | 0.4270 | 0.4985 | 0.1321 | 1.0209 | 0.2047 | 19.6 | 1.1 |
| PAttn | 0.9083 | 0.8927 | 0.9077 | 0.3564 | 0.6492 | 0.1143 | 13.3 | 135.2 |
| FreTS | 0.5878 | 0.6298 | 0.5925 | 0.1117 | 0.9342 | 0.1853 | 7.1 | 632.2 |

Table 7: XJTU dataset. NT for Nonstationary_Transformer. F1-mac stands for F1-macro score, and F1-w stands for F1-weighted score. ECE stands for expected calibration error. NLL stands for negative log likelihood. Parameters(k) indicate the number of parameters in thousands. The red font represents the best, the green font represents the second best, and the blue font represents the third best.

| model | ACC | F1-mac | F1-w | ECE | NLL | Brier | time cost | parameters(k) |
|---|---|---|---|---|---|---|---|---|
| SVM | 0.0855 | 0.0590 | 0.0821 | 0.1647 | 1.6186 | 0.1610 | 20.1 | 1.2 |
| XGB | 0.0060 | 0.1914 | 0.0061 | 0.3955 | 1.8958 | 0.1899 | 4.1 | 177.7 |
| MLP | 0.5659 | 0.3365 | 0.5356 | 0.3304 | 1.4882 | 0.1501 | 4.8 | 403.3 |
| LSTM | 0.7343 | 0.4519 | 0.7068 | 0.4134 | 1.2557 | 0.1264 | 3.8 | 405.7 |
| TimeXer | 0.7799 | 0.7088 | 0.7749 | 0.4361 | 1.1707 | 0.1175 | 25.1 | 603.1 |
| TimeMixer | 0.7684 | 0.7044 | 0.7660 | 0.4481 | 1.2280 | 0.1236 | 4.1 | 214.2 |
| iTransformer | 0.8987 | 0.8244 | 0.8946 | 0.5224 | 1.0295 | 0.1022 | 16.3 | 401.4 |
| PatchTST | 0.8802 | 0.7783 | 0.8730 | 0.5118 | 1.0617 | 0.1057 | 25.3 | 400.4 |
| TimesNet | 0.8775 | 0.8064 | 0.8740 | 0.5138 | 1.0698 | 0.1066 | 79.0 | 56252.7 |
| DLinear | 0.6904 | 0.4250 | 0.6668 | 0.3931 | 1.3221 | 0.1335 | 4.9 | 1.2 |
| NT | 0.9103 | 0.8549 | 0.9073 | 0.5283 | 1.0110 | 0.1002 | 6.1 | 450.3 |
| FEDformer | 0.8769 | 0.7989 | 0.8737 | 0.5061 | 1.0540 | 0.1048 | 36.2 | 701.3 |
| Pyraformer | 0.9079 | 0.8497 | 0.9053 | 0.5306 | 1.0206 | 0.1013 | 29.9 | 463.8 |
| Autoformer | 0.7951 | 0.7207 | 0.7946 | 0.4492 | 1.1729 | 0.1177 | 22.5 | 410.6 |
| Informer | 0.8934 | 0.8137 | 0.8894 | 0.5191 | 1.0376 | 0.1031 | 29.5 | 616.5 |
| Reformer | 0.8889 | 0.8121 | 0.8860 | 0.5173 | 1.0462 | 0.1040 | 52.1 | 363.0 |
| MICN | 0.8874 | 0.8172 | 0.8845 | 0.5183 | 1.0518 | 0.1047 | 15.7 | 1151.8 |
| Crossformer | 0.9098 | 0.8475 | 0.9063 | 0.5305 | 1.0183 | 0.1010 | 7.8 | 2791.6 |
| FiLM | 0.7868 | 0.5482 | 0.7765 | 0.4466 | 1.1750 | 0.1180 | 45.9 | 4325.5 |
| SCINet | 0.6852 | 0.5022 | 0.6676 | 0.3713 | 1.2814 | 0.1292 | 45.5 | 1.4 |
| PAttn | 0.7682 | 0.6795 | 0.7614 | 0.4289 | 1.1852 | 0.1191 | 18.0 | 136.1 |
| FreTS | 0.7415 | 0.6068 | 0.7207 | 0.4313 | 1.2679 | 0.1278 | 17.1 | 834.8 |

Table 8: FEMTO dataset. NT for Nonstationary_Transformer. F1-mac stands for F1-macro score, and F1-w stands for F1-weighted score. ECE stands for expected calibration error. NLL stands for negative log likelihood. Parameters(k) indicate the number of parameters in thousands. The red font represents the best, the green font represents the second best, and the blue font represents the third best.

| model | ACC | F1-mac | F1-w | ECE | NLL | Brier | time cost | parameters(k) |
|---|---|---|---|---|---|---|---|---|
| SVM | 0.1214 | 0.0700 | 0.0735 | 0.0249 | 2.3030 | 0.0900 | 5.0 | 3.1 |
| XGB | 0.5907 | 0.5885 | 0.5914 | 0.4114 | 1.9101 | 0.0796 | 18.8 | 282.8 |
| MLP | 0.1584 | 0.1085 | 0.1090 | 0.0554 | 2.2956 | 0.0899 | 6.0 | 406.4 |
| LSTM | 0.1641 | 0.0778 | 0.0788 | 0.0585 | 2.2929 | 0.0898 | 1.9 | 415.4 |
| TimeXer | 0.3788 | 0.3720 | 0.3746 | 0.2493 | 2.1417 | 0.0863 | 6.5 | 608.7 |
| TimeMixer | 0.2419 | 0.1913 | 0.1944 | 0.1289 | 2.2527 | 0.0889 | 7.0 | 258.3 |
| iTransformer | 0.4114 | 0.4045 | 0.4072 | 0.2804 | 2.1300 | 0.0860 | 6.4 | 405.0 |
| PatchTST | 0.4109 | 0.4027 | 0.4055 | 0.2796 | 2.1285 | 0.0860 | 6.5 | 403.6 |
| TimesNet | 0.4044 | 0.3948 | 0.3983 | 0.2685 | 2.1129 | 0.0855 | 14.1 | 56296.8 |
| DLinear | 0.1774 | 0.1281 | 0.1302 | 0.0730 | 2.2920 | 0.0898 | 5.6 | 4.8 |
| NT | 0.3624 | 0.3398 | 0.3423 | 0.2308 | 2.1508 | 0.0865 | 5.1 | 497.6 |
| FEDformer | 0.3363 | 0.3194 | 0.3222 | 0.2114 | 2.1746 | 0.0871 | 9.0 | 843.8 |
| Pyraformer | 0.3554 | 0.3392 | 0.3412 | 0.2239 | 2.1483 | 0.0864 | 5.6 | 596.3 |
| Autoformer | 0.2816 | 0.2623 | 0.2656 | 0.1645 | 2.2278 | 0.0884 | 5.6 | 454.8 |
| Informer | 0.2903 | 0.2660 | 0.2696 | 0.1693 | 2.1993 | 0.0877 | 1.9 | 660.6 |
| Reformer | 0.2596 | 0.2436 | 0.2460 | 0.1416 | 2.2172 | 0.0881 | 7.5 | 407.2 |
| MICN | 0.3725 | 0.3621 | 0.3646 | 0.2437 | 2.1485 | 0.0864 | 4.1 | 1203.4 |
| Crossformer | 0.3986 | 0.3928 | 0.3952 | 0.2684 | 2.1379 | 0.0862 | 6.1 | 2792.9 |
| FiLM | 0.2517 | 0.2278 | 0.2300 | 0.1375 | 2.2381 | 0.0885 | 8.2 | 9044.4 |
| SCINet | 0.2038 | 0.1599 | 0.1634 | 0.0951 | 2.2716 | 0.0893 | 8.6 | 3.3 |
| PAttn | 0.3363 | 0.3238 | 0.3252 | 0.2118 | 2.1770 | 0.0872 | 5.7 | 140.6 |
| FreTS | 0.2144 | 0.1749 | 0.1763 | 0.1042 | 2.2632 | 0.0891 | 7.5 | 1632.7 |

Table 9: IMS dataset. NT for Nonstationary_Transformer. F1-mac stands for F1-macro score, and F1-w stands for F1-weighted score. ECE stands for expected calibration error. NLL stands for negative log likelihood. Parameters(k) indicate the number of parameters in thousands. The red font represents the best, the green font represents the second best, and the blue font represents the third best.

| model | ACC | F1-mac | F1-w | ECE | NLL | Brier | time cost | parameters(k) |
|---|---|---|---|---|---|---|---|---|
| SVM | 0.3580 | 0.1544 | 0.5068 | 0.4961 | 0.8142 | 0.1028 | 249.2 | 0.9 |
| XGB | 0.9571 | 0.2771 | 0.9373 | 0.4935 | 0.8035 | 0.1011 | 43.8 | 112.4 |
| MLP | 0.9573 | 0.2445 | 0.9363 | 0.4957 | 0.8132 | 0.1026 | 4.3 | 477.1 |
| LSTM | 0.9849 | 0.8794 | 0.9848 | 0.5138 | 0.7617 | 0.0946 | 33.8 | 1924.3 |
| TimeXer | 0.9653 | 0.6907 | 0.9639 | 0.5001 | 0.7858 | 0.0984 | 24.1 | 695.2 |
| TimeMixer | 0.9573 | 0.2445 | 0.9363 | 0.4960 | 0.8132 | 0.1026 | 13.7 | 506.6 |
| iTransformer | 0.9588 | 0.4533 | 0.9462 | 0.4959 | 0.7982 | 0.1004 | 6.7 | 475.1 |
| PatchTST | 0.9707 | 0.7168 | 0.9672 | 0.4985 | 0.7730 | 0.0963 | 25.6 | 419.2 |
| TimesNet | 0.9735 | 0.6196 | 0.9690 | 0.4988 | 0.7699 | 0.0957 | 1216.5 | 56545.2 |
| DLinear | 0.9573 | 0.2445 | 0.9363 | 0.4957 | 0.8135 | 0.1027 | 4.7 | 723.6 |
| NT | 0.9714 | 0.7284 | 0.9700 | 0.4974 | 0.7717 | 0.0960 | 643.2 | 820.1 |
| FEDformer | 0.9580 | 0.4057 | 0.9435 | 0.4932 | 0.7972 | 0.1001 | 154.2 | 1301.0 |
| Pyraformer | 0.9761 | 0.7838 | 0.9741 | 0.5061 | 0.7702 | 0.0959 | 1179.4 | 1341.3 |
| Autoformer | 0.9547 | 0.3410 | 0.9411 | 0.4822 | 0.7894 | 0.0987 | 226.5 | 703.1 |
| Informer | 0.9638 | 0.6456 | 0.9628 | 0.4932 | 0.7805 | 0.0975 | 113.8 | 908.9 |
| Reformer | 0.9681 | 0.6970 | 0.9666 | 0.4951 | 0.7752 | 0.0966 | 382.2 | 655.5 |
| MICN | 0.9577 | 0.3732 | 0.9428 | 0.4951 | 0.8008 | 0.1007 | 65.0 | 2560.4 |
| Crossformer | 0.9747 | 0.7744 | 0.9732 | 0.5008 | 0.7686 | 0.0956 | 40.4 | 2814.5 |
| FiLM | 0.9594 | 0.5188 | 0.9498 | 0.4947 | 0.7950 | 0.0998 | 106.1 | 12585.3 |
| SCINet | 0.9623 | 0.5764 | 0.9550 | 0.4991 | 0.7936 | 0.0997 | 82.9 | 363.2 |
| PAttn | 0.9630 | 0.5667 | 0.9555 | 0.4931 | 0.7827 | 0.0978 | 20.7 | 165.4 |
| FreTS | 0.9702 | 0.7117 | 0.9670 | 0.4990 | 0.7747 | 0.0966 | 35.0 | 20034.4 |

Table 10: Azure dataset. NT for Nonstationary_Transformer. F1-mac stands for F1-macro score, and F1-w stands for F1-weighted score. ECE stands for expected calibration error. NLL stands for negative log likelihood. Parameters(k) indicate the number of parameters in thousands. The red font represents the best, the green font represents the second best, and the blue font represents the third best.

| model | ACC | F1-mac | F1-w | ECE | NLL | Brier | time cost | parameters(k) |
|---|---|---|---|---|---|---|---|---|
| SVM | 0.1667 | 0.1218 | 0.1563 | 0.1349 | 1.6682 | 0.1650 | 0.6 | 0.6 |
| XGB | 0.1429 | 0.0658 | 0.0626 | 0.2214 | 1.7155 | 0.1709 | 23.1 | 58.0 |
| MLP | 0.6024 | 0.1504 | 0.4529 | 0.3442 | 1.4942 | 0.1500 | 1.6 | 477.3 |
| LSTM | 0.6024 | 0.1504 | 0.4529 | 0.3231 | 1.4468 | 0.1454 | 1.5 | 1929.3 |
| TimeXer | 0.5976 | 0.1684 | 0.4631 | 0.3140 | 1.4329 | 0.1441 | 2.3 | 751.9 |
| TimeMixer | 0.6000 | 0.1628 | 0.4598 | 0.3015 | 1.4218 | 0.1430 | 2.4 | 584.1 |
| iTransformer | 0.6024 | 0.2325 | 0.5037 | 0.3362 | 1.4541 | 0.1462 | 1.7 | 475.4 |
| PatchTST | 0.5976 | 0.1499 | 0.4513 | 0.3138 | 1.4395 | 0.1446 | 1.2 | 475.8 |
| TimesNet | 0.6048 | 0.1574 | 0.4583 | 0.3277 | 1.4471 | 0.1455 | 14.4 | 56622.6 |
| DLinear | 0.4238 | 0.2599 | 0.4320 | 0.2133 | 1.5913 | 0.1585 | 2.2 | 726.6 |
| NT | 0.6024 | 0.1504 | 0.4529 | 0.3092 | 1.4173 | 0.1424 | 6.8 | 897.7 |
| FEDformer | 0.6024 | 0.1504 | 0.4529 | 0.3105 | 1.4325 | 0.1438 | 2.8 | 1378.4 |
| Pyraformer | 0.6024 | 0.1504 | 0.4529 | 0.3242 | 1.4413 | 0.1451 | 14.7 | 1573.6 |
| Autoformer | 0.5762 | 0.1640 | 0.4520 | 0.2949 | 1.4363 | 0.1444 | 2.3 | 780.5 |
| Informer | 0.6024 | 0.1504 | 0.4529 | 0.3116 | 1.4223 | 0.1430 | 1.8 | 986.4 |
| Reformer | 0.6024 | 0.1504 | 0.4529 | 0.3140 | 1.4292 | 0.1437 | 6.6 | 732.9 |
| MICN | 0.6000 | 0.1502 | 0.4525 | 0.3122 | 1.4391 | 0.1446 | 1.9 | 2564.7 |
| Crossformer | 0.6048 | 0.1640 | 0.4630 | 0.3278 | 1.4386 | 0.1447 | 2.0 | 2816.6 |
| FiLM | 0.6095 | 0.1767 | 0.4723 | 0.3759 | 1.5320 | 0.1534 | 4.1 | 12585.9 |
| SCINet | 0.5310 | 0.1728 | 0.4474 | 0.2460 | 1.4554 | 0.1460 | 2.3 | 365.0 |
| PAttn | 0.5833 | 0.1604 | 0.4541 | 0.3002 | 1.4496 | 0.1455 | 1.5 | 173.2 |
| FreTS | 0.6024 | 0.1504 | 0.4529 | 0.3016 | 1.4142 | 0.1422 | 1.6 | 20144.6 |

Table 11: Electric Motor dataset. NT for Nonstationary_Transformer. F1-mac stands for F1-macro score, and F1-w stands for F1-weighted score. ECE stands for expected calibration error. NLL stands for negative log likelihood. Parameters(k) indicate the number of parameters in thousands.The red font represents the best, the green font represents the second best, and the blue font represents the third best.

| model | ACC | F1-mac | F1-w | ECE | NLL | Brier | time cost | parameters(k) |
|---|---|---|---|---|---|---|---|---|
| SVM | 0.2727 | 0.2595 | 0.2745 | 0.0771 | 1.3559 | 0.1837 | 2.3 | 0.8 |
| XGB | 0.8281 | 0.8353 | 0.8303 | 0.4199 | 0.9690 | 0.1272 | 23.6 | 130.2 |
| MLP | 0.3676 | 0.2320 | 0.3033 | 0.1041 | 1.3719 | 0.1857 | 0.3 | 464.3 |
| LSTM | 0.3424 | 0.1884 | 0.2530 | 0.0744 | 1.3632 | 0.1847 | 1.3 | 1470.3 |
| TimeXer | 0.3375 | 0.1962 | 0.2586 | 0.0544 | 1.3619 | 0.1845 | 0.4 | 679.3 |
| TimeMixer | 0.6551 | 0.5935 | 0.6423 | 0.2990 | 1.1554 | 0.1553 | 0.3 | 455.4 |
| iTransformer | 0.6952 | 0.6951 | 0.6946 | 0.3935 | 1.2337 | 0.1674 | 0.3 | 462.3 |
| PatchTST | 0.9165 | 0.8952 | 0.9154 | 0.4627 | 0.8352 | 0.1063 | 0.5 | 415.6 |
| TimesNet | 0.9931 | 0.9921 | 0.9931 | 0.5220 | 0.7563 | 0.0938 | 11.4 | 56494.0 |
| DLinear | 0.4407 | 0.3855 | 0.4225 | 0.1693 | 1.3563 | 0.1836 | 1.1 | 503.0 |
| NT | 0.8735 | 0.8417 | 0.8737 | 0.4280 | 0.8754 | 0.1125 | 1.8 | 755.5 |
| FEDformer | 0.5030 | 0.4808 | 0.4972 | 0.1769 | 1.2557 | 0.1699 | 1.5 | 1249.8 |
| Pyraformer | 0.8799 | 0.8619 | 0.8820 | 0.4472 | 0.8924 | 0.1153 | 5.3 | 1187.7 |
| Autoformer | 0.7144 | 0.6666 | 0.7047 | 0.3483 | 1.1118 | 0.1480 | 0.9 | 651.9 |
| Informer | 0.7218 | 0.6340 | 0.6988 | 0.3394 | 1.0771 | 0.1435 | 1.2 | 857.7 |
| Reformer | 0.7367 | 0.6517 | 0.7159 | 0.3498 | 1.0597 | 0.1409 | 2.0 | 604.3 |
| MICN | 0.3394 | 0.2032 | 0.2689 | 0.0722 | 1.3712 | 0.1856 | 0.5 | 2311.9 |
| Crossformer | 0.3207 | 0.1888 | 0.2506 | 0.0444 | 1.3638 | 0.1847 | 0.4 | 2810.7 |
| FiLM | 0.4407 | 0.2624 | 0.3463 | 0.1673 | 1.3500 | 0.1828 | 1.0 | 12584.9 |
| SCINet | 0.4457 | 0.3856 | 0.4269 | 0.1386 | 1.3017 | 0.1763 | 1.1 | 252.8 |
| PAttn | 0.4170 | 0.3428 | 0.3915 | 0.1191 | 1.3281 | 0.1799 | 0.4 | 160.3 |
| FreTS | 0.4634 | 0.3647 | 0.4237 | 0.1555 | 1.3094 | 0.1773 | 0.5 | 16706.4 |

Table 12: Rotor Broken Motor dataset. NT for Nonstationary_Transformer. F1-mac stands for F1-macro score, and F1-w stands for F1-weighted score. ECE stands for expected calibration error. NLL stands for negative log likelihood. Parameters(k) indicate the number of parameters in thousands. The red font represents the best, the green font represents the second best, and the blue font represents the third best.

| model | ACC | F1-mac | F1-w | ECE | NLL | Brier | time cost | parameters(k) |
|---|---|---|---|---|---|---|---|---|
| SVM | 0.1677 | 0.1116 | 0.1103 | 0.0132 | 1.6096 | 0.1600 | 2.9 | 0.5 |
| XGB | 0.8785 | 0.8785 | 0.8785 | 0.5156 | 1.0728 | 0.1069 | 29.3 | 129.5 |
| MLP | 0.2068 | 0.1317 | 0.1313 | 0.0001 | 1.6088 | 0.1599 | 1.2 | 477.1 |
| LSTM | 0.3067 | 0.1823 | 0.1792 | 0.0943 | 1.5949 | 0.1588 | 1.7 | 1924.3 |
| TimeXer | 0.4362 | 0.4154 | 0.4109 | 0.1921 | 1.5169 | 0.1518 | 0.6 | 700.0 |
| TimeMixer | 0.2115 | 0.2025 | 0.2031 | 0.0064 | 1.6111 | 0.1601 | 0.4 | 583.4 |
| iTransformer | 0.5408 | 0.5254 | 0.5215 | 0.3072 | 1.5035 | 0.1511 | 0.6 | 475.3 |
| PatchTST | 0.7213 | 0.7193 | 0.7184 | 0.3926 | 1.2239 | 0.1232 | 0.3 | 424.1 |
| TimesNet | 0.8556 | 0.8574 | 0.8577 | 0.4898 | 1.0739 | 0.1066 | 23.3 | 56622.0 |
| DLinear | 0.1974 | 0.1131 | 0.1111 | 0.0060 | 1.6096 | 0.1600 | 0.4 | 724.2 |
| NT | 0.7885 | 0.7902 | 0.7889 | 0.4455 | 1.1493 | 0.1149 | 2.8 | 896.9 |
| FEDformer | 0.2449 | 0.1844 | 0.1846 | 0.0026 | 1.6101 | 0.1600 | 2.1 | 1377.8 |
| Pyraformer | 0.8482 | 0.8493 | 0.8491 | 0.4626 | 1.0574 | 0.1046 | 1.6 | 1571.7 |
| Autoformer | 0.2308 | 0.1801 | 0.1819 | 0.0271 | 1.5996 | 0.1591 | 1.5 | 779.9 |
| Informer | 0.7662 | 0.7649 | 0.7635 | 0.4261 | 1.1824 | 0.1183 | 1.0 | 985.7 |
| Reformer | 0.5209 | 0.5158 | 0.5125 | 0.2472 | 1.4214 | 0.1427 | 3.5 | 732.3 |
| MICN | 0.2429 | 0.2362 | 0.2351 | 0.0315 | 1.6025 | 0.1594 | 2.6 | 2564.6 |
| Crossformer | 0.4416 | 0.4158 | 0.4119 | 0.2035 | 1.5173 | 0.1519 | 0.5 | 2816.2 |
| FiLM | 0.1903 | 0.0959 | 0.0949 | 0.0124 | 1.6094 | 0.1600 | 1.6 | 12585.9 |
| SCINet | 0.2237 | 0.2239 | 0.2237 | 0.0016 | 1.6104 | 0.1601 | 1.2 | 363.8 |
| PAttn | 0.5921 | 0.5923 | 0.5903 | 0.2994 | 1.3670 | 0.1379 | 0.3 | 173.2 |
| FreTS | 0.2696 | 0.2347 | 0.2324 | 0.0575 | 1.5945 | 0.1588 | 1.1 | 20111.2 |

Table 13: MFPT dataset. NT for Nonstationary_Transformer. F1-mac stands for F1-macro score, and F1-w stands for F1-weighted score. ECE stands for expected calibration error. NLL stands for negative log likelihood. Parameters(k) indicate the number of parameters in thousands. The red font represents the best, the green font represents the second best, and the blue font represents the third best.

| model | ACC | F1-mac | F1-w | ECE | NLL | Brier | time cost | parameters(k) |
|---|---|---|---|---|---|---|---|---|
| SVM | 0.1755 | 0.0996 | 0.0524 | 0.4006 | 1.3759 | 0.2900 | 2.5 | 0.5 |
| XGB | 0.7089 | 0.6740 | 0.6531 | 0.2497 | 0.8657 | 0.1685 | 21.5 | 135.7 |
| MLP | 0.4961 | 0.4002 | 0.4481 | 0.0911 | 1.0289 | 0.2073 | 0.8 | 477.1 |
| LSTM | 0.5000 | 0.2222 | 0.3333 | 0.1605 | 1.0974 | 0.2219 | 2.2 | 1924.3 |
| TimeXer | 0.9071 | 0.8957 | 0.9074 | 0.3489 | 0.6457 | 0.1133 | 1.0 | 698.4 |
| TimeMixer | 0.4937 | 0.4634 | 0.4908 | 0.0320 | 1.0081 | 0.2018 | 1.0 | 429.8 |
| iTransformer | 0.7418 | 0.7272 | 0.7421 | 0.1976 | 0.8047 | 0.1518 | 1.1 | 475.0 |
| PatchTST | 0.9514 | 0.9431 | 0.9516 | 0.3857 | 0.6028 | 0.1026 | 0.8 | 414.3 |
| TimesNet | 0.9835 | 0.9839 | 0.9835 | 0.4116 | 0.5697 | 0.0944 | 21.8 | 56468.4 |
| DLinear | 0.4236 | 0.3049 | 0.3846 | 0.0271 | 1.0705 | 0.2159 | 1.2 | 723.0 |
| NT | 0.9154 | 0.9255 | 0.9154 | 0.3652 | 0.6423 | 0.1126 | 5.1 | 743.3 |
| FEDformer | 0.9385 | 0.9410 | 0.9384 | 0.3773 | 0.6162 | 0.1060 | 4.4 | 1224.2 |
| Pyraformer | 0.9781 | 0.9800 | 0.9781 | 0.4026 | 0.5737 | 0.0953 | 14.4 | 1110.9 |
| Autoformer | 0.8656 | 0.8394 | 0.8609 | 0.3187 | 0.6881 | 0.1236 | 1.3 | 626.3 |
| Informer | 0.7480 | 0.7728 | 0.7468 | 0.2247 | 0.7908 | 0.1497 | 1.2 | 832.1 |
| Reformer | 0.9212 | 0.9231 | 0.9212 | 0.3613 | 0.6334 | 0.1102 | 5.2 | 578.7 |
| MICN | 0.8676 | 0.8787 | 0.8672 | 0.3501 | 0.7167 | 0.1314 | 1.7 | 2556.2 |
| Crossformer | 0.9863 | 0.9863 | 0.9863 | 0.4172 | 0.5723 | 0.0951 | 2.1 | 2812.9 |
| FiLM | 0.7629 | 0.7469 | 0.7615 | 0.2720 | 0.8233 | 0.1570 | 1.2 | 12584.7 |
| SCINet | 0.5404 | 0.4270 | 0.4985 | 0.1321 | 1.0209 | 0.2047 | 1.1 | 362.5 |
| PAttn | 0.9083 | 0.8927 | 0.9077 | 0.3564 | 0.6492 | 0.1143 | 0.6 | 157.6 |
| FreTS | 0.5878 | 0.6298 | 0.5925 | 0.1117 | 0.9342 | 0.1853 | 0.9 | 19957.6 |

Table 14: UoC dataset. NT for Nonstationary_Transformer. F1-mac stands for F1-macro score, and F1-w stands for F1-weighted score. ECE stands for expected calibration error. NLL stands for negative log likelihood. Parameters(k) indicate the number of parameters in thousands. The red font represents the best, the green font represents the second best, and the blue font represents the third best.

| model | ACC | F1-mac | F1-w | ECE | NLL | Brier | time cost | parameters(k) |
|---|---|---|---|---|---|---|---|---|
| SVM | 0.8932 | 0.8919 | 0.8907 | 0.6873 | 1.5704 | 0.0776 | 4.7 | 1.8 |
| XGB | 0.9703 | 0.9702 | 0.9704 | 0.7290 | 1.4400 | 0.0724 | 44.0 | 134.1 |
| MLP | 0.1815 | 0.1213 | 0.1193 | 0.0672 | 2.1896 | 0.0986 | 0.5 | 477.3 |
| LSTM | 0.1785 | 0.1541 | 0.1530 | 0.0584 | 2.1728 | 0.0981 | 1.9 | 1924.5 |
| TimeXer | 0.8078 | 0.8083 | 0.8067 | 0.6524 | 1.8981 | 0.0900 | 0.3 | 719.5 |
| TimeMixer | 0.9674 | 0.9677 | 0.9675 | 0.7284 | 1.4484 | 0.0728 | 0.3 | 890.6 |
| iTransformer | 0.7426 | 0.7380 | 0.7367 | 0.5979 | 1.9710 | 0.0924 | 0.4 | 475.8 |
| PatchTST | 0.9828 | 0.9831 | 0.9829 | 0.7384 | 1.4184 | 0.0716 | 1.2 | 443.5 |
| TimesNet | 0.9923 | 0.9923 | 0.9923 | 0.7435 | 1.3958 | 0.0707 | 11.9 | 56929.2 |
| DLinear | 0.4763 | 0.4653 | 0.4614 | 0.3640 | 2.1911 | 0.0986 | 1.1 | 726.6 |
| NT | 0.9893 | 0.9894 | 0.9893 | 0.7364 | 1.3834 | 0.0701 | 4.0 | 1204.1 |
| FEDformer | 0.9597 | 0.9598 | 0.9597 | 0.7222 | 1.4605 | 0.0733 | 1.6 | 1685.0 |
| Pyraformer | 0.9964 | 0.9965 | 0.9964 | 0.7442 | 1.3800 | 0.0700 | 11.0 | 2493.3 |
| Autoformer | 0.9543 | 0.9545 | 0.9546 | 0.7212 | 1.4809 | 0.0741 | 0.8 | 1087.1 |
| Informer | 0.9786 | 0.9786 | 0.9787 | 0.7372 | 1.4334 | 0.0722 | 1.3 | 1292.9 |
| Reformer | 0.9786 | 0.9787 | 0.9787 | 0.7365 | 1.4308 | 0.0721 | 2.3 | 1039.5 |
| MICN | 0.7633 | 0.7647 | 0.7642 | 0.5928 | 1.8346 | 0.0875 | 0.7 | 2581.4 |
| Crossformer | 0.8488 | 0.8485 | 0.8481 | 0.6830 | 1.8360 | 0.0878 | 0.4 | 2822.8 |
| FiLM | 0.5801 | 0.5613 | 0.5591 | 0.4480 | 2.0578 | 0.0948 | 0.9 | 12588.3 |
| SCINet | 0.2734 | 0.2705 | 0.2705 | 0.1381 | 2.1397 | 0.0972 | 0.6 | 366.2 |
| PAttn | 0.8511 | 0.8512 | 0.8512 | 0.6712 | 1.7595 | 0.0850 | 0.4 | 204.4 |
| FreTS | 0.9567 | 0.9571 | 0.9568 | 0.7293 | 1.4972 | 0.0749 | 0.3 | 20418.4 |

Table 15: Planetary dataset. NT for Nonstationary_Transformer. F1-mac stands for F1-macro score, and F1-w stands for F1-weighted score. ECE stands for expected calibration error. NLL stands for negative log likelihood. Parameters(k) indicate the number of parameters in thousands. The red font represents the best, the green font represents the second best, and the blue font represents the third best.

| model | ACC | F1-mac | F1-w | ECE | NLL | Brier | time cost | parameters(k) |
|---|---|---|---|---|---|---|---|---|
| SVM | 0.5843 | 0.4752 | 0.6187 | 0.4791 | 0.9867 | 0.3896 | 142.7 | 0.2 |
| XGB | 0.2135 | 0.1851 | 0.0940 | 0.4882 | 1.0265 | 0.4046 | 23.5 | 92.1 |
| MLP | 0.7549 | 0.4302 | 0.6883 | 0.0748 | 0.5727 | 0.1918 | 35.0 | 477.0 |
| LSTM | 0.7389 | 0.5001 | 0.7074 | 0.0447 | 0.5695 | 0.1908 | 114.1 | 1924.2 |
| TimeXer | 0.7689 | 0.4748 | 0.7106 | 0.0669 | 0.5455 | 0.1796 | 54.9 | 685.4 |
| TimeMixer | 0.6682 | 0.4106 | 0.6444 | 0.0163 | 0.6313 | 0.2197 | 35.6 | 353.0 |
| iTransformer | 0.7493 | 0.4625 | 0.6981 | 0.0464 | 0.5603 | 0.1865 | 15.9 | 474.9 |
| PatchTST | 0.7641 | 0.4649 | 0.7050 | 0.0543 | 0.5477 | 0.1807 | 28.8 | 447.6 |
| TimesNet | 0.7247 | 0.4636 | 0.6881 | 0.0184 | 0.5810 | 0.1963 | 1526.6 | 56391.6 |
| DLinear | 0.7993 | 0.4442 | 0.7108 | 0.1224 | 0.5401 | 0.1762 | 35.0 | 722.4 |
| NT | 0.6949 | 0.4634 | 0.6749 | 0.0124 | 0.6061 | 0.2080 | 180.2 | 666.5 |
| FEDformer | 0.7395 | 0.4448 | 0.6875 | 0.0280 | 0.5683 | 0.1903 | 231.8 | 1147.4 |
| Pyraformer | 0.7258 | 0.4683 | 0.6903 | 0.0277 | 0.5820 | 0.1966 | 706.5 | 880.5 |
| Autoformer | 0.7466 | 0.4625 | 0.6970 | 0.1322 | 0.6014 | 0.2052 | 195.4 | 549.5 |
| Informer | 0.7024 | 0.4491 | 0.6732 | 0.0175 | 0.5968 | 0.2037 | 403.4 | 755.3 |
| Reformer | 0.6716 | 0.4471 | 0.6585 | 0.0172 | 0.6190 | 0.2142 | 826.8 | 501.9 |
| MICN | 0.7350 | 0.4252 | 0.6784 | 0.0616 | 0.5875 | 0.1989 | 181.2 | 2552.0 |
| Crossformer | 0.7332 | 0.4893 | 0.7011 | 0.0285 | 0.5724 | 0.1923 | 37.6 | 2811.2 |
| FiLM | 0.7214 | 0.4381 | 0.6775 | 0.0228 | 0.5858 | 0.1984 | 152.4 | 12584.1 |
| SCINet | 0.7849 | 0.4505 | 0.7077 | 0.0914 | 0.5366 | 0.1753 | 48.2 | 361.9 |
| PAttn | 0.7690 | 0.4537 | 0.7027 | 0.0807 | 0.5530 | 0.1828 | 16.6 | 149.8 |
| FreTS | 0.6790 | 0.4285 | 0.6555 | 0.0048 | 0.6189 | 0.2140 | 31.8 | 19880.8 |

Table 16: MAFAULDA dataset. NT for Nonstationary_Transformer. F1-mac stands for F1-macro score, and F1-w stands for F1-weighted score. ECE stands for expected calibration error. NLL stands for negative log likelihood. Parameters(k) indicate the number of parameters in thousands. The red font represents the best, the green font represents the second best, and the blue font represents the third best.

| model | ACC | F1-mac | F1-w | ECE | NLL | Brier | time cost | parameters(k) |
|---|---|---|---|---|---|---|---|---|
| SVM | 0.1851 | 0.1219 | 0.1266 | 0.0097 | 1.7970 | 0.1392 | 15.4 | 2.3 |
| XGB | 0.0035 | 0.0033 | 0.0035 | 0.3334 | 2.0245 | 0.1563 | 4.4 | 184.7 |
| MLP | 0.4146 | 0.4080 | 0.3935 | 0.2273 | 1.7205 | 0.1349 | 16.9 | 403.3 |
| LSTM | 0.3854 | 0.3236 | 0.3031 | 0.1793 | 1.6584 | 0.1306 | 4.8 | 405.7 |
| TimeXer | 0.6229 | 0.6277 | 0.6194 | 0.3632 | 1.4692 | 0.1173 | 13.5 | 603.4 |
| TimeMixer | 0.4996 | 0.5034 | 0.4915 | 0.2655 | 1.5777 | 0.1250 | 13.0 | 217.1 |
| iTransformer | 0.6729 | 0.6837 | 0.6742 | 0.4070 | 1.4234 | 0.1138 | 13.3 | 401.5 |
| PatchTST | 0.5843 | 0.5898 | 0.5809 | 0.3331 | 1.5073 | 0.1200 | 6.9 | 400.5 |
| TimesNet | 0.6683 | 0.6768 | 0.6680 | 0.3981 | 1.4217 | 0.1136 | 29.6 | 56255.6 |
| DLinear | 0.4444 | 0.4414 | 0.4276 | 0.2377 | 1.6646 | 0.1311 | 17.0 | 1.2 |
| NT | 0.6650 | 0.6725 | 0.6631 | 0.3959 | 1.4203 | 0.1135 | 13.6 | 453.3 |
| FEDformer | 0.6599 | 0.6699 | 0.6604 | 0.3949 | 1.4331 | 0.1144 | 16.3 | 704.3 |
| Pyraformer | 0.6927 | 0.7011 | 0.6920 | 0.4218 | 1.4036 | 0.1123 | 20.6 | 472.7 |
| Autoformer | 0.5474 | 0.5501 | 0.5397 | 0.2958 | 1.5283 | 0.1214 | 26.5 | 413.6 |
| Informer | 0.6516 | 0.6610 | 0.6511 | 0.3856 | 1.4355 | 0.1146 | 18.8 | 619.4 |
| Reformer | 0.6677 | 0.6782 | 0.6691 | 0.4037 | 1.4326 | 0.1145 | 23.1 | 366.0 |
| MICN | 0.6509 | 0.6608 | 0.6506 | 0.3888 | 1.4425 | 0.1152 | 11.4 | 1152.0 |
| Crossformer | 0.6654 | 0.6754 | 0.6664 | 0.3997 | 1.4324 | 0.1145 | 11.9 | 2791.7 |
| FiLM | 0.5609 | 0.5745 | 0.5622 | 0.3272 | 1.5530 | 0.1232 | 31.8 | 4325.5 |
| SCINet | 0.5092 | 0.5105 | 0.4996 | 0.2754 | 1.5770 | 0.1249 | 41.1 | 1.4 |
| PAttn | 0.5832 | 0.5877 | 0.5793 | 0.3344 | 1.5113 | 0.1203 | 11.7 | 136.5 |
| FreTS | 0.5190 | 0.5254 | 0.5123 | 0.2901 | 1.5802 | 0.1252 | 12.0 | 837.8 |

Table 17: Mendeley dataset. NT for Nonstationary_Transformer. F1-mac stands for F1-macro score, and F1-w stands for F1-weighted score. ECE stands for expected calibration error. NLL stands for negative log likelihood. Parameters(k) indicate the number of parameters in thousands. The red font represents the best, the green font represents the second best, and the blue font represents the third best.

| model | ACC | F1-mac | F1-w | ECE | NLL | Brier | time cost | parameters(k) |
|---|---|---|---|---|---|---|---|---|
| SVM | 0.5417 | 0.3514 | 0.3806 | 0.1090 | 0.7220 | 0.2642 | 93.7 | 0.2 |
| XGB | 0.7170 | 0.7158 | 0.7221 | 0.1470 | 0.5572 | 0.1857 | 21.4 | 48.2 |
| MLP | 0.5769 | 0.5768 | 0.5743 | 0.1034 | 0.6519 | 0.2303 | 9.3 | 477.0 |
| LSTM | 0.9699 | 0.9666 | 0.9701 | 0.2482 | 0.3419 | 0.0856 | 119.8 | 1924.2 |
| TimeXer | 0.9806 | 0.9782 | 0.9806 | 0.2522 | 0.3324 | 0.0812 | 30.3 | 685.4 |
| TimeMixer | 0.6171 | 0.6169 | 0.6140 | 0.1065 | 0.6249 | 0.2181 | 29.4 | 353.0 |
| iTransformer | 0.9620 | 0.9577 | 0.9622 | 0.2389 | 0.3499 | 0.0893 | 17.6 | 474.9 |
| PatchTST | 0.9797 | 0.9772 | 0.9797 | 0.2535 | 0.3324 | 0.0812 | 70.8 | 409.5 |
| TimesNet | 0.8642 | 0.8395 | 0.8605 | 0.1477 | 0.4435 | 0.1327 | 1706.8 | 56391.6 |
| DLinear | 0.3964 | 0.3513 | 0.2942 | 0.1269 | 0.7080 | 0.2574 | 13.9 | 722.4 |
| NT | 0.9689 | 0.9656 | 0.9692 | 0.2497 | 0.3416 | 0.0855 | 588.8 | 666.5 |
| FEDformer | 0.9636 | 0.9596 | 0.9638 | 0.2408 | 0.3484 | 0.0886 | 1139.6 | 1147.4 |
| Pyraformer | 0.9541 | 0.9491 | 0.9544 | 0.2387 | 0.3599 | 0.0938 | 778.8 | 880.5 |
| Autoformer | 0.9146 | 0.9083 | 0.9163 | 0.2065 | 0.3978 | 0.1113 | 1214.2 | 549.5 |
| Informer | 0.9100 | 0.9022 | 0.9114 | 0.2032 | 0.3984 | 0.1117 | 125.0 | 755.3 |
| Reformer | 0.9444 | 0.9392 | 0.9451 | 0.2268 | 0.3698 | 0.0984 | 239.0 | 501.9 |
| MICN | 0.9760 | 0.9731 | 0.9761 | 0.2511 | 0.3353 | 0.0826 | 183.0 | 2552.0 |
| Crossformer | 0.9683 | 0.9648 | 0.9685 | 0.2479 | 0.3433 | 0.0862 | 95.2 | 2811.2 |
| FiLM | 0.9591 | 0.9543 | 0.9592 | 0.2497 | 0.3598 | 0.0937 | 267.9 | 12584.1 |
| SCINet | 0.9471 | 0.9415 | 0.9475 | 0.2392 | 0.3750 | 0.1004 | 61.0 | 361.9 |
| PAttn | 0.8428 | 0.8328 | 0.8464 | 0.1731 | 0.4704 | 0.1448 | 57.6 | 281.9 |
| FreTS | 0.9626 | 0.9585 | 0.9629 | 0.2500 | 0.3553 | 0.0916 | 104.8 | 19880.8 |

Table 18: Test of accuracy on different segmentation lengths using the XJTU dataset. Here, length 5 indicates the time-series sequence contains only 5 data points. length 10 indicates the time-series sequence contains only 10 data points. length 15 indicates the time-series sequence contains only 15 data points. length 20 indicates the time-series sequence contains only 20 data points. The red font represents the best, the green font represents the second best, and the blue font represents the third best.

| model | length 5 | length 10 | length 15 | length 20 |
|---|---|---|---|---|
| SVM | 0.3880 | 0.1815 | 0.1279 | 0.0855 |
| XGB | 0.1541 | 0.0615 | 0.0086 | 0.0060 |
| MLP | 0.3903 | 0.3903 | 0.4009 | 0.5659 |
| LSTM | 0.4165 | 0.3903 | 0.5355 | 0.7343 |
| iTransformer | 0.4733 | 0.4306 | 0.7646 | 0.8987 |
| DLinear | 0.4015 | 0.3903 | 0.4006 | 0.6904 |
| NT | 0.4759 | 0.4308 | 0.7642 | 0.9103 |
| FEDformer | 0.4682 | 0.4307 | 0.7594 | 0.8769 |
| Pyraformer | 0.4751 | 0.4311 | 0.7566 | 0.9079 |
| Autoformer | 0.4388 | 0.3964 | 0.7268 | 0.7951 |
| Informer | 0.4713 | 0.4306 | 0.7589 | 0.8934 |
| Reformer | 0.4735 | 0.4310 | 0.7594 | 0.8889 |
| MICN | 0.4454 | 0.4305 | 0.7512 | 0.8874 |
| Crossformer | 0.4735 | 0.4306 | 0.7430 | 0.9098 |
| FiLM | 0.4178 | 0.4153 | 0.5905 | 0.7868 |
| FreTS | 0.4248 | 0.4010 | 0.5531 | 0.7415 |

models' performance on the HUST and CRWU datasets. However, MLflow increases the training time costs a lot. Even though the performance can be improved but it still underperforms the transformer-based baselines.

Table 19: Test of accuracy on MLFlow using the Paderborn, HUST, CRWU, and XJTU dataset on XGBoost baseline. F1-mac stands for F1-macro score, and F1-w stands for F1-weighted score. ECE stands for expected calibration error. NLL stands for negative log likelihood.

| dataset | model | ACC | F1-mac | F1-w | ECE | NLL | Brier | time cost |
|---------|-------|-----|--------|------|-----|-----|-------|-----------|
| Padeborn | XGB | 0.2177 | 0.1904 | 0.2426 | 0.0774 | 1.4006 | 0.1894 | 40.0 |
| Padeborn | XGB_MLFlow | 0.2469 | 0.1314 | 0.2175 | 0.0512 | 1.4016 | 0.1894 | 36207.8 |
| HUST | XGB | 0.5469 | 0.5252 | 0.5390 | 0.3468 | 1.7222 | 0.1116 | 78.6 |
| HUST | XGB_MLFlow | 0.2888 | 0.2145 | 0.2332 | 0.1429 | 1.9386 | 0.1221 | 41183.9 |
| CRWU | XGB | 0.9832 | 0.9801 | 0.9832 | 0.5113 | 0.7623 | 0.0947 | 1.7 |
| CRWU | XGB_MLFlow | 0.6639 | 0.4153 | 0.6259 | 0.3960 | 1.3444 | 0.1821 | 37330.7 |
| XJTU | XGB | 0.0060 | 0.1914 | 0.0061 | 0.3955 | 1.8958 | 0.1899 | 4.1 |
| XJTU | XGB_MLFlow | 0.0706 | 0.0264 | 0.0093 | 0.1294 | 1.6094 | 0.1600 | 41350.8 |

### D.3.4 EXPLANATION ANALYSIS

In this section, we present an explanatory analysis of model performance across different benchmarks. To provide interpretability, we employ SHAP values Lundberg & Lee (2017), which attribute model outputs to individual features. As a representative case study, we use the Paderborn dataset to illustrate how our framework generates quantitative insights into both the how and why of model predictions. This analysis not only clarifies the behavior of baseline models on benchmark datasets but also supports deeper understanding and more informed development in the predictive maintenance domain.

As shown in Figure 6, the contribution of individual features varies substantially across classes. For instance, in class 0 (healthy bearings), the most influential feature is f297 (the 297th point in the sequence), whereas in class 1, the dominant feature shifts to f593. Moreover, the SHAP distributions for class 0 appear narrower along the x-axis compared to those of faulty classes, suggesting that feature contributions in the healthy class are more evenly distributed. This observation aligns with domain knowledge: vibration signals from healthy bearings are typically stable, resulting in relatively uniform feature importance, while faulty bearings exhibit localized spikes or anomalies that strongly influence the model's predictions.

To quantify signal quality, we provide a variance-based analysis. We observe that datasets such as Gearbox UoC (variance = 0.07) and Paderborn (variance = 0.96) present much lower variability in signal compared to datasets like MAFAULDA (variance = 3379.56) and Mendeley (variance = 2.8 million). This variation translates directly into model performance: while models such as TimesNet and PatchTST achieve F1 scores exceeding 95% on structured datasets like FEMTO and HUST, their performance drops substantially (by 20–30%) on motor datasets with weaker structure and high-frequency noise. For example, Crossformer and TimesNet show strong performance on bearings but suffer considerable degradation on Electric Motor, indicating their sensitivity to unstructured signal dynamics. Furthermore, DLinear, a simple and lightweight model, performs surprisingly well on several low-SNR datasets. Its strong inductive bias and low inference cost make it robust under noisy or weakly labeled scenarios, unlike many high-capacity models that overfit or fail to generalize.

These observations are only made possible through PDMBench's standardized evaluation framework, which ensures consistent preprocessing, segment length, and train-test splits across 14 diverse datasets. This enables meaningful, apples-to-apples comparisons across fault types, sensor modalities, and operational conditions, precisely the kind of diagnostic analysis that was previously infeasible due to pipeline fragmentation in PdM research. Furthermore, we plan to expand this analysis with dedicated failure mode taxonomies, case studies on model misclassification patterns, and automated tools for failure clustering. Our interface will also be extended to include interpretability dashboards and signal quality diagnostics to help practitioners and researchers pinpoint failure causes and refine model architectures accordingly.

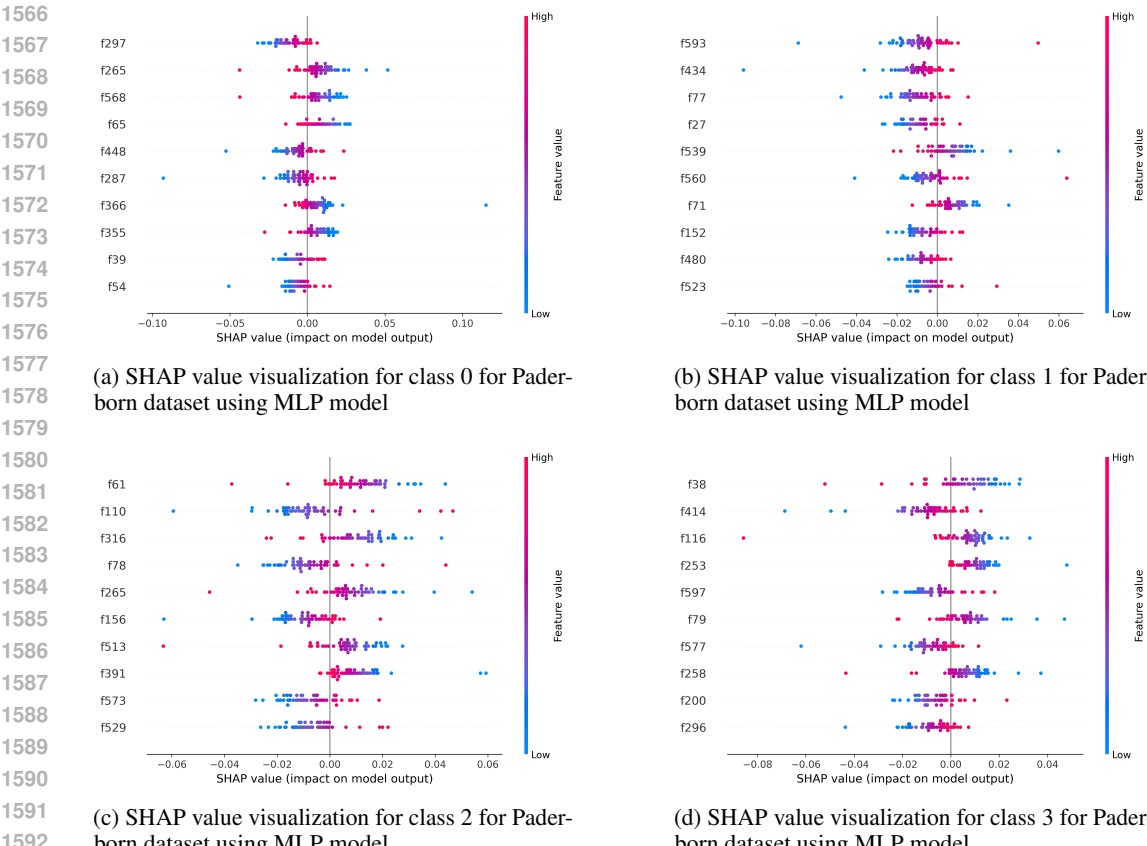

(a) SHAP value visualization for class 0 for Paderborn dataset using MLP model

(b) SHAP value visualization for class 1 for Paderborn dataset using MLP model

(c) SHAP value visualization for class 2 for Paderborn dataset using MLP model

(d) SHAP value visualization for class 3 for Paderborn dataset using MLP model

Figure 6: SHAP value for MLP model on Paderborn dataset

Table 20: The average of variance over the normalized signal for each dataset.

| dataset | variance |
|---|---|
| Padeborn | 0.96 |
| HUST | 0.99 |
| IMS | 0.99 |
| CRWU | 746.17 |
| XJTU | 1626.96 |
| MFPT | 1.00 |
| FEMTO | 1966.60 |
| MAFAULDA | 3379.56 |
| Mendeley | 2847302.65 |
| Planetary | 14.41 |
| Azure | 2.67 |
| Electric Motor Vibrations | 1815.03 |
| RotorBrokenBar | 10.00 |
| GearBoxUoC | 0.07 |

# E    USER INTERFACE GUIDELINE

Figure 7 presents an interactive environment for exploring sensor modalities in time and frequency domains. Upon dataset selection, users can visualize the distribution of specific features (e.g., vibration, current) and identify class imbalances. This step helps assess data quality, temporal patterns, and spectral characteristics crucial for model design and preprocessing decisions.

In Figure 8, users can configure model parameters (e.g., hidden size, number of layers), initiate training, and monitor logs in real time. The interface outputs test metrics in terms of prediction performance, uncertainty and efficiency, enabling in-depth performance diagnostics.

Figure 9 illustrates the PDMBench result analysis interface, which consolidates performance metrics across multiple datasets and models. The interface computes the average, standard deviation, and dataset coverage for each evaluation metric, enabling a holistic assessment of model behavior. It visualizes key aspects such as predictive accuracy, uncertainty calibration, and computational efficiency across diverse datasets. This comprehensive overview assists practitioners in selecting the most suitable model for a specific predictive maintenance (PdM) task and guides further model refinement and optimization.

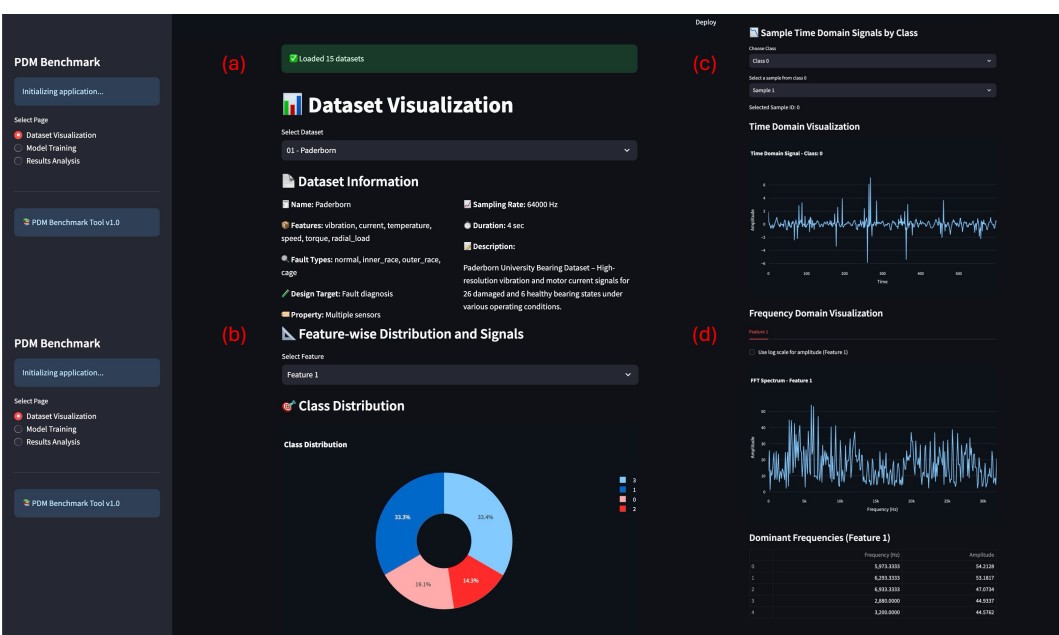

Figure 7: Data Visualization Section of PDMBench: (a) Users select a dataset. (b) Visualize feature distributions. (c) Inspect time-domain signals. (d) Analyze frequency domain characteristics via FFT.

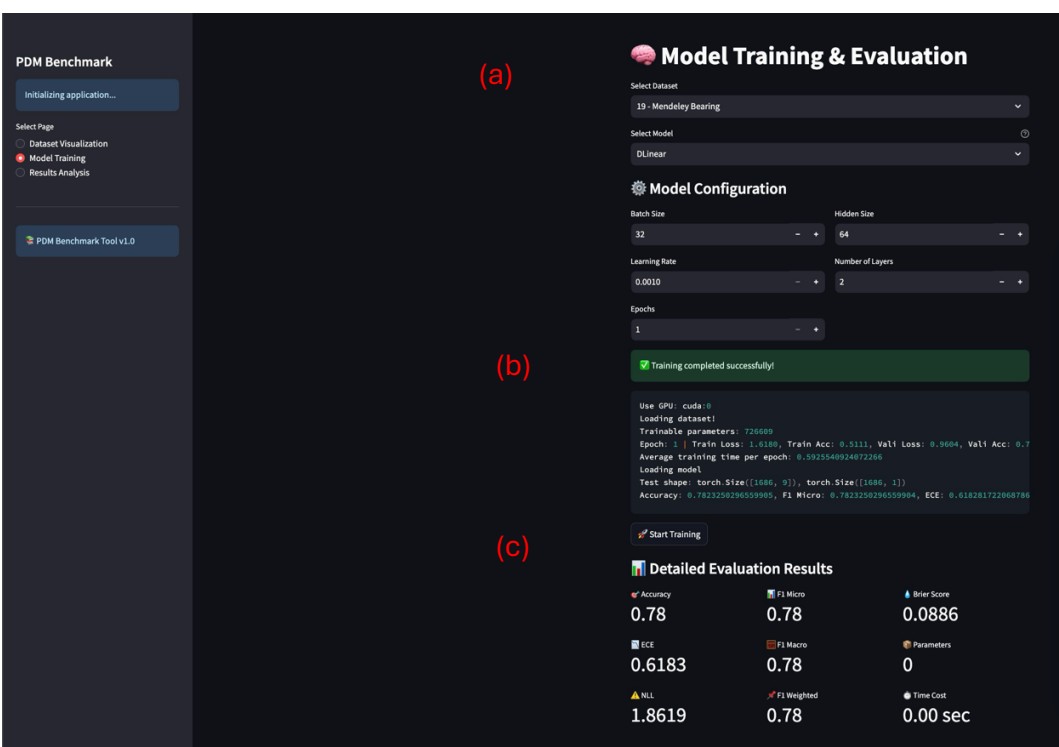

Figure 8: Model Training Section of PDMBench: (a) Users select a dataset and config a model. (b) Training log with real-time feedback. (c) Evaluation metrics are reported during and after execution.

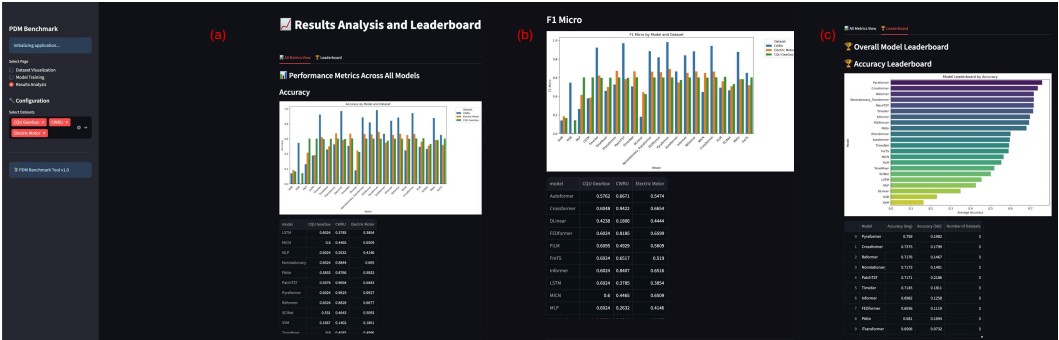

Figure 9: Result Analysis Section of PDMBench: (a,b) Cross-dataset performance comparison with different evaluation metrics. (c) A leaderboard analysis across various models and metrics.

