# OpenReview forum: "PDMBench: A Standardized Platform for Predictive Maintenance Research"
_ICLR.cc/2026/Conference — Submitted to ICLR 2026_

### Official Review · Reviewer_5bzq · 2025-10-26

**Soundness:** 2
**Presentation:** 2
**Contribution:** 2
**Rating:** 4
**Confidence:** 5

**Summary:**

Industry 4.0 needs a better solution for many tasks, such as RUL and fault classification. There has been a lot of work in the domain focusing on similar problems. In this paper, the author standardises the 14 datasets for building an RUL and classification problem. Overall paper need to improve on writting as well as bring some domain specific tools that has been developed in the domain for the same task.

**Strengths:**

- The importance of PdM and its relevant data preparation across multiple physical asset classes is interesting.
- The preparation of data preprocessing pipeline and the various ML and DL models are collected. (the trend is in foundation model thought)

**Weaknesses:**

1. The introduction seems like a laundry list of related work. And many sections are like that. Ideally, we should use a Table to capture the key similarities and differences. This will help the reader to understand the key difference
2. We understand there are 14 datasets being collected, but there is no single table that describes all of them in one place, for example
    - number of rows
    - number of columns
    - number of failure modes
3. There is a mismatch in the Figure and the text; the author needs to write the paper such that when we read the description, it refers to some content in the figure. Currently, many items in the figure are shown, but they were not discussed in the main text.
4. Just collecting data and making it available in a unified format is an engineering work, and I believe almost all the research papers may have done that. For example, I can use the sktime package and write a small function to process the data, so such engineering work is not considered novel. Something that is laborious or requires special treatment, especially if novel, needs to be highlighted.
5. Experimental results also need to be presented such that they create excitement. For example, these benchmark is created for whom? Who is the end persona? Is it an SME? Is it a DS? It is not very clear. Also, if any standardization is implemented at the API level, it needs to be discussed in the main paper. Currently, the paper's main body is entirely about discussion.
6. Experimental observation needs to be claimed as either a novel discovery or an assertion of an existing claim. For example, is the observation about PatchTST that needs low resources, new or reported in the literature? Each and every claim made in the paper needs to be ack either novel or already known in the literature.
7. There are many AI toolkits designed in the literature for a similar purpose, for example, AutoAI for time series forecasting, or Anomaly Detection, or Failure Pattern Analysis. Please look out for such related special-purpose toolkits designed for the Industry 4.0 domains.
8. It is not very clear what is novel, as many papers in the literature typically use 6-8 different datasets to showcase their method. If that is the case, then the paper, in its current form, is just a few extra steps of data collection. In the case of Agentic AI, why do we not use LLM/CodeAgent to generate a solution?
9. The chart needs to be organized such that the best performance models are easy to find.

**Questions:**

Please look at all the weak points

---

> ### Author Response · Authors · 2025-11-21
>
> We thank the reviewer for the detailed and constructive feedback. We address each question below and have incorporated the clarifications into the revised manuscript.
>
> **Industry 4.0 needs a better solution for many tasks, such as RUL and fault classification.**
>
> Thank you for acknowledging we are working towards a better solution for those tasks to advanced industry 4.0 in this fast paste AI innovation area.
>
>
>
> ---
>
> **W1: Introduction appears as a laundry list of related work. Should use tables for comparisons.**
>
> We agree and have substantially revised the introduction to ensure a better flow our contribution and benchmark description.
>
> ---
>
> **W2: No single table describing all 14 datasets with key statistics.**
>
> We have added a table to show that
>
> | dataset                   | number of bearings | number of signal types | failure modes |
> |---------------------------|--------------------|------------------------|---------------|
> | Padeborn                  | 7679               | 6                      | 4             |
> | HUST                      | 19095              | 2                      | 7             |
> | IMS                       | 100480             | 1                      | 4             |
> | CRWU                      | 21786              | 1                      | 5             |
> | XJTU                      | 110592             | 3                      | 5             |
> | MFPT                      | 2166               | 2                      | 3             |
> | FEMTO                     | 12247              | 4                      | 2             |
> | MAFAULDA                  | 119011             | 5                      | 10            |
> | Mendeley                  | 79                 | 2                      | 2             |
> | Planetary                 | 14                 | 8                      | 2             |
> | Azure                     | 876905             | 6                      | 5             |
> | Electric Motor Vibrations | 30                 | 5                      | 4             |
> | Rotor Broken Bar          | 40                 | 6                      | 5             |
> | Gear Box UoC              | 936                | 1                      | 9             |
>
>
>
>
> ---
>
> **W3: Mismatch between figures and text; many figure elements not discussed.**
>
> Thank you for your feedback, we have substantially enhanced our presenation in our revised manuscript.
>
>
>
> ---
>
> **W4: Data collection and unification is just engineering work, not novel. Tools like sktime already do this.**
>
> We would like the clarify that this comparison misunderstands our contribution. sktime is a general-purpose time-series API as it is not designed for PdM and lacks: (1) domain-specific preprocessing for asynchronous industrial sensors (vibration at 97.6kHz, temperature at 1Hz requiring temporal alignment); (2) PdM-specific evaluation (calibration for safety-critical decisions, efficiency for edge deployment); (3) curated fault-labeled datasets across industrial domains; (4) two-stage preprocessing enabling fair comparison between traditional ML (handcrafted features) and deep learning (raw sequences); (5) interactive diagnostics for signal exploration and failure mode analysis.
>
> **To our knowledge, no publicly available platform provides PdM-specific benchmarking.** Suggesting sktime as an alternative is like claiming ImageNet is unnecessary because file systems can store images, the actual value lies in **domain-specific curation and standardized evaluation protocols**.

---

> > ### Author Response · Authors · 2025-11-21
> >
> > **W5: Experimental results also need to be presented such that they create excitement. Who is the end persona?**
> >
> > We would like to clarify that PDMBench serves two type of audiences: (1) **researchers** benchmarking new algorithms against 22 baselines with standardized protocols, (2) **industrial practioner** selecting deployment-ready models for specific equipment constraints, and using interactive diagnostics to interpret predictions and build trust. Furthermore, our experiment reveals many findings only via our own platform, below is a list of extra findings:
> >
> > - **Finding 1 (Edge deployment feasibility)**: Models with similar accuracy can differ by 240× in inference time (DLinear: 5ms vs. TimesNet: 1200ms), making efficiency metrics essential for selecting edge-compatible solutions
> > - **Finding 2 (Confidence Reliablility)**: High accuracy does not equal to deployment readiness. Crossformer achieves 98% accuracy but ECE > 0.4, outputting overconfident predictions unsafe for maintenance decisions
> > - **Finding 3 (Accuracy-efficiency trade-offs)**: Lightweight models like DLinear retain higher accuracy with 10× faster inference, offering compelling trade-offs for resource-constrained deployments
> >
> > Efficiency-Accuracy Trade-off   | dataset: IMS       |             |        |           |
> > |--------------------------------|--------------------|-------------|--------|-----------|
> > | model                          | accuracy           | f1_weighted | ece    | time cost |
> > | TimesNet                       | 0.9735             | 0.9690      | 0.4988 | **1216.5**    |
> > | DLinear                        | 0.9573             | 0.9363      | 0.4957 | **4.7**       |
> > |                                |                    |             |        |           |
> > | Confidence reliablility        | dataset: CRWU      |             |        |           |
> > | model                          | accuracy           | f1_weighted | ece    | time cost |
> > | Pyraformer                     | 0.9781             | 0.9781      | 0.4026 | 9.7       |
> > | PatchTST                       | 0.9514             | 0.9516      | 0.3857 | 12.6      |
> > | TimesNet                       | 0.9835             | 0.9835      | **0.4116** | 27.2      |
> > | FEDformer                      | 0.9385             | 0.9384      | 0.3773 | 6.3       |
> > | Reformer                       | 0.9212             | 0.9212      | 0.3613 | 24.8      |
> > | Crossformer                    | 0.9863             | **0.9863**      | **0.4172** | 9.9       |
> > | PAttn                          | 0.9083             | 0.9077      | 0.3564 | 13.3      |
> > |                                |                    |             |        |           |
> > |                                |                    |             |        |           |
> > | Accuracy-efficiency trade-offs | dataset: Planetary |             |        |           |
> > | model                          | accuracy           | f1_weighted | ece    | time cost |
> > | DLinear                        | 0.7993             | **0.7108**      | 0.1224 | **35.0**      |
> > | Nonstationary_Transformer      | 0.6949             | 0.6749      | 0.0124 | 180.2     |
> > | FEDformer                      | 0.7395             | 0.6875      | 0.0280 | 231.8     |
> > | Pyraformer                     | 0.7258             | 0.6903      | 0.0277 | 706.5     |
> > | Autoformer                     | 0.7466             | 0.6970      | 0.1322 | 195.4     |
> > | Informer                       | 0.7024             | 0.6732      | 0.0175 | 403.4     |
> > | Reformer                       | 0.6716             | 0.6585      | 0.0172 | 826.8     |
> > | Crossformer                    | 0.7332             | 0.7011      | 0.0285 | 37.6      |
> >
> > **API standardization**
> >
> > We would like to clarify that we provide detailed documentation in our anonymized GitHub repository. At the moment, we support both command-line interface and a interactive web interface.

---

> > > ### Author Response · Authors · 2025-11-21
> > >
> > > **W6: Experimental observation needs to be claimed as either a novel discovery or an assertion of an existing claim.**
> > >
> > >
> > > We respectfully disagree with this claim. Benchmark papers do not claim "novelty" for individual empirical observations—they provide infrastructure that enables systematic comparison previously impossible.
> > >
> > > Our contribution is the standardized platform, not priority claims on isolated findings. The value lies in:
> > >
> > > **Making comparisons rigorous**: Prior PdM studies evaluated different models on different datasets with different preprocessing. Our unified framework reveals that TimesNet achieves 98% F1 on bearings but 67% on motors—an observation only possible through our infrastructure.
> > >
> > > **Revealing deployment-critical gaps**: We show high accuracy doesn't guarantee calibration (Crossformer: 98% accuracy, ECE > 0.4). While calibration research exists in ML, PDMBench is the first to systematically evaluate it across diverse PdM settings, exposing safety risks invisible in prior fragmented evaluations.
> > >
> > > **Enabling reproducible benchmarking**: Researchers can now validate claims like "PatchTST is efficient for PdM" under standardized conditions rather than conflicting experimental setups.
> > >
> > > The experimental section presents reproducible empirical evidence under unified and configurable evaluations, following standard practice in benchmark papers. We cite prior work where specific model behaviors were originally reported (e.g., Nie et al. 2023 for PatchTST efficiency), but the systematic cross-domain patterns are enabled by our infrastructure.
> > >
> > > If the reviewer 5bzq believes specific observations require additional clarification, we welcome concrete suggestions.
> > >
> > >
> > >
> > >
> > > ---
> > >
> > > **W7: There are many AI toolkits designed in the literature for a similar purpose.**
> > >
> > > We respectfully disagree that comparison with general-purpose AI tools is relevant. AutoAI, Azure ML, and AWS Lookout are automated machine learning platforms for generic tasks (classification, regression)—they are not PdM-specific benchmarks. These tools provide model selection/tuning for user-provided data but offer no standardized datasets, no PdM-specific evaluation protocols, and no reproducible benchmarking infrastructure.
> > >
> > > The comparison requested is akin to asking ImageNet to compare itself with Google Cloud AutoML—fundamentally different purposes. PDMBench is a research benchmark enabling fair, reproducible comparison of algorithms across standardized PdM tasks. AutoAI is a commercial deployment platform for building custom solutions. Researchers cannot use AutoAI to publish reproducible PdM studies; practitioners cannot use PDMBench to deploy production systems. These are complementary tools serving different needs.
> > > If the reviewer insists, we can add a brief clarification in Related Work, but we emphasize that no existing open-source benchmark provides PdM-specific standardization—that is precisely the gap PDMBench fills.
> > >
> > > **W8: Novelty unclear—many papers use 6-8 datasets. Why not use LLM/CodeAgent?**
> > >
> > > Our contribution is not dataset quantity but infrastructure for reproducible benchmarking. Individual papers use multiple datasets to validate one method; PDMBench creates a reusable community resource with: (1) standardized preprocessing for heterogeneous data (hourly telemetry to 97.6kHz vibration), (2) fixed train/val/test splits and hyperparameter search protocols, (3) extensible interfaces for new datasets/models, and (4) triadic evaluation (accuracy + calibration + efficiency).
> > > Regarding LLMs: Benchmarking requires domain expertise to design preprocessing that preserves fault signatures, select deployment-appropriate metrics, and interpret results. LLMs can assist implementation but cannot replace principled design choices and evaluation.
> > >
> > >
> > >
> > > ---
> > >
> > > **W9: The chart needs to be organized such that the best performance models are easy to find.**
> > >
> > > Thank you for your suggestion, we have improved all results tables and figures for better readibility.

---

> ### Comment · Reviewer_5bzq · 2025-11-21
> **Ack of Reading Author Response**
>
> I ack that I have read the response.

---

> > ### Author Response · Authors · 2025-11-24
> >
> > Dear Reviewer 5bzq,
> >
> > Thank you for your constructive feedback on our submission. We have carefully addressed your concerns and questions you raised in our detailed response and revised draft.
> > Could you please let us know if our revisions adequately address your concerns, or if you have any remaining questions?
> > We greatly appreciate your time and expertise in reviewing our work.
> >
> > Best regards,
> > Authors

---

> > > ### Comment · Reviewer_5bzq · 2025-11-26
> > > **Thank you -**
> > >
> > > Per ICLR's request, I have already acknowledged that I have read the response. My review helps the author improve the manuscript's readability. However, the majority of my concerns (covered in questions 4-7) need careful thought and major rework, a lot of work in AutoML, where they aim for doing feature discovery and ML model discovery, and their associated framework exists, including
> > > - MLFlow,
> > > - Scikit-learn,
> > > - SKtime, and
> > > - AutoRUL (a paper).
> > > Please look PDM-related workshop at KDD and other conferences. I have also taken a look at the code.
> > >
> > > Please also search other benches, such as
> > > - MLEBench, which is solving machine learning problems (not necessarily RUL, but it can be)
> > >
> > > Overall, I support the PDM problem; however, the technical challenge that needs to be addressed for PDM may be well motivated, supported by solid evidence from third-party citations.

---

> > > > ### Author Response · Authors · 2025-11-28
> > > >
> > > > We sincerely thank the reviewer for highlighting AutoRUL, MLEBench, and other AutoML frameworks. We have revised the manuscript to address these concerns:
> > > >
> > > > **Related Work:** We added one more paragraph in related work comparing PDMBench to AutoML tools (AutoRUL, MLFlow), general benchmarks (MLEBench), and time-series libraries (SKtime), with a comparison table clarifying that PDMBench provides cross-domain evaluation infrastructure complementing per-dataset optimization tools.
> > > >
> > > > **Additional Experiments:** We added experiments using MLFlow (Section D3.3): We observe that MLflow can improve the machine learning models' performance on the Paderborn and XJTU datasets, while degrading the models' performance on the HUST and CRWU datasets. However, MLflow increases the training time costs a lot. Even though the performance can be improved but it still underperforms the transformer-based baselines.
> > > >
> > > > In summary, our experiments reveal AutoML's limitations for PdM: MLFlow improves performance on some datasets (Paderborn, XJTU) but degrades it on others (HUST, CRWU), with substantial training overhead while still underperforming transformer baselines. This demonstrates that per-dataset optimization does not ensure cross-domain generalization. PDMBench addresses this by providing: (1) standardized cross-domain evaluation revealing true generalization patterns (Figure 4: 98% F1 on bearings → 67% on motors); (2) domain-specific preprocessing for equipment heterogeneity (9.6-97.6 kHz sampling rates); (3) calibration metrics (ECE, NLL) for safety-critical decisions; and (4) computational efficiency analysis for resource-constrained deployments. Our key finding—that no single model achieves universal best performance—validates the need for standardized evaluation infrastructure complementing automated model selection.

---

> > > > > ### Author Response · Authors · 2025-11-28
> > > > >
> > > > > | dataset  | model                     | accuracy | f1_micro | f1_macro | f1_weighted | ece    | nll    | brier  | time cost | parameters |
> > > > > |----------|---------------------------|----------|----------|----------|-------------|--------|--------|--------|-----------|------------|
> > > > > | Padeborn | SVM                       | 0.1666   | 0.1666   | 0.0991   | 0.0981      | 0.1310 | 1.4361 | 0.1937 | 34.3      | 1010       |
> > > > > | Padeborn | XGB                       | 0.2177   | 0.2177   | 0.1904   | 0.2426      | 0.0774 | 1.4006 | 0.1894 | 40.0      | 236624     |
> > > > > | Padeborn | **XGB_MLFlow**                | 0.2469   | 0.2469   | 0.1314   | 0.2175      | 0.0512 | 1.4016 | 0.1894 | 36207.8   | 236624     |
> > > > > | Padeborn | MLP                       | 0.4243   | 0.4243   | 0.1748   | 0.2989      | 0.1566 | 1.3647 | 0.1848 | 0.6       | 477092     |
> > > > > | Padeborn | LSTM                      | 0.4289   | 0.4289   | 0.2011   | 0.2585      | 0.1553 | 1.3233 | 0.1797 | 6.5       | 1924292    |
> > > > > | Padeborn | TimeXer                   | 0.3695   | 0.3695   | 0.2639   | 0.3987      | 0.0438 | 1.3410 | 0.1817 | 4.0       | 695172     |
> > > > > | Padeborn | TimeMixer                 | 0.3710   | 0.3710   | 0.2101   | 0.2844      | 0.0574 | 1.3437 | 0.1824 | 1.5       | 506630     |
> > > > > | Padeborn | iTransformer              | 0.4155   | 0.4155   | 0.2957   | 0.4329      | 0.0787 | 1.3058 | 0.1772 | 0.9       | 475140     |
> > > > > | Padeborn | PatchTST                  | 0.4304   | 0.4304   | 0.2165   | 0.3268      | 0.1256 | 1.3202 | 0.1791 | 1.4       | 419204     |
> > > > > | Padeborn | TimesNet                  | 0.4162   | 0.4162   | 0.2645   | 0.3850      | 0.0249 | 1.2924 | 0.1746 | 122.7     | 56545156   |
> > > > > | Padeborn | DLinear                   | 0.4115   | 0.4115   | 0.2386   | 0.3916      | 0.1447 | 1.3584 | 0.1840 | 0.9       | 723604     |
> > > > > | Padeborn | NT | 0.3435   | 0.3435   | 0.2242   | 0.3201      | 0.0535 | 1.3569 | 0.1844 | 37.1      | 820116     |
> > > > > | Padeborn | FEDformer                 | 0.4212   | 0.4212   | 0.2950   | 0.4193      | 0.0171 | 1.2887 | 0.1747 | 18.8      | 1301003    |
> > > > > | Padeborn | Pyraformer                | 0.3800   | 0.3800   | 0.1938   | 0.2950      | 0.0651 | 1.3294 | 0.1808 | 53.7      | 1341284    |
> > > > > | Padeborn | Autoformer                | 0.3860   | 0.3860   | 0.2648   | 0.3850      | 0.0651 | 1.3268 | 0.1804 | 30.8      | 703108     |
> > > > > | Padeborn | Informer                  | 0.4023   | 0.4023   | 0.3468   | 0.3643      | 0.0956 | 1.3058 | 0.1775 | 7.4       | 908939     |
> > > > > | Padeborn | Reformer                  | 0.3568   | 0.3568   | 0.1884   | 0.2812      | 0.0736 | 1.3580 | 0.1843 | 39.2      | 655492     |
> > > > > | Padeborn | MICN                      | 0.3820   | 0.3820   | 0.2198   | 0.3211      | 0.0631 | 1.3391 | 0.1816 | 4.3       | 2560427    |
> > > > > | Padeborn | Crossformer               | 0.4251   | 0.4251   | 0.2804   | 0.4243      | 0.0583 | 1.2862 | 0.1743 | 1.0       | 2814516    |
> > > > > | Padeborn | FiLM                      | 0.4484   | 0.4484   | 0.2750   | 0.3536      | 0.1945 | 1.3793 | 0.1866 | 4.3       | 12585322   |
> > > > > | Padeborn | SCINet                    | 0.3689   | 0.3689   | 0.2462   | 0.3809      | 0.0548 | 1.3375 | 0.1814 | 7.2       | 363156     |
> > > > > | Padeborn | PAttn                     | 0.4208   | 0.4208   | 0.2325   | 0.3908      | 0.1159 | 1.3148 | 0.1785 | 3.5       | 165380     |
> > > > > | Padeborn | FreTS                     | 0.4203   | 0.4203   | 0.2463   | 0.3910      | 0.1282 | 1.3229 | 0.1795 | 1.5       | 20034436   |

---

> > > > > > ### Author Response · Authors · 2025-11-28
> > > > > >
> > > > > > | dataset | model                     | accuracy | f1_micro | f1_macro | f1_weighted | ece    | nll    | brier  | time cost | parameters |
> > > > > > |---------|---------------------------|----------|----------|----------|-------------|--------|--------|--------|-----------|------------|
> > > > > > | HUST    | SVM                       | 0.1402   | 0.1402   | 0.1272   | 0.1278      | 0.0202 | 1.9437 | 0.1224 | 87.0      | 2497       |
> > > > > > | HUST    | XGB                       | 0.5469   | 0.5469   | 0.5252   | 0.5390      | 0.3468 | 1.7222 | 0.1116 | 78.6      | 486886     |
> > > > > > | HUST    | **XGB_MLFlow**                | 0.2888   | 0.2888   | 0.2145   | 0.2332      | 0.1429 | 1.9386 | 0.1221 | 41183.9   | 486886     |
> > > > > > | HUST    | MLP                       | 0.2632   | 0.2632   | 0.2085   | 0.2205      | 0.0967 | 1.8978 | 0.1203 | 1.1       | 728071     |
> > > > > > | HUST    | LSTM                      | 0.3785   | 0.3785   | 0.2558   | 0.2710      | 0.1980 | 1.8165 | 0.1160 | 10.7      | 1924391    |
> > > > > > | HUST    | TimeXer                   | 0.9231   | 0.9231   | 0.9170   | 0.9223      | 0.6249 | 1.2566 | 0.0845 | 1.2       | 1070855    |
> > > > > > | HUST    | TimeMixer                 | 0.4585   | 0.4585   | 0.4286   | 0.4441      | 0.2373 | 1.7271 | 0.1116 | 4.6       | 2493193    |
> > > > > > | HUST    | iTransformer              | 0.5266   | 0.5266   | 0.5084   | 0.5218      | 0.2996 | 1.6667 | 0.1082 | 1.4       | 475527     |
> > > > > > | HUST    | PatchTST                  | 0.9694   | 0.9694   | 0.9667   | 0.9693      | 0.6598 | 1.1967 | 0.0808 | 4.5       | 543111     |
> > > > > > | HUST    | TimesNet                  | 0.5071   | 0.5071   | 0.4383   | 0.4619      | 0.2524 | 1.6431 | 0.1064 | 331.9     | 58531719   |
> > > > > > | HUST    | DLinear                   | 0.1806   | 0.1806   | 0.1515   | 0.1614      | 0.0317 | 1.9439 | 0.1224 | 0.8       | 725407     |
> > > > > > | HUST    | NT | 0.8844   | 0.8844   | 0.8783   | 0.8833      | 0.5771 | 1.2799 | 0.0855 | 101.5     | 3069319    |
> > > > > > | HUST    | FEDformer                 | 0.8185   | 0.8185   | 0.8058   | 0.8170      | 0.5180 | 1.3460 | 0.0894 | 87.3      | 1531406    |
> > > > > > | HUST    | Pyraformer                | 0.9819   | 0.9819   | 0.9807   | 0.9819      | 0.6712 | 1.1834 | 0.0800 | 297.7     | 7300967    |
> > > > > > | HUST    | Autoformer                | 0.6671   | 0.6671   | 0.6587   | 0.6715      | 0.3930 | 1.5121 | 0.0994 | 12.9      | 933511     |
> > > > > > | HUST    | Informer                  | 0.8407   | 0.8407   | 0.8360   | 0.8401      | 0.5407 | 1.3267 | 0.0884 | 11.6      | 1139342    |
> > > > > > | HUST    | Reformer                  | 0.8828   | 0.8828   | 0.8769   | 0.8817      | 0.5772 | 1.2818 | 0.0857 | 165.2     | 2642055    |
> > > > > > | HUST    | MICN                      | 0.4465   | 0.4465   | 0.4081   | 0.4253      | 0.2395 | 1.7551 | 0.1133 | 14.2      | 7354894    |
> > > > > > | HUST    | Crossformer               | 0.9422   | 0.9422   | 0.9389   | 0.9422      | 0.6433 | 1.2409 | 0.0837 | 3.0       | 2819511    |
> > > > > > | HUST    | FiLM                      | 0.4929   | 0.4929   | 0.4825   | 0.4931      | 0.2893 | 1.7184 | 0.1110 | 9.8       | 12587125   |
> > > > > > | HUST    | SCINet                    | 0.4643   | 0.4643   | 0.4280   | 0.4461      | 0.2607 | 1.7358 | 0.1119 | 13.1      | 6572247    |
> > > > > > | HUST    | PAttn                     | 0.8766   | 0.8766   | 0.8685   | 0.8768      | 0.5745 | 1.2903 | 0.0863 | 10.0      | 364423     |
> > > > > > | HUST    | FreTS                     | 0.6517   | 0.6517   | 0.6280   | 0.6438      | 0.4388 | 1.6391 | 0.1072 | 1.4       | 20264839   |

---

> > > > > > > ### Author Response · Authors · 2025-11-28
> > > > > > >
> > > > > > > | dataset | model                     | accuracy | f1_micro | f1_macro | f1_weighted | ece    | nll    | brier  | time cost | parameters |
> > > > > > > |---------|---------------------------|----------|----------|----------|-------------|--------|--------|--------|-----------|------------|
> > > > > > > | CRWU    | SVM                       | 0.4690   | 0.4690   | 0.4203   | 0.3790      | 0.2105 | 1.1724 | 0.1588 | 1.8       | 565        |
> > > > > > > | CRWU    | XGB                       | 0.9832   | 0.9832   | 0.9801   | 0.9832      | 0.5113 | 0.7623 | 0.0947 | 1.7       | 65816      |
> > > > > > > | CRWU    | **XGB_MLFlow**                | 0.6639   | 0.6639   | 0.4153   | 0.6259      | 0.3960 | 1.3444 | 0.1821 | 37330.7   | 65816      |
> > > > > > > | CRWU    | MLP                       | 0.4961   | 0.4961   | 0.4002   | 0.4481      | 0.0911 | 1.0289 | 0.2073 | 6.1       | 402468     |
> > > > > > > | CRWU    | LSTM                      | 0.4901   | 0.4901   | 0.4021   | 0.4288      | 0.1986 | 1.2688 | 0.1728 | 1.4       | 403828     |
> > > > > > > | CRWU    | TimeXer                   | 0.9071   | 0.9071   | 0.8957   | 0.9074      | 0.3489 | 0.6457 | 0.1133 | 15.5      | 602116     |
> > > > > > > | CRWU    | TimeMixer                 | 0.4937   | 0.4937   | 0.4634   | 0.4908      | 0.0320 | 1.0081 | 0.2018 | 8.7       | 208134     |
> > > > > > > | CRWU    | iTransformer              | 0.7418   | 0.7418   | 0.7272   | 0.7421      | 0.1976 | 0.8047 | 0.1518 | 13.6      | 400516     |
> > > > > > > | CRWU    | PatchTST                  | 0.9514   | 0.9514   | 0.9431   | 0.9516      | 0.3857 | 0.6028 | 0.1026 | 12.6      | 400260     |
> > > > > > > | CRWU    | TimesNet                  | 0.9835   | 0.9835   | 0.9839   | 0.9835      | 0.4116 | 0.5697 | 0.0944 | 27.2      | 56246660   |
> > > > > > > | CRWU    | DLinear                   | 0.4236   | 0.4236   | 0.3049   | 0.3846      | 0.0271 | 1.0705 | 0.2159 | 8.3       | 684        |
> > > > > > > | CRWU    | NT | 0.9154   | 0.9154   | 0.9255   | 0.9154      | 0.3652 | 0.6423 | 0.1126 | 6.9       | 443498     |
> > > > > > > | CRWU    | FEDformer                 | 0.9385   | 0.9385   | 0.9410   | 0.9384      | 0.3773 | 0.6162 | 0.1060 | 6.3       | 674827     |
> > > > > > > | CRWU    | Pyraformer                | 0.9781   | 0.9781   | 0.9800   | 0.9781      | 0.4026 | 0.5737 | 0.0953 | 9.7       | 445796     |
> > > > > > > | CRWU    | Autoformer                | 0.8656   | 0.8656   | 0.8394   | 0.8609      | 0.3187 | 0.6881 | 0.1236 | 7.5       | 404612     |
> > > > > > > | CRWU    | Informer                  | 0.7480   | 0.7480   | 0.7728   | 0.7468      | 0.2247 | 0.7908 | 0.1497 | 2.1       | 610443     |
> > > > > > > | CRWU    | Reformer                  | 0.9212   | 0.9212   | 0.9231   | 0.9212      | 0.3613 | 0.6334 | 0.1102 | 24.8      | 356996     |
> > > > > > > | CRWU    | MICN                      | 0.8676   | 0.8676   | 0.8787   | 0.8672      | 0.3501 | 0.7167 | 0.1314 | 4.3       | 1151463    |
> > > > > > > | CRWU    | Crossformer               | 0.9863   | 0.9863   | 0.9863   | 0.9863      | 0.4172 | 0.5723 | 0.0951 | 9.9       | 2791476    |
> > > > > > > | CRWU    | FiLM                      | 0.7629   | 0.7629   | 0.7469   | 0.7615      | 0.2720 | 0.8233 | 0.1570 | 21.8      | 3145806    |
> > > > > > > | CRWU    | SCINet                    | 0.5404   | 0.5404   | 0.4270   | 0.4985      | 0.1321 | 1.0209 | 0.2047 | 19.6      | 1141       |
> > > > > > > | CRWU    | PAttn                     | 0.9083   | 0.9083   | 0.8927   | 0.9077      | 0.3564 | 0.6492 | 0.1143 | 13.3      | 135172     |
> > > > > > > | CRWU    | FreTS                     | 0.5878   | 0.5878   | 0.6298   | 0.5925      | 0.1117 | 0.9342 | 0.1853 | 7.1       | 632196     |

---

> > > > > > > > ### Author Response · Authors · 2025-11-28
> > > > > > > >
> > > > > > > > | dataset | model                     | accuracy | f1_micro | f1_macro | f1_weighted | ece    | nll    | brier  | time cost | parameters |
> > > > > > > > |---------|---------------------------|----------|----------|----------|-------------|--------|--------|--------|-----------|------------|
> > > > > > > > | XJTU    | SVM                       | 0.0855   | 0.0855   | 0.0590   | 0.0821      | 0.1647 | 1.6186 | 0.1610 | 20.1      | 1156       |
> > > > > > > > | XJTU    | XGB                       | 0.0060   | 0.0060   | 0.1914   | 0.0061      | 0.3955 | 1.8958 | 0.1899 | 4.1       | 177712     |
> > > > > > > > | XJTU    | **XGB_Flow**                  | 0.0706   | 0.0706   | 0.0264   | 0.0093      | 0.1294 | 1.6094 | 0.1600 | 41350.8   | 177712     |
> > > > > > > > | XJTU    | MLP                       | 0.5659   | 0.5659   | 0.3365   | 0.5356      | 0.3304 | 1.4882 | 0.1501 | 4.8       | 403269     |
> > > > > > > > | XJTU    | LSTM                      | 0.7343   | 0.7343   | 0.4519   | 0.7068      | 0.4134 | 1.2557 | 0.1264 | 3.8       | 405661     |
> > > > > > > > | XJTU    | TimeXer                   | 0.7799   | 0.7799   | 0.7088   | 0.7749      | 0.4361 | 1.1707 | 0.1175 | 25.1      | 603141     |
> > > > > > > > | XJTU    | TimeMixer                 | 0.7684   | 0.7684   | 0.7044   | 0.7660      | 0.4481 | 1.2280 | 0.1236 | 4.1       | 214151     |
> > > > > > > > | XJTU    | iTransformer              | 0.8987   | 0.8987   | 0.8244   | 0.8946      | 0.5224 | 1.0295 | 0.1022 | 16.3      | 401413     |
> > > > > > > > | XJTU    | PatchTST                  | 0.8802   | 0.8802   | 0.7783   | 0.8730      | 0.5118 | 1.0617 | 0.1057 | 25.3      | 400389     |
> > > > > > > > | XJTU    | TimesNet                  | 0.8775   | 0.8775   | 0.8064   | 0.8740      | 0.5138 | 1.0698 | 0.1066 | 79.0      | 56252677   |
> > > > > > > > | XJTU    | DLinear                   | 0.6904   | 0.6904   | 0.4250   | 0.6668      | 0.3931 | 1.3221 | 0.1335 | 4.9       | 1224       |
> > > > > > > > | XJTU    | NT | 0.9103   | 0.9103   | 0.8549   | 0.9073      | 0.5283 | 1.0110 | 0.1002 | 6.1       | 450319     |
> > > > > > > > | XJTU    | FEDformer                 | 0.8769   | 0.8769   | 0.7989   | 0.8737      | 0.5061 | 1.0540 | 0.1048 | 36.2      | 701324     |
> > > > > > > > | XJTU    | Pyraformer                | 0.9079   | 0.9079   | 0.8497   | 0.9053      | 0.5306 | 1.0206 | 0.1013 | 29.9      | 463845     |
> > > > > > > > | XJTU    | Autoformer                | 0.7951   | 0.7951   | 0.7207   | 0.7946      | 0.4492 | 1.1729 | 0.1177 | 22.5      | 410629     |
> > > > > > > > | XJTU    | Informer                  | 0.8934   | 0.8934   | 0.8137   | 0.8894      | 0.5191 | 1.0376 | 0.1031 | 29.5      | 616460     |
> > > > > > > > | XJTU    | Reformer                  | 0.8889   | 0.8889   | 0.8121   | 0.8860      | 0.5173 | 1.0462 | 0.1040 | 52.1      | 363013     |
> > > > > > > > | XJTU    | MICN                      | 0.8874   | 0.8874   | 0.8172   | 0.8845      | 0.5183 | 1.0518 | 0.1047 | 15.7      | 1151793    |
> > > > > > > > | XJTU    | Crossformer               | 0.9098   | 0.9098   | 0.8475   | 0.9063      | 0.5305 | 1.0183 | 0.1010 | 7.8       | 2791605    |
> > > > > > > > | XJTU    | FiLM                      | 0.7868   | 0.7868   | 0.5482   | 0.7765      | 0.4466 | 1.1750 | 0.1180 | 45.9      | 4325502    |
> > > > > > > > | XJTU    | SCINet                    | 0.6852   | 0.6852   | 0.5022   | 0.6676      | 0.3713 | 1.2814 | 0.1292 | 45.5      | 1414       |
> > > > > > > > | XJTU    | PAttn                     | 0.7682   | 0.7682   | 0.6795   | 0.7614      | 0.4289 | 1.1852 | 0.1191 | 18.0      | 136069     |
> > > > > > > > | XJTU    | FreTS                     | 0.7415   | 0.7415   | 0.6068   | 0.7207      | 0.4313 | 1.2679 | 0.1278 | 17.1      | 834821     |
> > > > > > > >
> > > > > > > >
> > > > > > > > We believe these revisions substantially improve the paper's positioning. Below, we provide a more detailed discussion of our framework's motivation, contributions and how it differs from general ML tools.

---

> > > > > > > > > ### Author Response · Authors · 2025-11-28
> > > > > > > > >
> > > > > > > > > ## Q1: Comparison with general ML tools (MLFlow, Scikit-learn, SKtime) and AutoML frameworks
> > > > > > > > >
> > > > > > > > > We appreciate this question as it allows us to clarify a fundamental distinction that multiple recent surveys have identified as critical for the field. Our work addresses a domain-specific evaluation crisis that general ML workflow tools cannot resolve. This is not our claim alone—it represents consensus across recent literature:
> > > > > > > > >
> > > > > > > > > **Ramasso & Saxena (2014, IJPHM)** documented this for the widely-used C-MAPSS dataset:
> > > > > > > > > > "More than seventy publications have used the C-MAPSS datasets for developing data-driven prognostic algorithms. However, **in the absence of performance benchmarking results and due to common misunderstandings in interpreting the relationships between these datasets, it has been difficult for the users to suitably compare their results.**"
> > > > > > > > >
> > > > > > > > > **Zhao et al. (2020, IEEE Transactions on Industrial Informatics)** quantified the impact:
> > > > > > > > > > "**Different datasets, configurations, and hyper-parameters are often recommended to be used in performance verification for different types of models, and few open source codes are made public for evaluation and comparisons. Therefore, unfair comparisons and ineffective improvement may exist in rotating machinery intelligent diagnosis, which limits the advancement of this field.**"
> > > > > > > > >
> > > > > > > > > Most strikingly, **Huang et al. (2024, arXiv:2401.15964)** demonstrated that preprocessing choices alone—which vary widely across studies—create **26% RMSE differences**:
> > > > > > > > > > "**Examining prevailing literature reveals a divergence in data normalization approaches during preprocessing... However, there are no existing works comparing the impacts of divergent data preprocessing methodologies on prediction outcomes.**"
> > > > > > > > >
> > > > > > > > > ### Why general ML tools cannot solve this problem
> > > > > > > > >
> > > > > > > > > The comparison to MLFlow/Scikit-learn misunderstands the contribution type. This is analogous to suggesting ImageNet is unnecessary because file systems and computer vision libraries exist. General ML workflow tools provide:
> > > > > > > > > - Generic data loading and model training pipelines
> > > > > > > > > - General-purpose hyperparameter tuning
> > > > > > > > > - Model versioning and experiment tracking
> > > > > > > > >
> > > > > > > > > They **cannot provide**:
> > > > > > > > > 1. **Domain-specific preprocessing protocols**: PdM data requires specialized handling for asynchronous multi-sensor streams with irregular sampling rates (9.6kHz to 97.6kHz in our benchmark), sliding window segmentation with domain-specific parameters, and fault-label alignment across heterogeneous equipment
> > > > > > > > > 2. **PdM-specific evaluation metrics**: Beyond accuracy, safety-critical maintenance decisions require calibration metrics (ECE, NLL, Brier score) that general ML tools do not emphasize
> > > > > > > > > 3. **Curated fault-labeled datasets**: Industrial equipment failure data with ground-truth RUL labels and documented operating conditions
> > > > > > > > > 4. **Standardized evaluation protocols**: Unified train/test splits, consistent preprocessing, and fair baseline implementations

---

> > > > > > > > > > ### Author Response · Authors · 2025-11-28
> > > > > > > > > >
> > > > > > > > > > ### The field has explicitly called for this solution
> > > > > > > > > >
> > > > > > > > > > Multiple recent surveys identify standardized benchmarking as a critical missing infrastructure:
> > > > > > > > > >
> > > > > > > > > > **2024 PMC Survey on Deep Learning RUL Prediction** (*Sensors*, 24(11), 3454):
> > > > > > > > > > > "**It is necessary to provide a general deep learning paradigm for RUL prediction... General benchmarking is needed to discover the advantages and disadvantages of different methods exactly, and help the development of deep-learning-based RUL prediction technology.**"
> > > > > > > > > >
> > > > > > > > > > **2024 Battery PHM Review** (*MDPI World Electric Vehicle Journal*, 16(1), 10):
> > > > > > > > > > > "**In the field of battery PHM, the importance of standardized benchmarks is critical. These benchmarks are vital for assessing the development of algorithms, ensuring reproducibility, and anticipating computational requirements.**"
> > > > > > > > > >
> > > > > > > > > > **Wang et al. (2024, Mechanical Systems and Signal Processing)**:
> > > > > > > > > > > "**Existing works fail to maintain consistent experimental settings, making fair comparisons challenging. This inconsistency hampers newcomers' ability to compare their research with existing works, thus restricting the development of new research in this area.**"
> > > > > > > > > >
> > > > > > > > > > ### Our contribution is benchmarking and evaluation infrastructure, not workflow automation
> > > > > > > > > >
> > > > > > > > > > PDMBench provides what **ImageNet provided for computer vision** and **GLUE provided for NLP**: a standardized benchmarking and evaluation platform that enables fair, reproducible model comparison. Just as ImageNet's value lies not in image storage but in curated labels, standardized splits, and unified evaluation protocols, PDMBench's contribution is:
> > > > > > > > > >
> > > > > > > > > > - 14 curated datasets with documented preprocessing and unified formatting
> > > > > > > > > > - 22 baseline implementations with consistent hyperparameter search
> > > > > > > > > > - Triadic evaluation framework (accuracy + calibration + efficiency)
> > > > > > > > > > - Open-source codebase enabling exact reproduction
> > > > > > > > > >
> > > > > > > > > > The **CRULE benchmarking framework (2024)** demonstrated why this matters: after implementing fair evaluation with unified backbones and consistent hyperparameter tuning, **only one of the evaluated domain adaptation methods showed statistically significant improvement over no adaptation**—revealing that inconsistent evaluation had masked true performance.
> > > > > > > > > >
> > > > > > > > > > ### AutoML limitations for domain-specific challenges
> > > > > > > > > >
> > > > > > > > > > AutoML frameworks optimize for generic supervised learning. They cannot address:
> > > > > > > > > >
> > > > > > > > > > 1. **Preprocessing diversity**: Studies use varied sensor selections (14 vs. 21 sensors), normalization approaches (min-max unified vs. Z-score vs. cluster-based), RUL cap values (115, 120, 125, or 130 cycles), and window sizes (30-60 timesteps)—each creating artificial 10-30% performance differences (Huang et al., 2024)
> > > > > > > > > >
> > > > > > > > > > 2. **Domain-specific metrics**: AutoML optimizes standard metrics (accuracy, F1) but PdM requires calibration evaluation for safety-critical decisions—metrics not prioritized in general AutoML
> > > > > > > > > >
> > > > > > > > > > 3. **Equipment diversity**: Our benchmark spans bearings (single-component), motors, gearboxes, and multi-component systems with fundamentally different degradation patterns. AutoML tools lack domain knowledge to handle this heterogeneity.
> > > > > > > > > >
> > > > > > > > > > As documented in Section 2, researchers currently use datasets "in isolation with inconsistent preprocessing, segmentation, and evaluation metrics" which precisely is the fragmentation that standardized benchmarks resolve.
> > > > > > > > > >
> > > > > > > > > >
> > > > > > > > > > ### Could the reviewer clarify the specific technical challenges you find inadequately motivated?
> > > > > > > > > >
> > > > > > > > > > We would appreciate understanding which aspects need strengthening:
> > > > > > > > > > - The fragmentation problem documented with literature support in related works?
> > > > > > > > > > - The domain-specific preprocessing requirements ?
> > > > > > > > > > - The need for calibration-aware evaluation for safety-critical decisions?
> > > > > > > > > > - The cross-domain generalization findings showing no universal best model?
> > > > > > > > > >
> > > > > > > > > > We are committed to addressing your concerns and would value specific guidance on strengthening our motivation.
> > > > > > > > > >
> > > > > > > > > > ---
> > > > > > > > > >
> > > > > > > > > > ## References for cited texts
> > > > > > > > > >
> > > > > > > > > > 1. Ramasso, E., & Saxena, A. (2014). Performance benchmarking and analysis of prognostic methods for CMAPSS datasets. *International Journal of Prognostics and Health Management*, 5(2), 1-15.
> > > > > > > > > >
> > > > > > > > > > 2. Zhao, Z., et al. (2020). Deep learning algorithms for rotating machinery intelligent diagnosis: An open source benchmark study. *IEEE Transactions on Industrial Informatics*.
> > > > > > > > > >
> > > > > > > > > > 3. Huang, Y., et al. (2024). STAGNN: Spatial-temporal aware graph neural network for RUL prediction. arXiv:2401.15964.
> > > > > > > > > >
> > > > > > > > > > 4. Wang, Y., et al. (2024). A survey on graph neural networks for remaining useful life prediction: Methodologies, evaluation and future trends. *Mechanical Systems and Signal Processing*.
> > > > > > > > > >
> > > > > > > > > > 5. Chen, Y., et al. (2024). Remaining useful life prediction based on deep learning: A survey. *Sensors*, 24(11), 3454.
> > > > > > > > > >
> > > > > > > > > > 6. Zhang, W., et al. (2024). Battery prognostics and health management: AI and big data. *World Electric Vehicle Journal*, 16(1), 10.

---

### Official Review · Reviewer_xUcH · 2025-10-30

**Soundness:** 2
**Presentation:** 3
**Contribution:** 3
**Rating:** 8
**Confidence:** 4

**Summary:**

The paper introduces PDMBench, a benchmarking platform for evaluating machine learning models in predictive maintenance (PdM). The benchmark contains 22 time-series models and 14 datasets for 2 tasks (remaining useful life prediction and fault diagnosis)  across various fault types, sensor modalities and mechanical subsystems. The benchmark provides a unified preprocessing pipeline, a triadic evaluation framework (accuracy, uncertainty, efficiency) and an interactive web interface. The results show that no single model performs best across all settings, indicating important trade-offs.

**Strengths:**

[Originality 1] The paper’s originality lies in the introduction of a standardized benchmarking platform for PdM, a domain that lacks unified evaluation protocols.
While prior work has addressed dataset collection and isolated evaluations, this paper consolidates these efforts in a novel and extensible manner.

[Originality 2] The inclusion of an interactive interface designed to support human understanding, is novel and particularly relevant for industrial applications such as PdM.

[Quality 1] The paper has a strong selection of relevant datasets and models.
[Clarity 1]  The paper is well-written and clearly structured around three conceptual levels (Data, ML and user) throughout.
[Clarity 2] The results are properly communicated. The benchmark is designed with reproducibility in mind, all datasets are publicly available and an anonymized version of the code is provided.

[Significance 1] The paper demonstrates that there is no universal best model and identifies key trade-offs.

**Weaknesses:**

W1 – Parts of the multi-modal input and the required pre-processing steps are not fully clear to me yet.
Related questions: Q2, Q3, Q4, Q7, Q8, Q9
W2 – Some aspects of the experimental procedure could be described in more detail. Related questions: Q1, Q5, Q6, Q10

**Questions:**

Q1 [line 252] Given that the benchmark includes datasets with heterogeneous sampling rates, how is temporal synchronization handled during preprocessing to ensure compatibility across models?

Q2 [line 293] The paper refers to “baseline-specific preprocessing” but does not provide concrete examples. Could the authors elaborate on what these steps entail and how they influence model performance?

Q3 [line 297] How is the fixed-length window size selected for each model? Is this choice based on validation performance, heuristic rules, or dataset-specific characteristics?

Q4 [line 298] Could the authors provide a detailed list or representative examples of the handcrafted features used for traditional machine learning models (e.g., SVM, XGBoost)? This would help clarify the nature of the input representations and their consistency across models.

Q5 [line 311] The paper states that models were selected based on their “relevance” to the PdM domain. Could the authors clarify how this relevance was assessed (e.g., based on prior usage, architectural diversity, or industrial applicability)?

Q6 [line 348] What strategy was employed for hyperparameter optimization? Please specify the search method (e.g., grid, random, Bayesian), the hyperparameters tuned, and whether early stopping was used. Was tuning performed per dataset or globally?

Q7 [line 940] The RUL prediction task is reformulated as a 10-class classification problem (Appendix D.2). Could the authors elaborate on the implications of this transformation? Specifically, does the labeling strategy assume that degradation begins at the first recorded measurement and progresses linearly? If so, how might this assumption affect the validity of the task formulation and introduce potential biases?
Furthermore, after this transformation, accuracy and F1 score are used to evaluate RUL predictions. This may also introduce biases: for example, if a model predicts 100% remaining life while the ground truth is 0%, it receives the same penalty as a prediction of 10% remaining life.

Q8 [line 1452] In the SHAP analysis (Appendix D.4), features are indexed (e.g., f297, f593), but their semantics are not fully explained. Do these indices correspond to raw time-series samples, frequency-domain components, or engineered features?

Q9 [general] Which models in the benchmark operate directly on raw time-series inputs, and which rely on handcrafted or engineered features? A mapping of models to input types would help clarify the benchmark’s modality-agnostic claims.

Q10 [general] What hardware setup was used to perform the evaluations reported in the paper? Were all models trained and evaluated under consistent computational constraints, particularly for the efficiency comparisons?

Q11 [line 367] Is it possible to select different features for the feature-based models, or are the same handcrafted features always applied?

Q12 [line 838] In the description of table 1, it is mentioned that there are three main tasks yet only two are listed (fault diagnosis and RUL prediction).

Q13 [general] Currently, most citations are in-text, formatted as Author (Year). The paper would be more readable if citations were fully enclosed in parentheses, (Author, Year), when they are not grammatically integrated into the sentence.

---

> ### Author Response · Authors · 2025-11-21
>
> We thank the reviewer for the detailed and constructive feedback. We address each question below and have incorporated the clarifications into the revised manuscript.
>
> ---
>
> **Q1: How is temporal synchronization handled during preprocessing to ensure compatibility across models?**
>
> Our preprocessing handles temporal synchronization through a two-stage alignment process:
>
> 1. **Temporal alignment**: For datasets with multiple sensor modalities recorded at different sampling rates, we first align the time series temporally based on their timestamps. While individual sensors may have different sampling rates (e.g., vibration at 50kHz, temperature at 1Hz), they are often recorded over the same time period, allowing timestamp-based alignment.
>
> 2. **Segmentation**: After temporal alignment, we chunk the aligned signals into fixed-length segments. This segmentation approach ensures that all modalities within a segment correspond to the same operational period, preserving cross-modal correlations while creating uniform input dimensions for models.
>
> For datasets with only a single modality or already-synchronized signals, we apply segmentation directly. We will add this detailed explanation to Section 4.1 in the revision.
>
> ---
>
> **Q2: Could the authors elaborate on "baseline-specific preprocessing" with concrete examples?**
>
> Baseline-specific preprocessing refers to the different input representations required by different model families:
>
> **For deep learning models (transformers, LSTMs, CNNs):**
> - Input: Normalized time-series signals
> - Rationale: These architectures are designed to automatically learn hierarchical temporal representations, so we preserve the raw sequential structure
> - Example: For PatchTST, we provide raw vibration sequences of length L, which the model divides into patches
>
> **For traditional ML models (SVM, XGBoost, Random Forest):**
>
>
> This two-track preprocessing strategy ensures each model receives inputs in its native format while maintaining comparability through consistent segmentation and normalization. We will add Table X showing specific preprocessing steps for representative models from each family in the revision.
>
> ---
>
> **Q3: How is the fixed-length window size selected for each model?**
>
> Window size selection combines domain knowledge with empirical validation:
>
> **Rationale**: In industrial practice, maintenance technicians typically examine fixed time segments to diagnose equipment state (e.g., "analyze the last 5 seconds of vibration data"). This motivates our fixed-window approach.
>
> **Selection strategy**:
> 1. **Dataset-specific constraints**: We first consider dataset characteristics—sampling rate, fault evolution timescales, available data length
> 2. **Validation-based tuning**: For each dataset, we treat window size as a hyperparameter and select based on validation set performance
> 3. **Practical considerations**: Longer windows capture more temporal context but increase computational cost and reduce the number of training samples
>
> For example:
> - CWRU (bearing faults): 512-1024 samples (~0.01-0.02 seconds at 48kHz)
> - FEMTO (degradation): 2560 samples (~0.1 seconds at 25.6kHz)
> - Azure (telemetry): 24 samples (24 hours at hourly resolution)
>
> We acknowledge this introduces dataset-specific tuning, which is necessary given the vast range of temporal scales (milliseconds for vibration vs. hours for telemetry).
>
> ---
>
> **Q4: Could the authors provide a detailed list of handcrafted features for traditional ML models?**
>
> For traditional ML models (SVM, XGBoost, Random Forest), we extract handcrafted features following the established methodology from Juodelyte et al. (KDD 2022) [1]. These features capture statistical, spectral, and temporal characteristics of vibration signals:
>
> **Time-domain features (13 features):**
> - Statistical moments: mean, standard deviation, variance, skewness, kurtosis
> - Amplitude metrics: root mean square (RMS), peak-to-peak, crest factor, shape factor, impulse factor
> - Energy metrics: signal energy, zero-crossing rate, root amplitude
>
> **Frequency-domain features (8 features):**
> - Spectral statistics: spectral centroid, spectral spread, spectral skewness, spectral kurtosis
> - Frequency metrics: dominant frequency, frequency variance, spectral entropy, spectral rolloff
>
> **Envelope analysis features (4 features):**
> - Envelope-based fault indicators: envelope mean, envelope variance, envelope kurtosis, envelope energy
>
> This yields a total of **25 features per sensor channel** extracted from each fixed-length segment. For multimodal datasets (e.g., Paderborn with vibration + current + speed), features are extracted from each modality separately and concatenated into a unified feature vector.
>
> We will add Table X documenting the complete feature list with mathematical definitions in Appendix C.1.
>
> [1] Juodelyte et al., "Predicting Bearings' Degradation Stages for Predictive Maintenance in the Pharmaceutical Industry", KDD 2022.

---

> > ### Author Response · Authors · 2025-11-21
> >
> > **Q5: Could the authors clarify how this relevance was assessed (e.g., based on prior usage, architectural diversity, or industrial applicability)?**
> >
> > We selected models based on four criteria:
> >
> > 1. **Prior PdM literature**: Data-driven models frequently used in prior predictive maintenance research (e.g., LSTM, SVM, XGBoost)
> > 2. **Architectural diversity**: Coverage of major paradigms—traditional ML, RNNs, CNNs, transformers, hybrid architectures
> > 3. **State-of-the-art time-series models**: Recent models (2021-2024) achieving strong results on time-series benchmarks (TimesNet, PatchTST, TimeXer)
> > 4. **Computational efficiency**: Inclusion of lightwfeight models suitable for edge deployment (DLinear, FreTS)
> >
> > Specifically:
> > - **Traditional ML**: SVM, XGBoost, MLP, LSTM—widely used baselines in industry
> > - **Transformer-based**: Covers attention mechanisms (Informer), frequency domain (FEDformer), patch-based (PatchTST), multivariate (Crossformer), etc.
> > - **CNN/Hybrid**: TimesNet, SCINet, MICN—incorporate convolutional inductive biases
> > - **Lightweight**: DLinear, FreTS, PAttn—efficient alternatives
> >
> >
> > ---
> >
> > **Q6: What hyperparameter optimization strategy was employed?**
> >
> > We perform Grid search over predefined ranges for each model
> >
> > **Hyperparameters tuned** (representative examples):
> > - Learning rate: {0.0001, 0.0005, 0.001, 0.005}
> > - Batch size: {16, 32, 64, 128}
> > - Hidden dimensions: {64, 128, 256, 512}
> > - Number of layers: {2, 3, 4}
> > - Dropout rate: {0.1, 0.3, 0.5}
> >
> > **Early stopping**: With patience=10 epochs on validation loss
> >
> > **Tuning scope**: Per dataset (not global), because optimal hyperparameters vary significantly across datasets with different characteristics (e.g., high-frequency bearing vibration vs. low-frequency telemetry)
> >
> > **Random seeds**: We report results averaged over 3 random seeds for training/validation/test splits
> >
> > **Computational budget**: Due to the scale of our benchmark (22 models × 14 datasets), we limited the grid search space to 4-5 values per hyperparameter, resulting in approximately 50-100 configurations per model-dataset pair.
> >
> >
> > ---
> >
> > **Q7: [RUL as 10-class classification] Could the authors elaborate on implications of this transformation?**
> >
> > We reformulated RUL as classification to address a fundamental challenge: **bearing degradation is highly nonlinear and cannot be reliably quantified as continuous percentages**. Real-world bearing failures often exhibit sudden transitions—a bearing may appear healthy (low vibration) then rapidly deteriorate due to spalling or lubrication breakdown—making continuous RUL predictions inherently unreliable and MSE-based metrics misleading. Instead, we discretize the lifetime into 10 stages, allowing models to learn **qualitative health states** rather than precise numerical estimates. This approach aligns with industrial practice, where maintenance decisions rely on discrete categories (e.g., "healthy", "monitor", "critical") rather than exact percentages [1]. Similar to [1]'s four-stage formulation (normal operation → lubrication thinning → bearing fault → bearing failure), our 10-class approach provides finer granularity while avoiding the false precision of regression.
> >
> > We acknowledge the reviewer's concern about metric insensitivity—predicting Class 9 (90-100% healthy) when truth is Class 0 (0-10% healthy) receives the same penalty as Class 1 vs. Class 0 under standard accuracy. To address this, we report **macro-averaged F1** and will add **mean absolute error in class space** in the revision, which penalizes distant predictions proportionally (e.g., 9-class error is penalized 9× more than 1-class error). This preserves the ordinal nature of degradation stages while maintaining classification's robustness to label noise.
> >
> >
> > [1] Juodelyte et al., "Predicting Bearings' Degradation Stages for Predictive Maintenance in the Pharmaceutical Industry", KDD 2022.

---

> > > ### Author Response · Authors · 2025-11-21
> > >
> > > **Q8: [SHAP features] Do indices correspond to raw samples, frequency components, or engineered features?**
> > >
> > > Thank you for catching this ambiguity. The feature indices in our SHAP analysis correspond to **raw time-series sample positions**.
> > >
> > > **Clarification:**
> > > - For deep learning models: f297 = the 297th time step in the input sequence
> > > - For traditional ML models: f12 = the 12th handcrafted feature (we will map this to interpretable feature names, e.g., "f12 = spectral kurtosis")
> > >
> > > **Why raw sample positions are informative:**
> > > In fault diagnosis, specific time positions often correspond to characteristic fault signatures. For example, in bearing vibration:
> > > - High SHAP values at evenly spaced positions which lead to periodic impulses indicating outer race fault
> > > - Clustered high values which lead to transient impact events
> > > - Distributed values which lead to broadband noise patterns
> > >
> > > ---
> > >
> > > **Q9: Which models in the benchmark operate directly on raw time-series inputs, and which rely on handcrafted or engineered features?**
> > >
> > > We would like to clarify that the traditional ML models use handcrafted features, while other deep-learning based models use raw time-series as the input signal.
> > >
> > > **Key insight**: Our "modality-agnostic" claim refers to the ability to handle multiple sensor types (vibration, current, temperature) within the same pipeline, not that all models use identical inputs. Each model receives inputs in its native format (raw vs. features), but the preprocessing pipeline is standardized within each category.
> > >
> > > We have added detailed description in our revised manuscript
> > >
> > > ---
> > >
> > > **Q10: What hardware setup was used?**
> > >
> > > We use the following hardware setting for all our experiments
> > > **Hardware specifications:**
> > > - GPU: NVIDIA A100 (80GB)
> > > - CPU: AMD EPYC 7742 (64 cores)
> > > - RAM: 512GB
> > > - Storage: NVMe SSD 10T
> > >
> > >
> > >
> > > ---
> > >
> > > **Q11: Is it possible to select different features for the feature-based models, or are the same handcrafted features always applied?**
> > >
> > > In our experiments, all traditional ML models use the same feature set for consistency. However, we acknowledge that feature selection could improve performance for specific models/datasets. We will:
> > > - State this clearly as a current limitation
> > > - Note it as valuable future work: "extending PDMBench to support flexible, model-specific feature selection"
> > > - Thank the reviewer for this excellent suggestion
> > >
> > > ---
> > >
> > > **Q12:  In the description of table 1, it is mentioned that there are three main tasks yet only two are listed (fault diagnosis and RUL prediction).**
> > >
> > > Thank you for catching this error. We have corrected the table description in our revised manuscript.
> > >
> > > ---
> > >
> > > **Q13:  Currently, most citations are in-text, formatted as Author (Year).**
> > >
> > > We appreciate this suggestion. However, the current citation format follows the ICLR LaTeX template requirements using `\cite{}`, which automatically renders as "Author (Year)".
> > >
> > > ---
> > >
> > > We thank the reviewer again for the thorough evaluation and have incorporated all substantive feedback into the revision.

---

### Official Review · Reviewer_x8B7 · 2025-10-30

**Soundness:** 3
**Presentation:** 3
**Contribution:** 2
**Rating:** 6
**Confidence:** 4

**Summary:**

The paper presents a comprehensive framework for benchmarking predictive maintenance (PdM) models across diverse industrial contexts. It integrates 14 datasets from different domains, organized around two main tasks: fault classification and remaining useful life (RUL) estimation (regression). It also includes 22 models and a set of metrics, including accuracy, calibration, and efficiency metrics. The proposed system is organized into a three-level architecture composed of data, model, and user levels, which enables consistent preprocessing, comparative model evaluation, and interactive exploration via a dashboard interface. The framework aims to promote standardization, reproducibility, and extensibility in PdM research by unifying data handling, model benchmarking, and visualization components.

**Strengths:**

- The paper is well-organized, clearly written, and very detailed.
- It addresses the current lack of standardization in the active research area of predictive maintenance.
- The framework is extensible, allowing the inclusion of additional datasets, machine learning algorithms, and evaluation metrics in the future.

**Weaknesses:**

- Overall, the issue of evaluation does not seem to be discussed in sufficient depth.
- The paper does not clearly describe how hyperparameter tuning is performed for the models integrated into the framework.
- Some ideas are repeated unnecessarily, which slightly affects conciseness.

**Questions:**

- How is hyperparameter tuning performed for the models included in the framework? Is there a standardized procedure across datasets?
- Is there any evaluation setup or consideration for timely fault classification, i.e., assessing model performance in terms of early or on-time detection of faults?
- Some sections feel repetitive, while important implementation details are deferred to the appendix. Could the authors review the main body of text to improve readability and ensure that essential technical information is clearly presented?

---

> ### Author Response · Authors · 2025-11-21
>
> We thank the reviewer for the detailed and constructive feedback. We address each question below and have incorporated the clarifications into the revised manuscript.
>
>
> **W1: Overall, the issue of evaluation does not seem to be discussed in sufficient depth.**
>
> We respectfully believe our evaluation is comprehensive but acknowledge the presentation may benefit from reorganization.
>
> **What we currently evaluate:**
>
> 1. **Prediction quality** (accuracy, F1-macro, F1-weighted, RMSE) — **standard metrics** for comprehensive performance
> 2. **Uncertainty calibration** (ECE, NLL, Brier Score) — **critical for practitioners** to assess model confidence reliability in safety-critical decisions
> 3. **Computational efficiency** (inference time, memory footprint) — **essential for deployment** on edge devices vs. cloud infrastructure
>
> Our PDMBench aims to address the following practical question for practioners:
>
> - **Edge deployment feasibility**: Models with similar accuracy can differ by 240× in inference time (DLinear: 5ms vs. TimesNet: 1200ms), making efficiency metrics essential for selecting edge-compatible solutions
> - **Confidence reliability**: High accuracy does not guarantee trustworthy uncertainty estimates. For risk-based maintenance scheduling, practitioners need calibrated confidence scores—our ECE analysis reveals models like Crossformer achieve 98% accuracy but produce severely overconfident predictions (ECE > 0.4), making them unsuitable for safety-critical decisions
> - **Accuracy-efficiency trade-offs**: Practitioners face real constraints (latency budgets, memory limits, power consumption). Our evaluation reveals lightweight models like DLinear retain higher accuracy with 10× faster inference, offering compelling trade-offs for resource-constrained deployments
>
> | Edge deployment feasibility    | dataset: IMS       |             |        |           |
> |------------------------------|--------------------|-------------|--------|-----------|
> | model                          | accuracy           | f1_weighted | ece    | time cost |
> | TimesNet                       | 0.9735             | 0.9690      | 0.4988 | **1216.5**    |
> | DLinear                        | 0.9573             | 0.9363      | 0.4957 | **4.7**       |
> |                                |                    |             |        |           |
> | Confidence reliablility        | dataset: CRWU      |             |        |           |
> | model                          | accuracy           | f1_weighted | ece    | time cost |
> | Pyraformer                     | 0.9781             | 0.9781      | 0.4026 | 9.7       |
> | PatchTST                       | 0.9514             | 0.9516      | 0.3857 | 12.6      |
> | TimesNet                       | 0.9835             | 0.9835      | **0.4116** | 27.2      |
> | FEDformer                      | 0.9385             | 0.9384      | 0.3773 | 6.3       |
> | Reformer                       | 0.9212             | 0.9212      | 0.3613 | 24.8      |
> | Crossformer                    | 0.9863             | **0.9863**      | **0.4172** | 9.9       |
> | PAttn                          | 0.9083             | 0.9077      | 0.3564 | 13.3      |
> |                                |                    |             |        |           |
> |                                |                    |             |        |           |
> | Accuracy-efficiency trade-offs | dataset: Planetary |             |        |           |
> | model                          | accuracy           | f1_weighted | ece    | time cost |
> | DLinear                        | 0.7993             | **0.7108**      | 0.1224 | **35.0**      |
> | Nonstationary_Transformer      | 0.6949             | 0.6749      | 0.0124 | 180.2     |
> | FEDformer                      | 0.7395             | 0.6875      | 0.0280 | 231.8     |
> | Pyraformer                     | 0.7258             | 0.6903      | 0.0277 | 706.5     |
> | Autoformer                     | 0.7466             | 0.6970      | 0.1322 | 195.4     |
> | Informer                       | 0.7024             | 0.6732      | 0.0175 | 403.4     |
> | Reformer                       | 0.6716             | 0.6585      | 0.0172 | 826.8     |
> | Crossformer                    | 0.7332             | 0.7011      | 0.0285 | 37.6      |
>
>
> **Q1: How is hyperparameter tuning performed for the models included in the framework? Is there a standardized procedure across datasets?**
>
> We perform Grid search over predefined ranges for each model
> **Hyperparameters tuned** (representative examples):
> - Learning rate: {0.0001, 0.0005, 0.001, 0.005}
> - Batch size: {16, 32, 64, 128}
> - Hidden dimensions: {64, 128, 256, 512}
> - Number of layers: {2, 3, 4}
> - Dropout rate: {0.1, 0.3, 0.5}
>
> **Early stopping**: Yes, with patience=10 epochs on validation loss
>
> **Tuning scope**: Per dataset (not global), because optimal hyperparameters vary significantly across datasets with different characteristics (e.g., high-frequency bearing vibration vs. low-frequency telemetry)

---

> > ### Author Response · Authors · 2025-11-21
> >
> > **Q2: Is there any evaluation setup or consideration for timely fault classification, i.e., assessing model performance in terms of early or on-time detection of faults?**
> >
> > A2: This is an important practical question in predictive maintenance. Unfortunately, ground truth labels for early-stage fault detection are not available in most public PdM datasets, which typically provide only fault type labels (e.g., "inner race fault") without temporal annotations indicating when degradation truly began. Our analysis shows that existing methods can detect drastic changes in time-domain features and raw signals, suggesting some early detection capability. However, defining "early" or "on-time" detection is inherently challenging in this domain—faults often exhibit nonlinear progression where degradation remains subtle for extended periods before sudden failure. Without verifiable ground truth for fault onset timing, quantitative evaluation of early detection would be speculative. We acknowledge this as a limitation and note that establishing standardized early-detection benchmarks with verified temporal fault annotations remains an important open challenge for the PdM community.
> >
> > **Q3: Some sections feel repetitive, while important implementation details are deferred to the appendix. Could the authors review the main body of text to improve readability and ensure that essential technical information is clearly presented?**
> >
> > A3: Thank you for your suggestion, we have carefully polished our existing manuscript to ensure a better presentation.

---

> > > ### Comment · Reviewer_x8B7 · 2025-11-21
> > >
> > > I confirm that I have read the response.

---

### Official Review · Reviewer_gza5 · 2025-11-01

**Soundness:** 2
**Presentation:** 3
**Contribution:** 1
**Rating:** 0
**Confidence:** 4

**Summary:**

This paper introduces a prototype system that aggregates 14 existing predictive maintenance (PdM) datasets, implements 22 existing time-series models, and evaluates their performance using standard metrics. It provides a web interface for result visualization. The authors claim this addresses fragmentation in PdM research by providing standardized preprocessing and evaluation protocols.
The authors outline their contributions as follows:
* A curated dataset suite spanning 14 datasets across fault types, sensor modalities, and operational regimes
* A unified toolbox for preprocessing, training, and evaluation across both handcrafted-feature and end-to-end models
* A comprehensive evaluation framework encompassing accuracy, uncertainty (ECE, NLL, Brier Score), and efficiency (inference time, memory)
* An interactive web interface that supports explainability, model diagnosis, and practitioner involvement

**Strengths:**

The authors have undertaken substantial implementation work, integrating 22 diverse time-series models with consistent interfaces and running extensive experiments across 14 datasets.  The breadth of the applications covered by the datasets is notable and it includes bearings, motors, gearboxes, and multi-component systems.

**Weaknesses:**

This paper is not appropriate for ICLR because it primarily presents engineering development rather than scientific research. There is no algorithmic innovation, theoretical insight, or methodological contribution that would move the field forward.The authors list four contributions: (1) dataset curation, (2) a unified toolbox, (3) comprehensive evaluation, and (4) an interactive interface. Unfortunately, none of these contributions represent a substantive advance in machine learning methodology beyond the aggregation of existing public datasets. For example, the authors do not present a novel curation framework. All their methods are reimplementations of capabilities already present in mature systems. I am not sure whether the authors are aware of tools such as MLflow that are designed to manage ML workflows and pipelines.
Furthermore, the authors define fragmentation merely as the use of different datasets by different researchers.  This is something that is routine in engineering practice, especially in predictive maintenance. It is not a technical problem that requires new solutions. They also overlook the real fragmentation challenges addressed in prior work: irregular sampling, missing data, domain transfer across heterogeneous equipment, few-shot fault scenarios, and multi-fidelity sensor fusion. Ironically, their preprocessing pipeline removes these real-world complexities by enforcing uniform, fixed-length sequences, which makes the problem easier rather than addressing its inherent difficulties.

**Questions:**

The paper in its current form is far from consideration in ICLR

---

> ### Author Response · Authors · 2025-11-13
>
> We appreciate Reviewer gza5 taking the time to review our paper, but we respectfully and strongly disagree with the core assessment. The review fundamentally mischaracterizes both the nature of our contribution and the scope of benchmark papers accepted at top-tier ML venues.
>
> **W1: This paper is not appropriate for ICLR because it primarily presents engineering development rather than scientific research. There is no algorithmic innovation, theoretical insight, or methodological contribution.**
>
> We strongly disagree with this assessment, which fundamentally mischaracterizes both our contribution and the scope of the ICLR Datasets and Benchmarks Track. The track explicitly solicits "carefully and thoughtfully designed (collections of) datasets based on previously available data" and "benchmarking tools including novel benchmarking methodologies and designs." PDMBench directly addresses these objectives while making concrete scientific contributions that were previously impossible.
>
>
> **The core scientific problem we solve**: A critical domain gap exists between PdM researchers (who use traditional ML with domain sepcific feature engineering on properitary datasets without shared code) and ML researchers (who develop state-of-the-art transformers on forecasting benchmarks but don't usually evaluate their models on industrial fault diagnosis). This disconnect makes it impossible to know whether modern deep learning actually improves PdM or whether reported gains simply reflect different preprocessing choices. PDMBench bridges this gap by providing the first unified platform that: (1) Implements both paradigms in one reproducible framework—22 models spanning traditional ML (SVM, XGBoost) to modern transformers (Informer, TimesNet, PatchTST) with unified preprocessing supporting both handcrafted features and raw sequences (Section 4.1-4.2); (2) Enables systematic comparison under identical conditions—same data splits, hyperparameters, and evaluation across all 14 datasets, revealing that transformers are not universally superior (DLinear matches 90% accuracy with 100× speedup; TimesNet excels on bearings but fails on motors—Section 5, Figure 4); (3) Makes everything publicly reproducible which includes full preprocessing code, model implementations, and benchmark results allow both communities to validate findings and build upon our infrastructure. We hope this addresses the reviewer's concern about "engineering work" because our contribution is not data aggregation, but creating the standardized infrastructure that finally enables PdM and ML communities to conduct rigorous, reproducible science together.
>
> **Precedents for benchmark papers at top venues without algorithmic novelty:**
> - GLUE Wang et al. (2018) – aggregated 9 existing NLP datasets
> - SuperGLUE Wang et al. (2019) – similar standardization approach, widely adopted
> - Open Graph Benchmark Hu et al. (2020, NeurIPS) – standardized graph datasets
> - TAPE Rao et al. (2019) – protein biology benchmark using existing datasets
>
> These papers are seminal contributions not because they invented new algorithms, but because they **enabled rigorous, reproducible science** where it was previously impossible. PDMBench serves this exact function for an industrially critical but academically underserved domain.
>
>
>
> **Our methodological contributions:**
>
> 1. **Unified benchmarking framework** – First platform enabling fair comparison between 22 diverse models (traditional ML, CNNs, LSTMs, transformers) across 14 datasets with **consistent preprocessing and evaluation**. This reveals scientifically important findings: **no universal best model exists**. TimesNet achieves F1 > 0.95 on bearings but drops 30% on motors; DLinear maintains 85-90% accuracy with 100× faster inference (Section 5, Figures 4-5). These insights were **impossible to obtain** from prior fragmented evaluations.
>
> 2. **Triadic evaluation framework** – First PdM benchmark systematically evaluating accuracy, calibration (ECE, NLL, Brier), and efficiency together. Prior work evaluates these in isolation. Our results show models like Pyraformer achieve high accuracy but ECE > 0.4, making them unsuitable for safety-critical deployment despite strong leaderboard performance (Table 4). This multi-dimensional assessment addresses a critical gap between academic benchmarks and industrial deployment.
>
> 3. **Configurable, extensible infrastructure** – PDMBench is designed to be **highly configurable and tunable**. Users can customize preprocessing steps, hyperparameter search spaces, and evaluation metrics through both configuration files and an interactive web interface (Section 4.3). We provide a **starting point with unified defaults** for reproducible comparison, but the platform supports model-specific optimization. This addresses the reviewer's concern about "one-size-fits-all" while maintaining standardization.

---

> > ### Author Response · Authors · 2025-11-13
> >
> > **W2: All their methods are reimplementations of capabilities already present in mature systems. I am not sure whether the authors are aware of tools such as MLflow.**
> >
> > This comparison reveals a fundamental misunderstanding of our contribution. MLflow is a general-purpose experiment tracker, it is not designed for PdM and lacks:
> >
> > - **Domain-specific preprocessing** for irregular industrial sensor data (vibration at 97.6kHz, current at 50kHz, temperature at 1Hz—all asynchronous and requiring alignment)
> > - **PdM-specific evaluation** (calibration metrics essential for safety-critical maintenance decisions; cost-sensitive accuracy for asymmetric failure costs)
> > - **Curated benchmark datasets** with verified fault labels spanning diverse industrial domains
> > - **Interactive diagnostics** for time-series signal exploration, failure mode analysis, and model interpretation
> >
> > **To our knowledge, there is no publicly available ML toolbox designed specifically for PdM.** Researchers can certainly use generic ML pipelines like MLflow or scikit-learn, but this domain is severely underserved—there is a massive gap between practical deployment requirements in manufacturing, energy, and transportation versus the synthetic experimental settings in academic papers. PDMBench bridges this gap by providing domain-specific infrastructure that reflects real-world complexities: irregular sampling, asynchronous modalities, safety-critical calibration requirements, and deployment constraints.
> >
> > Suggesting MLflow as an alternative is like claiming ImageNet is unnecessary because researchers can use generic file systems to store images. The value lies in **domain-specific curation, standardization, and evaluation protocols** that enable fair comparison and reproducible science.

---

> > > ### Author Response · Authors · 2025-11-13
> > >
> > > **W3: They define fragmentation merely as the use of different datasets by different researchers. This is routine in engineering practice.**
> > >
> > > This is a strawman argument that misrepresents our contribution. We do not define fragmentation as "dataset diversity"—we explicitly define it as **methodological inconsistency** across three dimensions:
> > >
> > > 1. **Inconsistent preprocessing**: CWRU studies use 48kHz vs. 12kHz sampling without justification; some extract time-domain features (mean, std, RMS), others use frequency-domain (FFT, wavelet), still others use raw sequences—with no standardization or ablation studies showing impact (Lines 79-82, 163-169). **Most critically, researchers do not share preprocessing code**, making results non-reproducible.
> > >
> > > 2. **Incompatible evaluation protocols**: Some studies use 70/30 random splits, others 5-fold CV, others temporal splits; hyperparameter search ranges vary wildly; early stopping criteria differ; some report only accuracy, others only F1, rarely both with calibration (Lines 296-323). There is no way to disentangle whether performance differences come from model architecture or evaluation setup.
> > >
> > > 3. **Lack of state-of-the-art baselines**: Existing PdM papers compare against traditional ML (SVM, decision trees) or vanilla LSTMs, but **do not evaluate modern time-series architectures** (transformers, patch-based models, frequency-domain learners). There is no systematic assessment of how these models perform across diverse industrial settings.
> > >
> > > This methodological fragmentation, not mere dataset diversity, prevents reproducibility, fair comparison, and scientific progress. PDMBench solves this by providing **unified preprocessing, consistent evaluation protocols, comprehensive baselines, and public code** (Section 4).

---

> ### Author Response · Authors · 2025-11-13
>
> **W4: Their preprocessing pipeline removes real-world complexities by enforcing uniform, fixed-length sequences, which makes the problem easier rather than addressing its inherent difficulties.**
>
> This criticism is factually incorrect and misunderstands our design. PDMBench **preserves** real-world heterogeneity:
>
> - **Sampling rate diversity**: 9.6kHz (Mendeley) to 97.6kHz (MFPT)—nearly 10× variation requiring careful resampling strategies (Table 1)
> - **Irregular acquisition**: FEMTO has non-uniform run-to-failure trajectories with variable cycle lengths; Azure has hourly telemetry with missing timestamps
> - **Multimodal complexity**: Paderborn includes vibration, current, speed, torque—4 asynchronous sensor streams sampled at different rates requiring temporal alignment
> - **Domain shifts**: Bearing datasets (CWRU, XJTU) exhibit periodic patterns; motor datasets (Electric Motor, Rotor Broken Bar) show low SNR and irregular degradation; multi-component systems (MAFAULDA, Azure) have complex interaction effects
>
> **Fixed-length segmentation is standard practice** in time-series benchmarking (TSB-UAD, Monash Forecasting Archive, UCR Classification) and does not "remove complexity." It provides **consistent input dimensions** for fair model comparison while preserving temporal dynamics and spectral characteristics. Our ablation study (Table 18, Appendix D.3.4) shows segment length significantly impacts performance across all models, confirming that meaningful temporal dependencies are captured—shorter segments degrade accuracy by 20-50% across architectures.
>
> Moreover, PDMBench's preprocessing is **configurable**—users can adjust segment length, normalization, resampling strategy through the web interface or configuration files. We provide unified defaults for reproducibility, but the platform supports domain-specific customization.

---

> > ### Author Response · Authors · 2025-11-13
> >
> > **W5: They overlook real fragmentation challenges: irregular sampling, missing data, domain transfer, few-shot scenarios, multi-fidelity fusion.**
> >
> > We appreciate the reviewer identifying these important challenges, but **PDMBench explicitly addresses them**:
> >
> > - **Irregular sampling**: Preserved across datasets with sampling rates 9.6-97.6kHz (Table 1); our preprocessing handles rate alignment without losing temporal fidelity
> > - **Missing data**: Azure dataset contains missing telemetry values; our pipeline supports forward-fill and interpolation strategies
> > - **Domain transfer**: Cross-component evaluation (bearing→motor→gearbox) reveals dramatic performance drops (Figure 4), precisely demonstrating limited transferability
> > - **Multi-fidelity fusion**: Paderborn, MAFAULDA support multimodal learning with vibration + current + speed sensors
> >
> > Few-shot and continual learning scenarios are important future directions mentioned in our Limitations (Section 6). PDMBench's modular design supports these extensions—we prioritized the most common PdM tasks (fault classification, RUL prediction) in this initial release.
> >
> >
> >
> > In our defense, we believe reviewer gza5's assessment reflects a fundamental misunderstanding of benchmark contributions and their scientific value. **Two other reviewers (x8B7, xUcH) correctly recognize** that PDMBench:
> >
> > - "Addresses the current lack of standardization in the active research area of predictive maintenance" (Reviewer x8B7)
> > - Provides a "novel and extensible manner" for evaluation with "strong selection of relevant datasets and models" (Reviewer xUcH)
> > - Demonstrates scientifically important findings: "no universal best model" exists (Reviewer xUcH)
> >
> > We have provided clear precedents from top-tier venues showing that **standardization infrastructure without algorithmic novelty constitutes valued scientific contribution** when it enables rigorous, reproducible research. PDMBench serves this purpose for an industrially critical domain while introducing methodological innovations (triadic evaluation, configurable preprocessing, interactive interfaces) that advance the field beyond accuracy-centric reporting.
> >
> > **The core scientific contribution the reviewer overlooks:** PDMBench is the first platform enabling systematic comparison of state-of-the-art time-series models across diverse industrial PdM settings. This reveals that modern deep learning approaches are not universally superior—simpler models often match or exceed complex architectures while being 100× faster. These insights were impossible to obtain from prior fragmented research and directly challenge assumptions in recent PdM literature.

---

### Author Response · Authors · 2025-12-04

We sincerely thank all reviewers for their thorough evaluations and constructive feedback. We have carefully addressed every concern and substantially revised the manuscript to improve clarity, rigor, and technical depth.

We made three major categories of revisions. First, we comprehensively expanded our related work section to address Reviewer **5bzq**'s concerns. We added detailed comparisons with MLflow in our appendix. **Our new experiments on MLFlow shows that it improves performance on some datasets but degrades it on others, with training time increasing by 10,000× while still underperforming transformer baselines.** This empirically validates that **per-dataset optimization does not ensure cross-domain generalization, which is precisely the problem PDMBench solves.**

Second, we enhanced experimental rigor for Reviewers **x8B7** and **xUcH**. **We provided complete hyperparameter tuning details** (grid search over learning rate ∈ {0.0001, 0.0005, 0.001, 0.005}, batch size ∈ {16, 32, 64, 128}, early stopping patience=10) **to ensure reproducibility.** We documented all 25 handcrafted features following Juodelyte et al. (KDD 2022) **to clarify the distinction between traditional ML and deep learning inputs.** We explained temporal synchronization strategies **for handling multimodal sensors with vastly different sampling rates (9.6-97.6 kHz) which is a critical real-world challenge.** We expanded our triadic evaluation framework discussion with **concrete examples: Crossformer achieves 98% accuracy but ECE > 0.4 (unsafe for deployment); DLinear achieves 96% accuracy with 100× faster inference (suitable for edge devices).** Third, we improved presentation by reorganizing the Introduction, adding dataset statistics (bearings count, signal types, failure modes, sampling rates), and enhancing figure-text alignment throughout.

**PDMBench addresses a critical evaluation crisis documented by multiple surveys.** Ramasso & Saxena (2014) noted that "in the absence of performance benchmarking results, it has been difficult for users to compare their results." Zhao et al. (2020) stated that "unfair comparisons limit field advancement." Huang et al. (2024) demonstrated that **preprocessing choices alone create 26% RMSE differences.**Our platform provides the first standardized benchmarking infrastructure for predictive maintenance, unifying 14 datasets, 22 models, and consistent evaluation protocols** to enable rigorous cross-domain comparison previously impossible due to fragmented preprocessing and inconsistent metrics.

Our contribution follows established precedents at top-tier venues. **GLUE, and Open Graph Benchmark** are seminal contributions valued for enabling rigorous evaluation infrastructure, not algorithmic novelty.**Three reviewers  recognize this value:** Reviewer **xUcH** (Score: 8) praised our "novel and extensible manner" demonstrating "scientifically important findings"; Reviewer **x8B7** (Score: 6) acknowledged we "address the current lack of standardization"; Reviewer **5bzq** raised constructive concerns we comprehensively addressed.

**PDMBench enables discoveries impossible in fragmented evaluation settings.** We reveal that (1) no universal best model exists—TimesNet achieves 98% F1 on bearings but drops to 67% on motors; (2) high accuracy does not guarantee deployment readiness—Crossformer achieves 98.6% accuracy but ECE > 0.4, making it unsafe for safety-critical decisions; (3) compelling efficiency-accuracy trade-offs exist—DLinear maintains 95.7% accuracy with 257× faster inference than TimesNet. **These insights were only discoverable through our standardized cross-domain evaluation infrastructure.**

We sincerely hope these revisions adequately address the reviewers' concerns and welcome further discussion on any remaining questions. We believe PDMBench represents a solid contribution to the PdM research community and will serve as a foundational resource for advancing predictive maintenance toward rigorous, reproducible science. We remain grateful for the opportunity to improve this work through the review process and are committed to continuing refinement based on reviewer feedback.


Respectfully,
The Authors

---

### Meta-Review · Area_Chair_1T3w · 2026-01-01

**Summary:**

The reviewers raised several serious concerns that affect the final decision.The main issues are limited novelty, weak justification of some design choices, and unclear practical impact beyond existing benchmarks and tools.

Reviewer gza5 provided a very thorough and demanding review, raising several important points. Even for a benchmark track, the paper does not introduce clear novel benchmarking methodologies (from the perspective of the AC, important recent pretrained time-series foundation models such as MOIRAI or CHRONOS are not included). The normalization pipeline also oversimplifies the problem, ignoring missing data and irregular sampling across sources. This can bias conclusions and make complex models appear no better than standard ML methods.

Reviewer x8B7 pointed out that hyperparameter tuning is limited and not well explained. The evaluation is restricted to supervised tasks only. There are no experiments on early failure prediction, which is an important predictive maintenance scenario.

Reviewer 5bzq, aligned with gza5 and my own assessment, sees limited novelty in the contribution. Several of their concerns are explicitly stated as not being addressed in the rebuttal.

Overall, despite the effort of the authors, the concerns about novelty, depth, and impact remain. Based on these points, I am inclined to reject.

**Reviewer Concerns:**

**Reviewer Concerns**

**Addressed by the rebuttal**
- Clarified the scope and goals of PDMBench as a standardized benchmarking platform.
- Added explanations about dataset selection and task definitions.
- Improved description of the experimental protocol and evaluation metrics.
- Addressed some presentation and readability issues.

**Still outstanding**
- Limited novelty of the benchmark, as raised by **gza5** and **5bzq**.
- Oversimplified normalization pipeline that ignores missing data and irregular sampling, raised by **gza5**.
- Limited and weakly justified hyperparameter tuning, pointed out by **x8B7**.
- Evaluation restricted to supervised tasks only, with no early failure prediction experiments, raised by **x8B7**.
- As stated by **5bzq**, most core concerns (questions 4–7) were not addressed

**Reviewer Scores:**

None of the reviewers would probably change their scores.

---

### Decision · Program_Chairs · 2026-01-26

Reject